# Meiotic nuclear pore complex remodeling provides key insights into nuclear basket organization

Grant A. King[1]*, Rahel Wettstein[2,3]*, Joseph M. Varberg[4], Keerthana Chetlapalli[1], Madison E. Walsh[1], Ludovic C.J. Gillet[5], Claudia Hernández-Armenta[6], Pedro Beltrao[5,6], Ruedi Aebersold[5], Sue L. Jaspersen[4,7], Joao Matos[2,3], and Elçin Ünal[1]

**Nuclear pore complexes (NPCs) are large proteinaceous assemblies that mediate nuclear compartmentalization. NPCs undergo large-scale structural rearrangements during mitosis in metazoans and some fungi. However, our understanding of NPC remodeling beyond mitosis remains limited. Using time-lapse fluorescence microscopy, we discovered that NPCs undergo two mechanistically separable remodeling events during budding yeast meiosis in which parts or all of the nuclear basket transiently dissociate from the NPC core during meiosis I and II, respectively. Meiosis I detachment, observed for Nup60 and Nup2, is driven by Polo kinase-mediated phosphorylation of Nup60 at its interface with the Y-complex. Subsequent reattachment of Nup60-Nup2 to the NPC core is facilitated by a lipid-binding amphipathic helix in Nup60. Preventing Nup60-Nup2 reattachment causes misorganization of the entire nuclear basket in gametes. Strikingly, meiotic nuclear basket remodeling also occurs in the distantly related fission yeast, *Schizosaccharomyces pombe*. Our study reveals a conserved and developmentally programmed aspect of NPC plasticity, providing key mechanistic insights into the nuclear basket organization.**

## Introduction

The nuclear pore complex (NPC) is a conserved supramolecular structure embedded in the nuclear envelope that acts as the gatekeeper between the nucleus and cytoplasm (reviewed by Hampoelz et al., 2019; Lin and Hoelz, 2019). NPCs are composed of multiple copies of ~30 proteins called nucleoporins organized into six modular subcomplexes, which in turn form eight symmetric spokes. Despite its size and complexity, the makeup and structure of the NPCs are surprisingly plastic. Individual NPCs within the same cell can differ in composition (Akey et al., 2022; Galy et al., 2004) and exhibit conformational changes in response to the cellular environment (Schuller et al., 2021; Zimmerli et al., 2021). NPCs also undergo extensive organizational changes during fungal and metazoan mitosis, including partial or full disassembly, that often result in the alteration of nucleocytoplasmic transport (De Souza et al., 2004; Dey et al., 2020; Expósito-Serrano et al., 2020; Laurell et al., 2011; Linder et al., 2017; reviewed in Kutay et al., 2021). The extent of NPC plasticity in many other cellular contexts, however, remains largely uncharacterized.

In the budding yeast *Saccharomyces cerevisiae*, the nuclear envelope and its constituent NPCs remain largely intact during both mitosis and meiosis (King et al., 2019; Moens, 1971; Winey et al., 1997). During mitosis, a cytoplasmic pool of the channel nucleoporin, Nsp1, mediates NPC inheritance to daughter cells (Colombi et al., 2013; Makio et al., 2013). During meiosis, a large-scale NPC turnover event occurs (King et al., 2019). Core NPC subcomplexes (Fig. 1, A and B) are sequestered to a nuclear envelope-bound compartment, the GUNC (for Gametogenesis Uninherited Nuclear Compartment), which remains outside of the gametes and is ultimately degraded during gamete maturation (King et al., 2019; King and Ünal, 2020). In contrast, the entire nuclear basket is inherited: it detaches from the NPC core and returns to nascent gamete nuclei (King et al., 2019). The precise molecular events that control meiotic NPC remodeling are unknown.

The nuclear basket serves as the connection between the nuclear periphery and chromatin, playing roles in diverse nuclear processes, including mRNA export and the DNA-damage response (reviewed in Buchwalter et al., 2019; Strambio-De-Castillia et al., 2010). Organizational understanding of the nuclear basket, however, remains limited due to the highly

[1]Department of Molecular and Cell Biology, University of California, Berkeley, CA;  [2]Institute of Biochemistry, ETH Zürich, Zürich, Switzerland;  [3]Max Perutz Labs, University of Vienna, Vienna, Austria;  [4]Stowers Institute for Medical Research, Kansas City, MO;  [5]Institute of Molecular Systems Biology, Department of Biology, ETH Zürich, Zürich, Switzerland;  [6]European Molecular Biology Laboratory, European Bioinformatics Institute (EMBL-EBI), Wellcome Genome Campus, Cambridge, UK;  [7]Department of Molecular and Integrative Physiology, University of Kansas Medical Center, Kansas City, KS.

*G.A. King and R. Wettstein contributed equally to this paper.   Correspondence to Elçin Ünal: elcin@berkeley.edu;   Joao Matos: joao.matos@univie.ac.at

K. Chetlapalli's current address is Department of Biology, Massachusetts Institute of Technology, Cambridge, MA.

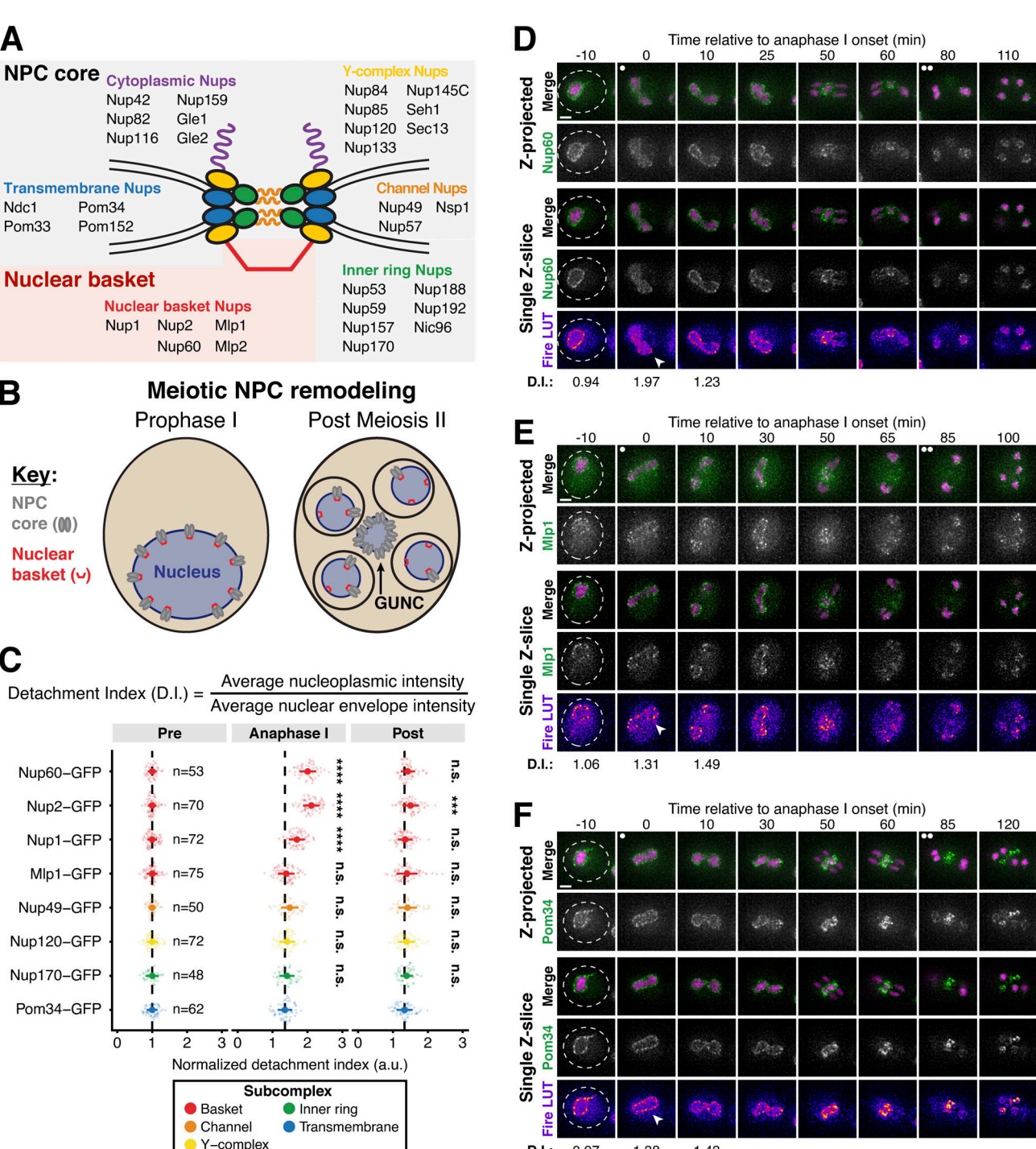

Figure 1. **A subset of basket nucleoporins relocalizes from the nuclear periphery to the nucleoplasm during anaphase I. (A)** Schematic of the NPC, adapted from King et al. (2019). Nup100 and Nup145N are linkers between subcomplexes and are not depicted in the schematic. The gray background denotes the subcomplexes that comprise the NPC core; the red background denotes the nuclear basket. **(B)** A schematic depicting NPC remodeling during meiosis as described in King et al. (2019). Core nucleoporins are sequestered to the GUNC during meiosis II, while basket nucleoporins return to nascent gamete nuclei. **(C)** Quantification of nucleoporin detachment before (–10 min, "Pre"), coincident with (0 min, "Anaphase I"), and after (+10 min, "Post") the onset of anaphase I. The detachment index (DI) for individual cells was calculated from single z-slices by dividing the average nucleoplasmic signal intensity by the average nuclear envelope signal intensity. For each nucleoporin (color-coded by subcomplex), individual DI values were normalized to the average DI at the "Pre" time point. Asterisks indicate statistical significance calculated using Dunn's test for multiple comparisons when each nucleoporin was compared to Pom34-GFP, a transmembrane nucleoporin, for a given time point (see Materials and methods for an explanation as to why mean DI values for Pom34-GFP change at different meiotic stages; see Table S5 for P values). The dashed lines indicate the average DI for Pom34-GFP for each time point. Sample sizes (*n*) are the number of cells quantified for each nucleoporin; for Nup120-GFP and Nup49-GFP, cells from two independent replicates were pooled. For all figures in this article, mean and standard deviation are displayed as a dot and whiskers and significance values are denoted with asterisks: *, P < 0.05; **, P < 0.01; ***, P < 0.001; and ****, P < 0.0001. **(D)** Montage of a cell with Nup60-GFP, a nuclear basket nucleoporin, and Htb1-mCherry, a histone, progressing through meiosis (UB14646).

**(E)** Montage of a cell with Mlp1-GFP, a nuclear basket nucleoporin, and Htb1-mCherry, a histone, progressing through meiosis (UB14648). **(F)** Montage of a cell with Pom34-GFP, a transmembrane nucleoporin, and Htb1-mCherry, a histone, progressing through meiosis (UB13503). For all panels, the onset of anaphase I was defined as the Htb1-mCherry chromatin mass exhibiting distortion from a spherical shape consistent with chromosome segregation. For each montage, normalized DI values are indicated when calculated. The white arrowheads in the "Fire LUT" images denote nuclei at the onset of anaphase I, the stage when Nup60-Nup2 detachment is observed. For all figures in this article, the "Merge" rows display both the GFP and RFP signals together, and the "Fire LUT" (LookUp Table) row displays the GFP signal pseudocolored using the Fire LUT in FIJI (Schindelin et al., 2012). A single white dot (see merged z-projection panels) denotes the time of the meiosis I remodeling event (defined as Nup60-Nup2 relocalization to the nucleoplasm) and two white dots denote the time of the meiosis II remodeling event (defined as near complete nuclear basket return to gamete nuclei). Scale bars, 2 µm.

---

disordered nature of many basket nucleoporins (Cibulka et al., 2022). Five nucleoporins comprise the budding yeast nuclear basket: Nup1, Nup2, Nup60, Mlp1, and Mlp2. Nup1 and Nup60 contain lipid-binding amphipathic helices that bind the nuclear envelope and helical regions (HRs) that bind the NPC core (Mészáros et al., 2015). Nup60 recruits Mlp1 and Nup2 to the NPC via short linear sequence motifs, and Mlp1 is in turn required for Mlp2 localization to the NPC (Cibulka et al., 2022; Dilworth et al., 2001; Feuerbach et al., 2002; Palancade et al., 2005). It is unclear if these features are regulated to achieve nuclear basket detachment during meiosis and whether other currently unknown organizational principles are involved.

In this study, we undertook a mechanistic investigation of nuclear basket remodeling during budding yeast meiosis. Using high time-resolution live-cell fluorescence microscopy, we elucidated two distinct NPC remodeling events in meiosis: partial nuclear basket detachment during meiosis I, involving Nup60 and Nup2, and full nuclear basket detachment during meiosis II. Focusing on the meiosis I remodeling event, we found that partial nuclear basket detachment is coupled to meiotic progression by the Polo kinase Cdc5. We used an unbiased proteomics approach to identify Nup60 as a target of Cdc5-dependent phosphorylation and demonstrated that this phosphorylation drives Nup60 detachment by disrupting its interaction with the NPC core. Nup60 reattachment to the NPC requires its lipid-binding amphipathic helix (AH); this reattachment is necessary for the timely association and organization of the entire nuclear basket in gametes. Differences in dynamics between basket nucleoporins during meiosis I resulted in the discovery of new organizational principles for the nuclear basket, including that Mlp1 can remain associated with the NPC independently of its recruiter Nup60. Notably, meiosis I nuclear basket remodeling is conserved in *Schizosaccharomyces pombe*, a distantly related yeast without NPC sequestration to the GUNC, suggesting that GUNC formation and basket modularity are functionally separable features of NPC remodeling during meiosis. Our study uncovers a new mode of NPC plasticity in a developmental context and provides mechanistic insights into nuclear basket organization.

## Results

### A subset of nuclear basket nucleoporins transiently detaches from the nuclear periphery during meiosis I

We previously demonstrated that the nuclear basket nucleoporins Nup60, Nup2, and Nup1 behave distinctly from the NPC core, returning to gamete nuclei during meiosis II instead of remaining sequestered to the GUNC (Fig. 1 B; King et al., 2019).

To gain a deeper understanding of nuclear basket behavior, we first performed high time-resolution, live-cell imaging of various GFP-tagged nucleoporins in meiotic cells relative to a fluorescently tagged histone (Htb1-mCherry). Surprisingly, in addition to the previously characterized meiosis II event, the nuclear basket exhibited dynamic behavior during meiosis I (Fig. 1). Nup60-GFP and its binding partner Nup2-GFP became transiently nucleoplasmic during anaphase I (Fig. 1, C and D; and Fig. S1, A and B; and Videos 1 and 2), undergoing detachment from and subsequent reattachment to the nuclear periphery. Both changes in localization took place within a narrow timeframe (Fig. S1 A), with detachment coinciding with the onset of anaphase I chromosome segregation (within <5 min for all cells observed) and lasting for ~10 min (mean ± SD: replicate 1 = 10.3 ± 2.9 min, replicate 2: 11.1 ± 3.3 min). Nup1-GFP exhibited a moderate detachment phenotype, with prominent peripheral localization throughout meiosis I (Fig. 1 C and Fig. S1 C, and Video 3), while Mlp1-GFP remained peripheral throughout meiosis I (Fig. 1, C and E; and Video 4). Mlp2-GFP could not be monitored during the meiotic divisions due to its weak signal, likely as a result of lower expression relative to other nuclear basket members (Cheng et al., 2018). Importantly, all members of the NPC core that were tested, including the transmembrane nucleoporin Pom34-GFP and members of three other subcomplexes, remained peripheral throughout meiosis I (Fig. 1, C and F; and Fig. S1, D–F; and Video 5). We also monitored nuclear basket behavior during mitosis and found that, although minor detachment of Nup60 and Nup2 was detectable in anaphase, all basket members largely remained at the nuclear periphery (Fig. S2, A–F). Taken altogether, these data reveal that the nuclear basket is partially disassembled during meiosis I, with Nup60 and Nup2 robustly and transiently detaching from the NPC in a previously overlooked remodeling event.

### The NPC undergoes two distinct remodeling events during budding yeast meiosis

Our microscopy data established that members of the nuclear basket dissociate from the nuclear periphery during both meiotic divisions (Fig. 1); however, the relationship between these two dissociation events remained unclear. Since different subsets of basket nucleoporins underwent detachment during meiosis I and meiosis II, we hypothesized that the nuclear basket undergoes two distinct remodeling events. Accordingly, we predicted that (1) the basket members that detach during both meiosis I and II (Nup60 and Nup2) should be reincorporated into NPCs between the two meiotic divisions and (2) basket nucleoporins exhibiting different dynamics should be able to detach from the NPC independently of one another.

To assess whether the nuclear basket was reassembled after meiosis I, we performed Structured Illumination Microscopy (SIM) of fixed yeast cells containing Nup2-GFP, a basket nucleoporin that detaches during both meiosis I and II, and Pom34-mCherry, a transmembrane nucleoporin marking the NPC core. Nup2 localization is also a proxy for Nup60 localization since Nup60 is necessary for Nup2 recruitment to the NPC (Dilworth et al., 2001). We found that Nup2-GFP colocalized with Pom34-mCherry both before and after meiosis I (Fig. 2 A), indicating that Nup2 and Nup60 were indeed reassociating with individual NPCs between meiosis I and II. Moreover, we confirmed at high-resolution that the NPC core and basket behave differently during meiosis II: Nup2-GFP largely returned to gamete nuclei during meiosis II, while Pom34-mCherry was largely sequestered to the GUNC (Fig. 2 A). In contrast, two core nucleoporins—the Y-complex member Nup84-GFP and the transmembrane Pom34-mCherry—colocalized throughout both meiotic divisions, including upon GUNC formation during meiosis II (Fig. 2 B). The reassociation of Nup60 and Nup2 with NPCs after meiosis I indicates that they indeed detach from the NPC in two distinct events.

To determine if basket nucleoporins exhibiting different meiotic behaviors dissociate from the NPC independently, we tethered individual basket members to the NPC core using the FKBP12-FRB inducible dimerization system and then monitored whether other basket nucleoporins still exhibited detachment during meiosis II (Haruki et al., 2008). Based on the interaction between the nuclear basket and the Y-complex in the cryo-EM structure of the NPC, we tagged the N-terminus of Nup60 or Mlp1 in conjunction with the C-terminus of Seh1 to minimally disrupt native NPC organization (Kim et al., 2018). Tethering of Nup60 (FKBP12-Nup60) to the NPC core (Seh1-FRB) resulted in its sequestration to the GUNC during meiosis II, indicating that active detachment from the sequestered NPC core enables nuclear basket return to gamete nuclei (Fig. 2, C–E and Fig. S3 A). Among the basket nucleoporins tested, only Nup2 followed tethered Nup60 to the GUNC; both Mlp1 and Nup1 were still able to detach from the NPC core and return to gamete nuclei (Fig. 2 F). Nup60 and Nup2, therefore, remain physically coupled during meiosis, suggesting that they form a distinct module that detaches from the NPC together. Tethering Mlp1 (FKBP12-Mlp1) to the NPC core resulted in only its sequestration to the GUNC: Nup2, Nup1, and Nup60 were all able to return to gamete nuclei (Fig. S3, B–D). These results confirm that the nuclear basket does not behave as a uniform entity during meiosis, explaining how nuclear basket nucleoporins are able to exhibit differential meiotic localization patterns.

We therefore propose that two distinct remodeling events occur during budding yeast meiosis, whereby NPCs adopt different forms with varying nucleoporin constituents. During meiosis I, partial nuclear basket disassembly takes place, with Nup60 and Nup2 transiently and robustly detaching from the NPC (Fig. 2 G, "Meiosis I Remodeling"). During meiosis II, full nuclear basket disassembly takes place, with all tested basket nucleoporins (Nup1, Nup2, Mlp1, Nup60) detaching from the NPC and returning to the gamete nuclear periphery (Fig. 2 G, "Meiosis II Remodeling"). Consequently, the nuclear basket is

inherited, while the NPC core remains in the GUNC. Using the Recombination Induced Tag Exchange (RITE) system (Fig. S3 E; Verzijlbergen et al., 2010), we confirmed that the same pool of basket nucleoporins detaches from and reattaches to the nuclear periphery during both anaphase I and II (Fig. S3, F and G). Accordingly, budding yeast meiosis is a developmental context in which the nuclear basket exhibits novel structural plasticity, offering an opportunity to deeply interrogate its organizational principles.

### The Polo kinase Cdc5 is necessary for partial nuclear basket detachment during meiosis I

We next sought to gain mechanistic insights into how nuclear basket detachment is regulated by focusing on the remodeling event that occurs during meiosis I. To more precisely stage when the Nup60-Nup2 detachment took place, we monitored Nup60-GFP in a strain with a fluorescently tagged spindle pole body (SPB) marker, Spc42-mCherry (Fig. 3 A). Nup60 detachment was stereotyped with respect to SPB behavior (Fig. 3 B), taking place ~25 min after metaphase I SPB separation (mean ± SD replicate 1 = 24.2 ± 5.4 min, replicate 2 = 24.5 ± 7.2 min). The rapid and precisely timed detachment of Nup60-Nup2 led us to reason that it may be regulated by a cell cycle-dependent kinase. The Polo kinase Cdc5 was an attractive candidate as it is induced shortly before the meiotic divisions and is necessary for proper meiosis I chromosome segregation (Clyne et al., 2003; Lee and Amon, 2003). Moreover, Polo kinases have been shown to phosphorylate nucleoporins and regulate NPC disassembly during mitosis in human and worm cells (Linder et al., 2017; Martino et al., 2017).

To determine whether the Polo kinase Cdc5 was necessary for nuclear basket remodeling in meiosis I, we utilized a meiotic null (mn) allele of CDC5 where the endogenous promoter is replaced by that of CLB2, a B-type cyclin that is transcriptionally repressed during meiosis (cdc5-mn; Lee and Amon, 2003). We ensured stage-matched comparison between CDC5 and cdc5-mn cells by performing these experiments in cells depleted for the anaphase-promoting complex/cyclosome (APC/C) activator Cdc20 (cdc20-mn) and therefore arrested in metaphase I (Lee and Amon, 2003). We found that Nup60-GFP detached from the nuclear periphery in cdc20-mn cells carrying the wild-type CDC5 allele following SPB separation (Fig. 3, C and E), albeit to a lesser extent than observed in a wild-type meiosis (Fig. 1 C). In contrast, Nup60-GFP remained associated with the nuclear periphery in cdc5-mn cdc20-mn cells (Fig. 3, D and E). Similar results were obtained using the cdc5-mn allele alone (Fig. 3 E and Fig. S4 A). We, therefore, conclude that the Polo kinase Cdc5 couples partial nuclear basket disassembly to meiotic cell cycle progression.

### Ectopic Polo kinase activity is sufficient for partial detachment of the nuclear basket

The lack of Nup60-Nup2 detachment in cdc5-mn mutants could indicate a direct role for Polo kinase in mediating nuclear basket detachment or an indirect role via the facilitation of proper meiotic progression. To distinguish between these two possibilities, we sought to determine whether ectopic Cdc5 activity

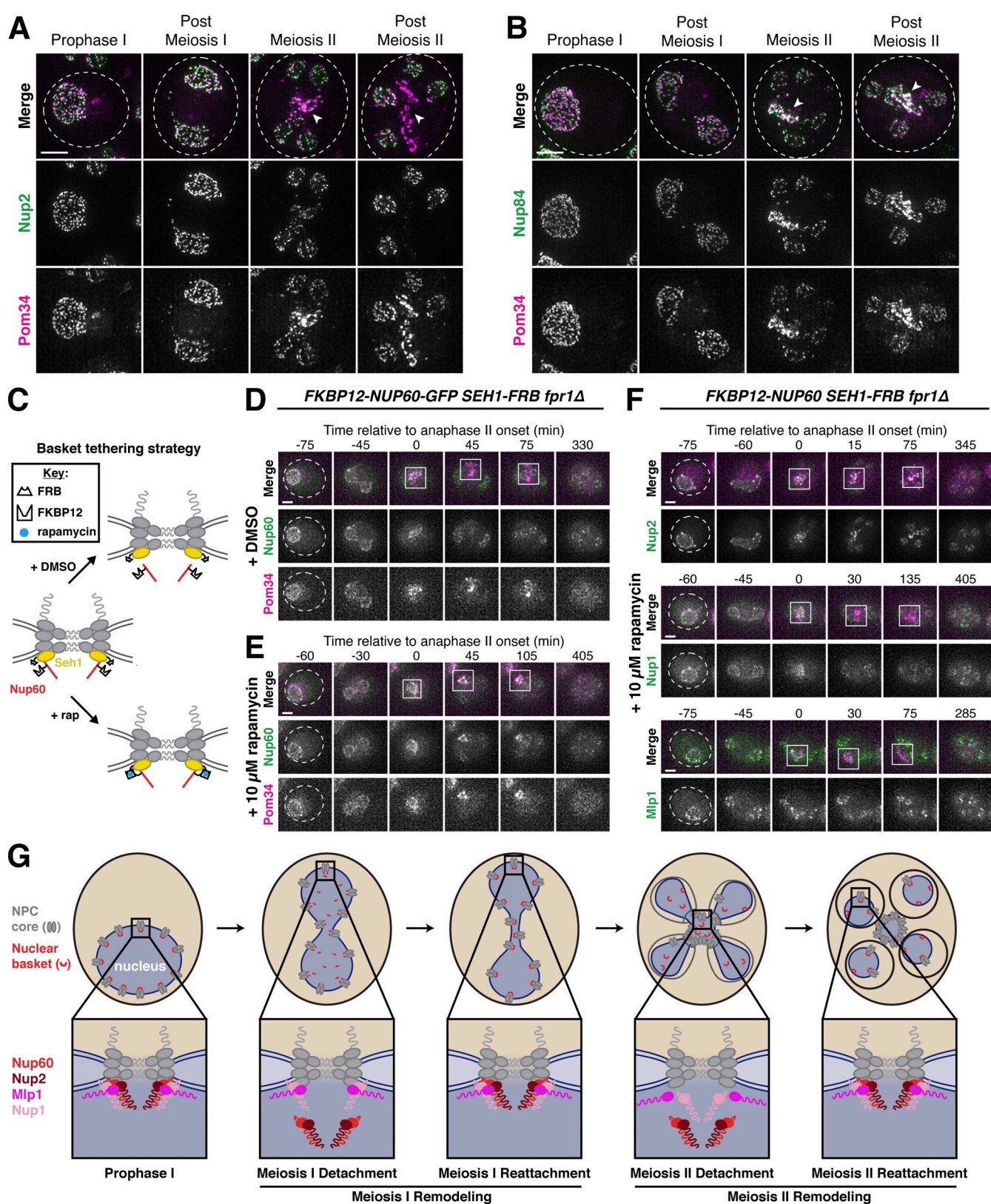

Figure 2. **Two distinct NPC remodeling events occur during budding yeast meiosis. (A)** SIM images of fixed cells with Nup2-GFP, a nuclear basket nucleoporin, and Pom34-mCherry, a transmembrane nucleoporin (UB20080). **(B)** SIM of fixed cells with Nup84-GFP, a Y-complex nucleoporin, and Pom34-mCherry, a transmembrane nucleoporin (UB21079). For A and B, the white arrowheads in the "Merge" images denote the GUNC. **(C)** Schematic of the FKBP12-FRB inducible dimerization approach used to tether Nup60, a nuclear basket nucleoporin, to Seh1, a Y-complex nucleoporin. **(D and E)** Montages of cells containing FKBP12-Nup60-GFP and Seh1-FRB, treated with either (D) DMSO or (E) 10 μM rapamycin after 4 h in SPM (UB27298). **(F)** Montages of cells with

different fluorescently tagged basket nucleoporins—Nup2-GFP (UB25843), Nup1-GFP (UB27143), and Mlp1-GFP (UB27725)—and the inducible Nup60 tether (FKBP12-Nup60 and Seh1-FRB) treated with 10 µM rapamycin after 4 h in SPM. For D–F, the transmembrane nucleoporin Pom34-mCherry was used to monitor the NPC core, with the GUNC indicated by a white box. The onset of anaphase II was defined as the first time point with GUNC formation. All cells were *fpr1Δ* to facilitate rapamycin access to the tether. **(G)** Model depicting the two distinct NPC remodeling events that occur during budding yeast meiosis: (1) partial basket detachment (Nup60 and Nup2) during meiosis I and (2) full basket detachment (Nup60, Nup2, Nup1, and Mlp1) during meiosis II. Note that, although Nup1 is depicted as remaining associated with the NPC core during meiosis I, it exhibits moderate detachment. Mlp2 is not shown as we were unable to monitor its localization during meiosis. Scale bars, 2 µm.

was sufficient to drive partial detachment of the nuclear basket outside of the meiotic divisions. *CDC5* is a direct target of the meiotic transcription factor Ndt80 (Clyne et al., 2003). In the absence of *NDT80* function (*ndt80Δ*), cells successfully enter meiosis but arrest in prophase I, since many of the genes critical for meiotic progression including *CDC5* are not expressed (Chu and Herskowitz, 1998; Xu et al., 1995). As expected, Nup60-GFP remained at the nuclear periphery during *ndt80Δ* prophase I arrest (0 min, Fig. 3, F–H and Fig. S4 B). We then specifically reintroduced Cdc5 activity in *ndt80Δ* cells by inducing either a wild-type (*CDC5*) or kinase-dead *CDC5* transgene (*CDC5^{KD}*, carrying the K110M mutation; Charles et al., 1998) from a copper-inducible promoter (*P_{CUP1}*). Strikingly, upon induction of *CDC5*, we observed a significant increase in nucleoplasmic localization of Nup60-GFP (Fig. 3, G and H). Induction of *CDC5^{KD}*, however, had no observable effect on Nup60 localization (Fig. 3, F and H). Thus, ectopic Polo kinase activity is sufficient to induce Nup60 detachment from the NPC.

**SWATH-MS proteomics identifies a Cdc5-dependent Nup60 phosphorylation site and additional novel Polo kinase targets**
Our data so far indicate that the Polo kinase Cdc5 regulates nuclear basket remodeling in meiosis I; however, the downstream targets critical for this regulation remained unknown. It is possible that Cdc5 drives the phosphorylation of one or more nucleoporins, or another factor, to trigger basket detachment. The nuclear basket is anchored to the NPC core via interactions with the Y-complex (Kim et al., 2018; Mészáros et al., 2015); as such, nucleoporins belonging to either of these two subcomplexes are promising candidates for Cdc5-dependent phosphorylation. To identify downstream Cdc5 targets, we employed an unbiased approach using Sequential Window Acquisition of all THeoretical Fragment Ion Mass Spectra (SWATH-MS; Gillet et al., 2012; Ludwig et al., 2018; Schubert et al., 2015) that allowed mapping of Cdc5-dependent phosphorylation sites across the proteome (Fig. 4 A). We induced either *CDC5* or *CDC5^{KD}* expression in *ndt80Δ* prophase I-arrested cells and collected samples over several time points after *CDC5* induction (Fig. 4 A). Approximately 7,500 phosphopeptides were identified in each of the samples analyzed, with the vast majority being detected in all samples (Fig. S4, C and D; and Table S6).

To determine which phosphopeptides accumulated in response to Polo kinase activity, we compared the peptide identifications made in either *CDC5*- or *CDC5^{KD}*-expressing cells and calculated the fold change in abundance (Cdc5/Cdc5^{KD}). We focused specifically on samples collected at 2–5 h following induction, which showed robust Cdc5 protein expression levels (Fig. S4 E). This analysis recapitulated the identification of several known targets of the Polo kinase Cdc5, including Slk19,

Spo13, Net1, and Sgs1 (Grigaitis et al., 2020; Matos et al., 2008; Park et al., 2008; Shou et al., 2002; Fig. 4 B, dark blue dots), validating our approach. Notably, it also led to the identification of various putative novel targets of Cdc5 (Fig. 4 B, dark gray dots).

The proteome-wide dataset identified 88 phosphorylation sites among 16 nucleoporins from various NPC subcomplexes (Fig. 4 C, and Table S6). All the basket nucleoporins displayed prominent phosphorylation (Table S6). However, only one of the sites, Nup60 S89, had increased phosphorylation in response to *CDC5* induction with high statistical confidence (P < 0.01; Fig. 4, C and D; and Table S6). Consistent with Nup60 being phosphorylated in a Polo kinase-dependent manner, we observed reduced mobility species of Nup60 during a wild-type meiosis at time points when Cdc5 was expressed (Fig. 4 E). Importantly, Polo kinase activity was both necessary and sufficient for Nup60 phosphorylation (Fig. 4, F and G; and Fig. S4, F and G), with Cdc5-dependent phosphorylation coinciding precisely with Nup60-Nup2 detachment from NPCs (compare Fig. S4, F and G with Fig. 3, C–H). As described in detail below, these findings provided an important foundation to further investigate how Cdc5 regulates nuclear basket remodeling in meiosis I.

We note that the SWATH-MS dataset offers a valuable tool to identify previously uncharacterized Polo kinase targets in meiosis (Table S6; see additional notes regarding data interpretation in Materials and methods). As a proof of concept, we confirmed that the putative target Swi6 and the known target Slk19 were modified in a Cdc5-dependent manner during meiosis (Fig. 4, B and D; and Fig. S4, H and I). We predict that the SWATH-MS dataset described here will serve as a broad resource for the study of various other processes regulated by the Polo kinase during meiosis.

**Identification of additional phosphorylation sites required for Cdc5-mediated Nup60 detachment**
To determine the extent to which the Cdc5-dependent phosphosite Serine 89 contributes to meiotic Nup60 phosphorylation, we generated strains expressing a phosphorylation-resistant mutant, *NUP60-S89A-9myc*, at the endogenous locus. The Nup60^{S89A} mutant protein displayed a reduced yet detectable mobility shift (Fig. 4 H), indicating that S89 is phosphorylated but additional residues may be concurrently modified during meiosis. To identify these additional sites, we took advantage of a second SWATH-MS dataset that characterizes the phosphoproteome during the entire budding yeast meiotic program (Wettstein et al., unpublished data). This new dataset has comprehensive coverage of the NPC with a total of 155 phosphopeptides and 106 individual phosphosites for nucleoporins, including 20 phosphosites in Nup60.

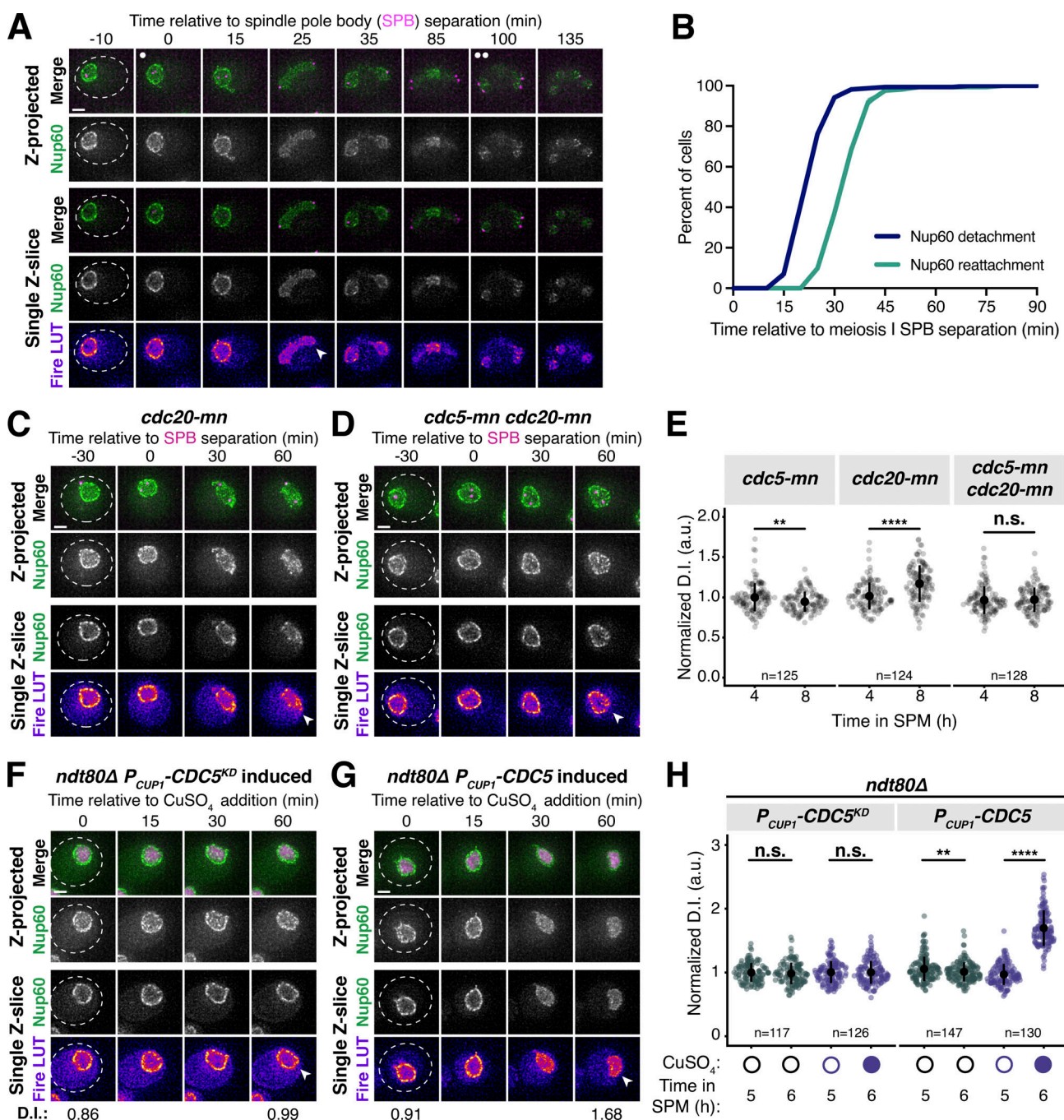

Figure 3. **CDC5 is necessary and sufficient for partial nuclear basket disassembly. (A)** Montage of a cell with Nup60-GFP, a nuclear basket nucleoporin, and Spc42-mCherry, a SPB component, progressing through meiosis (UB28201). **(B)** Quantification of Nup60-GFP detachment and reattachment timing relative to meiosis I SPB separation, corresponding to A. The mean ± range (shaded range) of two independent biological replicates is displayed (*n* = 91 cells for replicate 1, 81 cells for replicate 2). **(C and D)** Montages of cells with Nup60-GFP, a nuclear basket nucleoporin, and Spc42-mCherry, a spindle pole body component, entering metaphase I arrest in the following strains: (C) *cdc20-mn* (UB29253) or (D) *cdc5-mn cdc20-mn* (UB29249). Note that *cdc20-mn* nuclei become highly deformed during extended metaphase I arrest. **(E)** Quantification of Nup60-GFP detachment before (4 h in SPM) or during (8 h in SPM) metaphase I arrest in *cdc5-mn* (UB28492), *cdc20-mn* (UB28211), and *cdc5-mn cdc20-mn* (UB28614) cells. Htb1-mCherry, a histone, was used to define the nucleoplasm; due to slightly altered meiotic progression with Htb1-mCherry, a wild-type strain (UB14646) was used to assess sporulation progression and determine comparable timing to Spc42-mCherry containing strains. Individual DI values were normalized to the average DI for *cdc5-mn* cells (UB28492) at the premeiotic entry time point (4 h in SPM, Fig. S4 A). Asterisks indicate statistical significance calculated using a Wilcoxon signed-rank test when metaphase I arrest (8 h in SPM) values were compared with premeiotic entry (4 h in SPM) values for each genetic background (see Table S5 for P values). Sample sizes (*n*) are the number of cells quantified for each genetic background. For panels A–D, SPB separation was defined as the first time point that two distinct Spc42-mCherry puncta were visible. **(F and G)** Montages of cells with Nup60-GFP, a nuclear basket nucleoporin, and Htb1-mCherry, a histone, in prophase I arrest (*ndt80Δ*) with (F) *P_CUP1-CDC5^KD-3xFLAG-10xHis* induced (UB29069) or (G) *P_CUP1-CDC5-3xFLAG-10xHis* induced (UB29129). *CDC5* expression was induced at 5 h in SPM with 50 µM CuSO_4. **(H)** Quantification of Nup60 detachment for the experiment depicted in F and G and Fig. S4 B. Individual DI values were normalized to

In the new dataset, S89 phosphorylation peaked during the meiotic divisions (Fig. 5 A, red line), consistent with Cdc5 mediating its modification during a wild-type meiosis. Given the well-defined pattern of S89 phosphorylation, we reasoned that Cdc5-responsive phosphorylation sites would likely have a similar temporal profile to S89. This led to the identification of eight additional Nup60 phosphorylation sites exhibiting similar upregulation during meiosis (Fig. 5 A). These sites were located in an N-terminal cluster (T112, S118, S171, and S162) and a C-terminal cluster (S371, S374, S394, and S395). Many of the phosphosites (T112, S118, S171, S371, and S374) exhibited additional signatures of Cdc5-dependence, with low phosphorylation during prophase I arrest (ndt80Δ) and high phosphorylation during metaphase I arrest (cdc20-mn). Excitingly, the N-terminal phosphosite cluster overlapped with Nup60's HR, which mediates interaction with the NPC core (Fig. 5 B; Mészáros et al., 2015; Niño et al., 2016). Mapping of the phosphosites on the cryo-EM structure of the budding yeast NPC highlighted that both N- and C-terminal phosphosite clusters were well-positioned to regulate Nup60 binding to the Y-complex (Fig. 5 C; Kim et al., 2018).

To determine whether the identified phosphosites in Nup60 play a role in regulating its detachment from the NPCs, we constructed a series of NUP60 phosphorylation-resistant mutants tagged with GFP (Fig. 5 B): NUP60-S89A, NUP60-Nterm3A, and NUP60-Nterm5A, in which the N-terminal phosphosites were mutated to alanine; NUP60-Cterm4A, in which the C-terminal phosphosites were mutated to alanine; and NUP60-9A, in which both the N- and C-terminal phosphosites were mutated to alanine. Using these alleles, we monitored Nup60 localization and phosphorylation upon ectopic CDC5 induction during prophase I arrest (ndt80Δ background, Fig. 5, D–F). Consistent with our previous data, mutation of the S89 phosphosite alone (Nup60^S89A) slightly reduced Nup60 mobility as assayed by immunoblotting; however, this mutation did not abrogate CDC5-dependent Nup60 detachment (Fig. 5, D–F). Strikingly, mutating the additional N-terminal phosphosites (Nup60^Nterm3A and Nup60^Nterm5A) strongly reduced both Nup60 phosphorylation and detachment (Fig. 5, D–F). In contrast, mutating the C-terminal phosphosites alone did not impair Nup60 phosphorylation or detachment (Nup60^Cterm4A). Likewise, we did not observe any additional defects when the N- and C-terminal phosphomutations were combined (Nup60^9A; Fig. 5, D–F). Notably, all phosphorylation-resistant mutants, besides NUP60-Cterm4A, exhibited reduced phosphorylation in a cdc20-mn background, suggesting that the same sites may be responsible for regulating Nup60 detachment from NPCs during meiosis I (Fig. 5 G; and Fig. S5, A, C, E, and G). Overall, these data demonstrate that the Polo kinase Cdc5 regulates the phosphorylation of Nup60's HR, resulting in Nup60 detachment from NPCs.

## Cdc5-dependent Nup60 phosphorylation mediates meiosis I NPC remodeling

Since Nup60 and Nup2 detach during both meiosis I and meiosis II, it was unclear whether similar mechanisms underly both NPC remodeling events. Cdc5 is present during both meiotic divisions, but individual Cdc5 substrates can be targets of phosphorylation exclusively during meiosis I (Attner et al., 2013). To elucidate the role of Cdc5-driven Nup60 phosphorylation in NPC plasticity, we assessed the localization of the Nup60-GFP phosphorylation-resistant mutants by live-cell imaging throughout meiosis (Fig. 6). During meiosis I, we observed significantly reduced detachment for the same phosphomutants that exhibited reduced phosphorylation and detachment upon ectopic CDC5 expression (Fig. 5, and Fig. 6, A–C; and Fig. S5, B, D, F, and H; and Video 6). This confirms that Cdc5-dependent phosphorylation drives the meiosis I NPC remodeling event by disrupting Nup60 interaction with the NPC core (Fig. 6 D). During meiosis II, however, we found that Nup60 detachment and return to gamete nuclei occurred normally for all phosphorylation-resistant mutants tested (Fig. 6, A–C; Fig. S5, B, D, F, and H; and Video 6). Nup60 detachment is therefore regulated by different means during meiosis I and II. These data indicate that the two meiotic NPC remodeling events are mechanistically separable, further establishing them as distinct cellular phenomena.

## The lipid-binding N-terminus of Nup60 is required for NPC reassociation

Nup60 detachment during meiosis I is rapidly followed by its reassociation with the NPC (Fig. 1, C and D). While we had gained mechanistic insights into how Nup60's interaction with the NPC is disrupted by Cdc5-dependent phosphorylation, it remained unclear what structural features facilitate its reattachment to the NPC. Two N-terminal regions of Nup60 mediate its recruitment to the nuclear periphery: an AH, which binds directly to the inner nuclear membrane, and an HR, which interacts with the NPC core itself (Mészáros et al., 2015). Since Cdc5-dependent phosphorylation likely disrupts the binding of the HR to the Y-complex, we hypothesized that the AH may become crucial in mediating NPC reassembly after meiosis I. To test this hypothesis, we generated a mutant in which the N-terminal region of Nup60 spanning its AH was deleted (nup60-ΔAH [nup60-Δ2-47], Fig. 7 A; Mészáros et al., 2015). Prior to the meiotic divisions, Nup60^ΔAH-GFP largely localized to the nuclear periphery, exhibiting only minor nucleoplasmic mislocalization (–45 min, Fig. 7, B and C; and Video 7). During meiosis I, Nup60^ΔAH-GFP detached from the NPC core, similar to Nup60-GFP (–10 and 0 min, Fig. 7, B and C; and Video 7). However, instead of subsequently reattaching to NPCs, Nup60^ΔAH-GFP remained largely nucleoplasmic, with only a

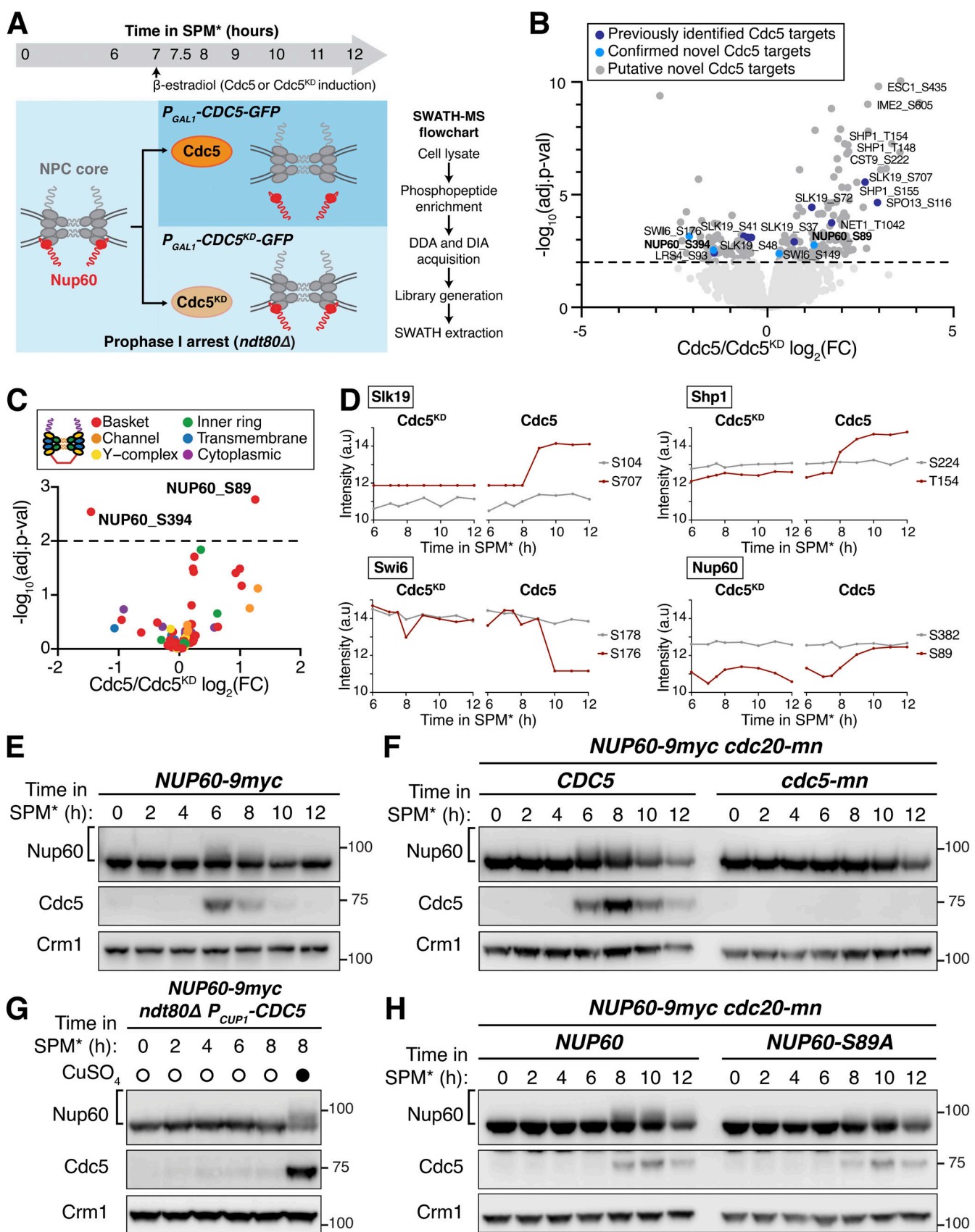

**Figure 4. SWATH-MS proteomics identifies Nup60 as a target of Cdc5-dependent phosphorylation. (A)** Schematic illustration of the experimental setup and flowthrough used for the proteomics screen. Strains that arrest in prophase I (*ndt80Δ*) carrying either $P_{GAL1}$-*CDC5-GFP* (YML3993) or $P_{GAL1}$-*CDC5^{KD}-GFP* (YML3994) were induced to enter meiosis by transfer to SPM* and, after 7 h in SPM*, treated with 2 μM β-estradiol to initiate Cdc5 or Cdc5^{KD} expression.

Samples for protein analyses, immunoblot, or SWATH-MS proteomics were collected at the indicated time points after transfer to SPM*. The experiment was performed in biological triplicates. **(B)** Volcano plot depicting the differential phosphorylation of peptides in cells ectopically expressing Cdc5 or Cdc5$^{KD}$, as described in A. For peptides of interest, the protein name and the phosphorylation site are indicated. The log$_2$ fold change (log$_2$(FC)) is plotted on the x-axis and the P value corrected by false discovery rate (–log$_{10}$(adj. P value)) is plotted on the y-axis. Phosphopeptides with adjusted P values >0.01 are represented by light gray dots below the dashed line. Phosphopeptides with adjusted P values ≤0.01 are represented by dark gray dots and are putative Cdc5 targets. Within this category, phosphopeptides marked in dark blue belong to previously reported Cdc5 targets, and phosphopeptides marked in light blue belong to novel Cdc5 targets further validated in this study. All data points are included in Table S6. **(C)** Volcano plot depicting the differential phosphorylation of nucleoporin peptides in cells ectopically expressing Cdc5 or Cdc5$^{KD}$. The data were plotted as in B, but with phosphopeptides colored according to the subcomplex that the nucleoporin belongs to. **(D)** Examples of phosphopeptides in Slk19, Shp1, Swi6, and Nup60 that either do not change (gray lines) or change (red lines) in abundance upon Cdc5 expression. Phosphopeptide abundance (the average of the measurements from three biological replicates) is plotted upon either *CDC5$^{KD}$* induction (left plot) or *CDC5* induction (right plot). In samples where a peptide could not be detected, data were imputed for that time point and used for plotting and statistical analysis (e.g., see the Slk19 S707 plot upon *CDC5$^{KD}$* induction). Note: In the case of Swi6, the peptide containing phosphorylated S176 is downregulated upon Cdc5 expression. Such downregulation may be a consequence of concurrent phosphorylation of a second/multiple residue(s) in the same peptide, not detected in this study. Therefore, the Cdc5 target site(s) would not be S176, but the additional site(s) in the same peptide. The same effect may be relevant for other disenriched peptides in B and C. **(E)** Immunoblots of Nup60-9myc and Cdc5 protein in a meiotic time course (YML6662). Samples were collected in 2-h intervals and cover the full meiotic cell division program. **(F)** Immunoblots for Nup60-9myc and Cdc5 protein from either *cdc20-mn* (YML6665) or *cdc20-mn cdc5-mn* (YML6664) cells during metaphase I arrest. **(G)** Immunoblots for Nup60-9myc and Cdc5 protein (YML12234) before (0–6 h in SPM*) or after (8 h in SPM*) treatment (either addition of copper or not) during prophase I arrest (*ndt80Δ*). Cdc5 was under control of the *CUP1* promoter (P$_{CUP1}$-CDC5). **(H)** Immunoblots for Nup60-9myc (YML6665) or Nup60$^{S89A}$-9myc (YML7956) in *cdc20-mn* strains induced to enter meiosis and arrest in metaphase I. For E–H, Crm1 was used as a loading control and the brackets to the left of the blots denote apparent phosphoshifts. For all immunoblots in this article, the values to the right of the blots indicate molecular weight in kilodaltons (kD), assessed using a ladder. Source data are available for this figure: SourceData F4.

---

small fraction returning to the nuclear periphery (+10 min, Fig. 7, B and C; and Video 7). Strikingly, the nucleoplasmic localization of Nup60$^{ΔAH}$-GFP persisted during both meiotic divisions and subsequent gamete formation (Fig. 7 C; and Video 7). A point mutation in the AH that renders it unable to bind the nuclear envelope (*nup60-I36R*; Mészáros et al., 2015) also exhibited impaired reassociation with the nuclear periphery (Fig. 7 D; and Video 8), confirming that the observed defect was due to the disruption of lipid binding. Nup60 protein levels were reduced in both *nup60-ΔAH* and *nup60-I36R* mutants during the meiotic divisions, suggesting that binding to the nuclear periphery may also affect Nup60 stability (Fig. S6, A–C). Notably, no reduced molecular weight band containing GFP was specific to either lipid-binding mutant, indicating that the differences in localization were not due to Nup60 degradation or truncation (source data, Fig. S6, A and B). Together, these data establish that Nup60 reassociation with the nuclear periphery, and therefore NPCs, requires its AH, suggesting that this domain may facilitate redocking to the nuclear envelope after meiotic detachment.

The meiotic role of Nup60's AH was specific, since removing a similar AH in Nup1 (*nup1-ΔAH*) had no effect on meiotic localization (Fig. S6, D and E). We hypothesized that the observed specificity stemmed from the AH of Nup60 becoming essential for its interaction with NPCs downstream of Cdc5-dependent detachment. To directly assess this prediction, we monitored Nup60$^{ΔAH}$-GFP dynamics during metaphase I arrest with or without Cdc5 activity (Fig. 7 E). Nup60$^{ΔAH}$-GFP exhibited significant detachment from the nuclear periphery and increased phosphorylation in *cdc20-mn* cells, but not in *cdc5-mn cdc20-mn* cells (Fig. 7, E–G). *cdc5-mn* single mutants were indistinguishable from *cdc5-mn cdc20-mn* double mutants (Fig. 7, E–G). Ectopic *CDC5* expression was also sufficient to trigger the complete detachment of Nup60$^{ΔAH}$-GFP from the nuclear periphery during prophase I (Fig. S6, F–H). Cdc5-mediated disruption of the interaction between the HR and the NPC core, therefore, results in the AH becoming necessary for Nup60 association with the NPC. Consistent with the AH and HR

acting in parallel to coordinate proper Nup60 localization, the extent of detachment in *nup60-ΔAH cdc20-mn* cells was more profound than what we observed for *NUP60 cdc20-mn* cells (compare Fig. 7, E–G with Fig. 3, C–E). Our data reveal that the N-terminus of Nup60 facilitates both detachment and reattachment with the NPC core during meiosis I: Polo kinase-dependent phosphorylation of the HR results in Nup60 detachment and redocking by the AH results in Nup60 reattachment (Fig. 7 H).

### Nup60 reattachment is required for proper gamete nuclear basket organization

Nup60 is a key subunit of the nuclear basket, responsible for recruiting both Nup2 and Mlp1 to NPCs (Dilworth et al., 2001; Feuerbach et al., 2002). We, therefore, hypothesized that reassociation of Nup60 with the NPCs might be necessary for proper nuclear basket organization in gamete nuclei. Accordingly, all the nuclear basket subunits that we tested were mislocalized in *nup60-ΔAH* gametes (Fig. 8 and Fig. S7). Nup2-GFP phenocopied Nup60$^{ΔAH}$-GFP, remaining nucleoplasmic instead of returning to the gamete nuclear periphery (Fig. S7, A and B). Nup1-GFP was barely detectable at the gamete nuclear periphery after meiosis II (Fig. 8, A and B). Mlp1-GFP exhibited nucleoplasmic mislocalization and often formed puncta akin to those observed in *nup60Δ* cells, structural abnormalities that persisted for hours after the meiotic divisions (Feuerbach et al., 2002; Fig. 8, D–F; and Video 9). Mlp1 misorganization was similar in *nup60-I36R* gametes (Fig. S7, C–E), confirming that the basket defects observed were due to defective Nup60 reassociation with the nuclear periphery. In contrast, core nucleoporins—including the inner ring nucleoporin Nup170-GFP and transmembrane nucleoporin Pom34-GFP—were sequestered to the GUNC and appeared to be normally distributed around gamete nuclei in *nup60-ΔAH* cells, suggesting that NPC organizational defects were specific to the nuclear basket (Fig. S7, F–I). Proper execution of meiotic NPC remodeling is therefore required for nuclear basket organization in gametes.

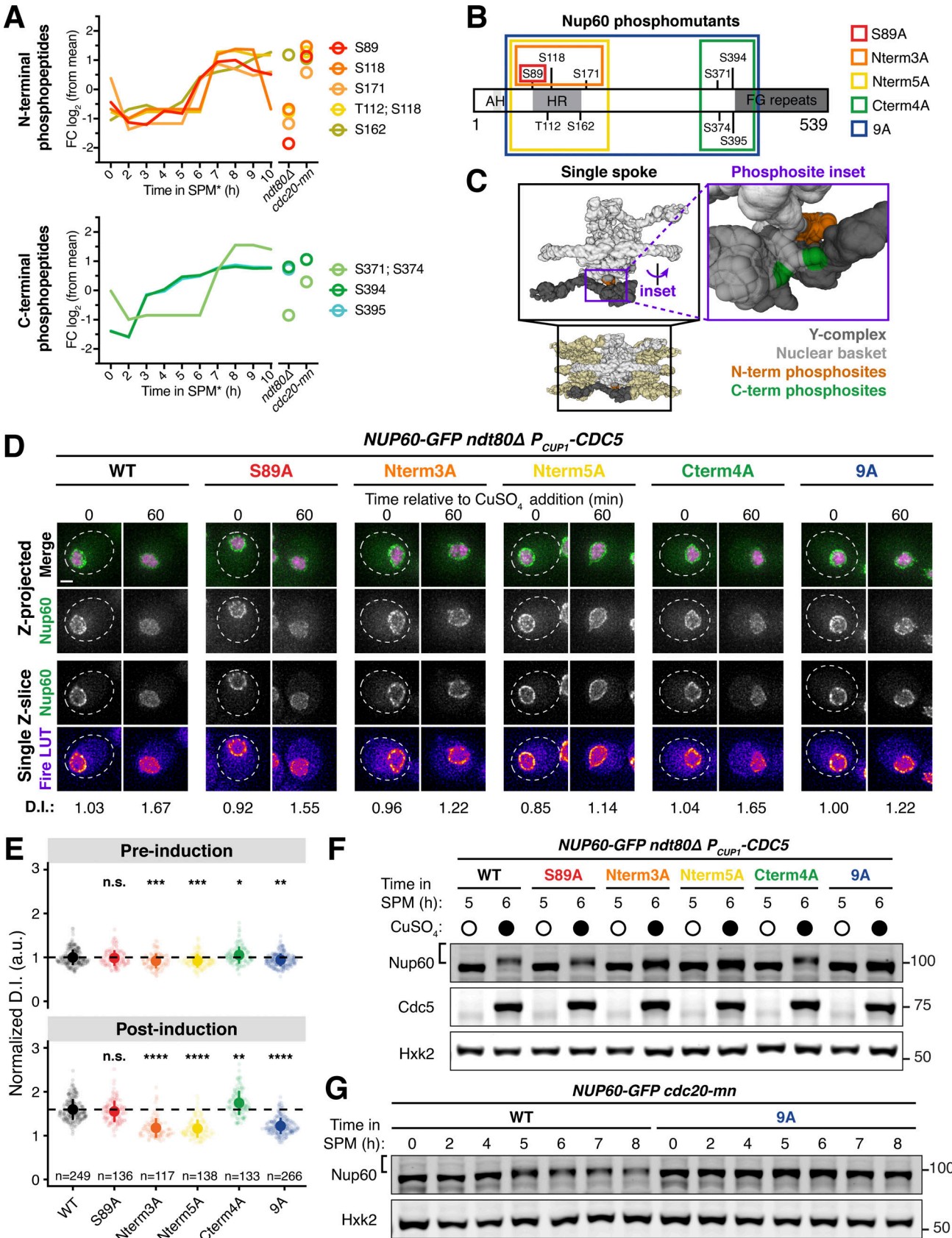

Figure 5. **Identification of Nup60 phosphosites at the interface with the NPC core that mediate Cdc5-dependent detachment. (A)** N-terminal and C-terminal Nup60 phosphopeptides that exhibit meiotic upregulation (data from Wettstein et al., *in preparation*). Each line represents an individual phosphopeptide originating from Nup60 protein, measured in 1-h intervals across the entire meiotic cell division program, as well as in prophase I arrested cells

(*ndt80Δ*, 8 h in SPM*) and metaphase I arrested cells (*cdc20-mn*, 10 h in SPM*). Log$_2$ fold change (log$_2$(FC)) relative to mean expression over all samples of each phosphopeptide is plotted on the y-axis. The average measurement of triplicates is plotted. **(B)** Schematic of Nup60 depicting the position of phosphosites relative to known structural features. The phosphomutants generated are indicated by colored boxes. AH = amphipathic helix, HR = helical region. **(C)** Visualization of the N-terminal (orange) and C-terminal phosphosites (green) of Nup60 on a cryo-EM structure of the NPC (Kim et al., 2018) visualized using Mol* (Sehnal et al., 2021). **(D)** Montages of cells containing different *NUP60-GFP* alleles and *HTB1-mCherry* in a prophase I arrest (*ndt80Δ*) before (5 h in SPM) and after (6 h in SPM) induction of P$_{CUP1}$-*CDC5-3xFLAG-10xHis*. The following alleles were tested: *NUP60-GFP* (UB29129), *NUP60-S89A-GFP* (UB29560), *NUP60-Nterm3A-GFP* (UB29636), *NUP60-Nterm5A-GFP* (UB29638), *NUP60-Cterm4A-GFP* (UB29562), and *NUP60-9A-GFP* (UB29564). Induction was performed at 5 h in SPM medium with 50 μM CuSO$_4$. **(E)** Quantification of Nup60 detachment for the experiment depicted in D. Individual DI values were normalized to the average DI for Nup60-GFP cells at the pre-induction time point (5 h in SPM). Asterisks indicate statistical significance calculated using Dunn's test for multiple comparisons when each allele was compared with wild type for a given time point (see Table S5 for P values). Sample sizes (*n*) are the number of cells quantified for each strain; for Nup60-GFP and Nup60$^{9A}$-GFP, cells from two independent replicates were pooled. **(F)** Immunoblot for different Nup60-GFP alleles and Cdc5-3xFLAG-10xHis before (5 h in SPM) or after (6 h in SPM) copper induction during prophase I arrest (*ndt80Δ*), corresponding to the images in D. Hxk2 was used as a loading control. **(G)** Immunoblot for Nup60-GFP (UB29253) or Nup60$^{9A}$-GFP (UB30438) in *cdc20-mn* background. Hxk2 was used as a loading control. For F and G, the brackets to the left of the blots denote apparent phosphoshifts. For each montage, normalized DI values are indicated when calculated. Scale bars, 2 μm. Source data are available for this figure: SourceData F5.

## Meiotic dynamics reveal underlying principles of nuclear basket organization

Intriguingly, the meiotic phenotypes displayed by basket nucleoporins in *nup60-ΔAH* cells were suggestive of previously unknown organizational principles for the nuclear basket. Nup1, for example, appears to partially rely on Nup60 for its localization. Even before the meiotic divisions, Nup1 localization to the nuclear periphery was reduced in *nup60-ΔAH* mutants (Fig. 8 C). This dependence on Nup60 might also explain the partial detachment of Nup1 from the nuclear periphery during meiosis I (Fig. 1 C). Most strikingly, Mlp1 remained peripheral during meiosis I after Nup60$^{ΔAH}$ became entirely nucleoplasmic, despite Nup60 being required for Mlp1 recruitment to the NPC (Fig. 8, E and F). Mlp1 only began to exhibit mislocalization after it detached from the NPC core during meiosis II (Fig. 8, E and F). This led us to hypothesize that Nup60 has two separate roles in organizing nuclear basket nucleoporins: (1) a role in the recruitment of basket nucleoporins to the NPCs, critical for both Nup2 and Mlp1, and (2) a role in the retention of basket nucleoporins at the NPCs, critical for Nup2 but not Mlp1 (Fig. 9 A). This model could also explain how Mlp1, but not Nup2, remains associated with NPCs when Nup60 detaches during meiosis I (Fig. 1 C and Fig. 2 G).

To directly test this model, we monitored Mlp1-GFP and Nup2-GFP localization upon different Nup60 conditional depletion regimens. We generated strains with an auxin-inducible degron fused to Nup60 (Nup60-3V5-IAA17) as well as a copper-inducible F-box receptor (P$_{CUP1}$-*osTIR1*$^{F74G}$), allowing us to achieve rapid depletion of Nup60 in both mitotic and meiotic cells (Fig. 9 B; Yesbolatova et al., 2020). If Nup60 is required for nuclear basket recruitment to the NPCs, then constitutive depletion in mitotic cells should result in the mislocalization of newly synthesized Mlp1 and Nup2 (Fig. 9 A). Indeed, both Mlp1 and Nup2 exhibited mislocalization upon constitutive mitotic depletion, similar to that observed in *nup60Δ* cells (Fig. 9, C, D, F, and G). If Nup60 is required for the retention of basket nucleoporins, then transient depletion should also result in rapid mislocalization, since Mlp1 and Nup2 would no longer be anchored to the NPCs. To ensure that the cells were matched for the meiotic stage, we induced transient depletion during prophase I arrest (*ndt80Δ* background). We found that while Nup2 rapidly became nucleoplasmic upon Nup60 depletion, Mlp1 was

retained at the nuclear periphery for hours (Fig. 9, E and H). Mlp1 can therefore remain associated with the NPC core independently of Nup60 after its initial recruitment. Moreover, the retention of Mlp1 upon Nup60 depletion suggests that the association of Mlp1 with the NPC core is very stable as it remained perinuclear for hours without any Nup60-mediated recruitment (Fig. 9 H). The distinct organization of Mlp1 within the nuclear basket explains how the two modes of NPC remodeling observed during budding yeast meiosis are possible.

## Meiotic NPC remodeling is conserved in *S. pombe*

Our work established that the nuclear basket displays extensive remodeling during budding yeast meiosis (Fig. 2 G); the extent to which basket remodeling is conserved remains unknown. To address this question, we monitored nuclear basket dynamics in the fission yeast *S. pombe* (Sp), which is highly diverged from the budding yeast *S. cerevisiae* (Sc; Sipiczki, 2000). *S. pombe* cells do not form a nuclear envelope compartment akin to the GUNC, and their gametes inherit the NPC core during meiosis (Asakawa et al., 2010). Excitingly, SpNup60 (the ortholog of ScNup60) and SpNup61 (the ortholog of ScNup2) both detached from the nuclear periphery during meiosis I with similar timing to that observed in *S. cerevisiae* (Fig. 10, A–C; and Video 10). On the other hand, SpNup124 (the ortholog of ScNup1), SpNup211 (the ortholog of ScMlp1 and ScMlp2), and SpAlm1 (a putative Tpr-like nucleoporin without an *S. cerevisiae* ortholog) all remained associated with the NPC throughout meiosis I (Fig. 10, D and E; and Fig. S8 A). The transmembrane nucleoporin SpPom34 (the ortholog of ScPom34) also remained peripheral, consistent with the core remaining intact during partial basket detachment (Fig. 10 F). Together, these data suggest that the meiosis I NPC remodeling event is conserved and occurs independently of NPC sequestration to the GUNC.

During meiosis II, SpNup60 and SpNup61 detached from the nuclear periphery and SpPom34 remained associated with the NPC, as observed in meiosis I (Fig. S8, B, C, and G). However, unlike in *S. cerevisiae*, SpNup124, SpAlm1, and SpNup211 did not detach (Fig. S8, D–F). The meiosis II NPC remodeling event is therefore less well conserved and its relationship to GUNC formation remains unclear. Notably, both SpNup60 and SpNup61 also detached and reattached during mitotic anaphase, suggesting that partial nuclear basket detachment is a general

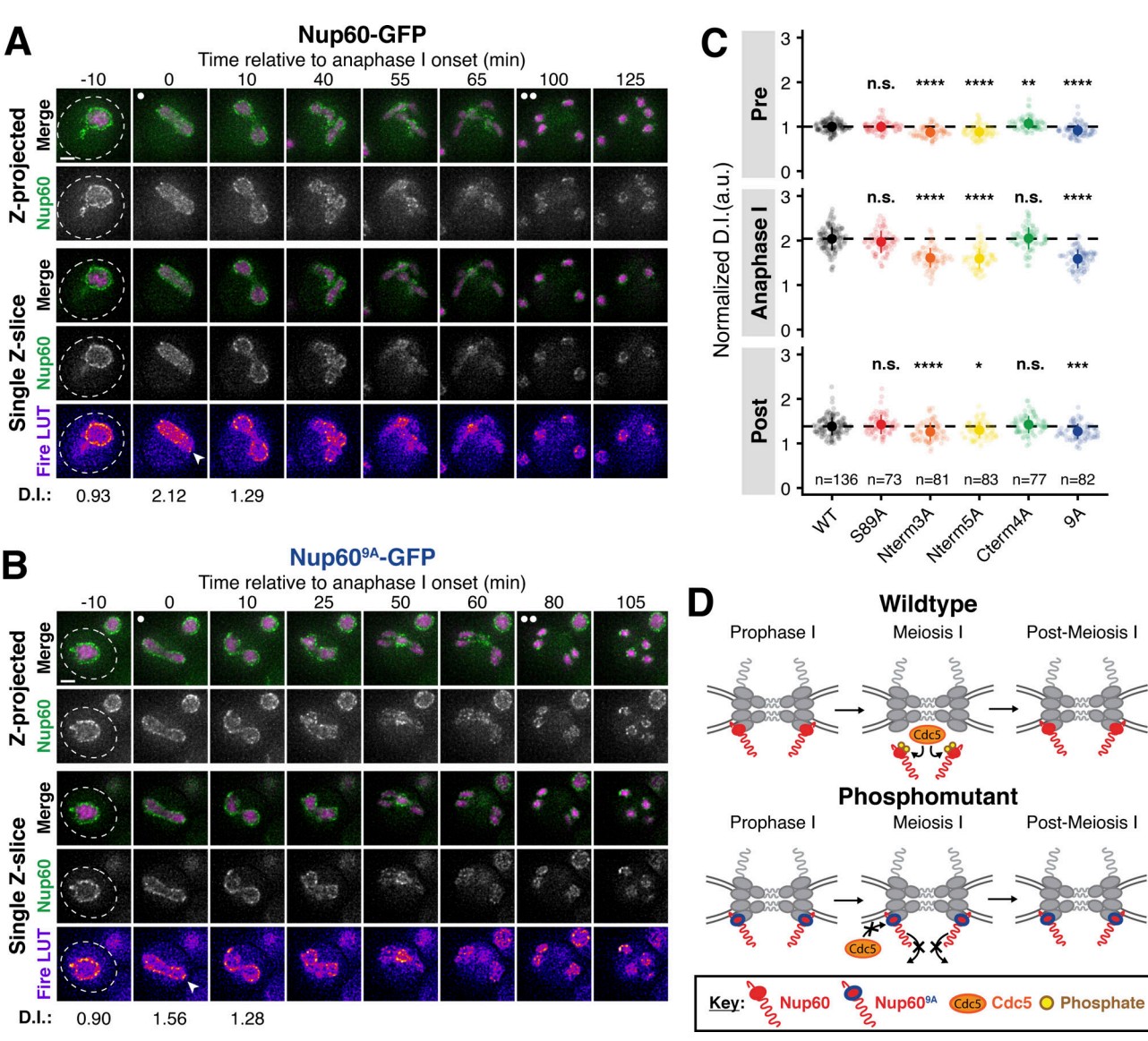

Figure 6. **Cdc5-dependent Nup60 phosphorylation mediates meiosis I, but not meiosis II, NPC remodeling. (A and B)** Montages of cells with (A) Nup60-GFP (UB14646) or (B) Nup60$^{9A}$-GFP (UB29358) and Htb1-mCherry progressing through meiosis. **(C)** Quantification of Nup60 detachment before (−10 min, "Pre"), coincident with (0 min, "Anaphase I"), and after (+10 min, "Post") the onset of anaphase I. Individual DI values were normalized to the average DI of Nup60-GFP at the "Pre" time point. Asterisks indicate statistical significance calculated using Dunn's test for multiple comparisons when each allele was compared to Nup60-GFP for a given time point (see Table S5 for P values). The dashed lines indicate the average DI for Nup60-GFP for each time point. Sample sizes (n) are the number of cells quantified for each allele; for Nup60-GFP, cells from two independent replicates were pooled. **(D)** A schematic that depicts Cdc5-dependent phosphorylation of Nup60 driving detachment from the NPC during meiosis I in wild type, but not phosphomutant, cells. For all panels, the onset of anaphase I was defined as the Htb1-mCherry chromatin mass exhibiting distortion from a spherical shape consistent with chromosome segregation. For each montage, normalized DI values are indicated when calculated. Scale bars, 2 μm.

feature of cell divisions in *S. pombe* (Fig. S9, A–F). Since the same nuclear basket nucleoporins detach from the NPC with similar cell cycle timing in both *S. pombe* and *S. cerevisiae*, the organizational principles of the nuclear basket uncovered in this study seem to be conserved across 400 million years of evolution.

## Discussion

In this study, we provide key mechanistic insights into two distinct NPC remodeling events in budding yeast meiosis (Fig. 2 G). During meiosis I, two nuclear basket nucleoporins—Nup60 and Nup2—transiently dissociate from NPCs. The Polo kinase Cdc5 drives the detachment of Nup60 and Nup2 from NPCs via developmentally programmed phosphorylation at the interface between Nup60 and the Y-complex. Nup60 and Nup2 subsequently reassociate with NPCs in a manner dependent on the lipid-binding AH of Nup60. During meiosis II, the entire nuclear basket detaches from the NPC core and returns to gamete nuclei, avoiding sequestration to the GUNC. Nuclear basket organization is therefore subject to extensive plasticity during gametogenesis, with potential implications for gamete health.

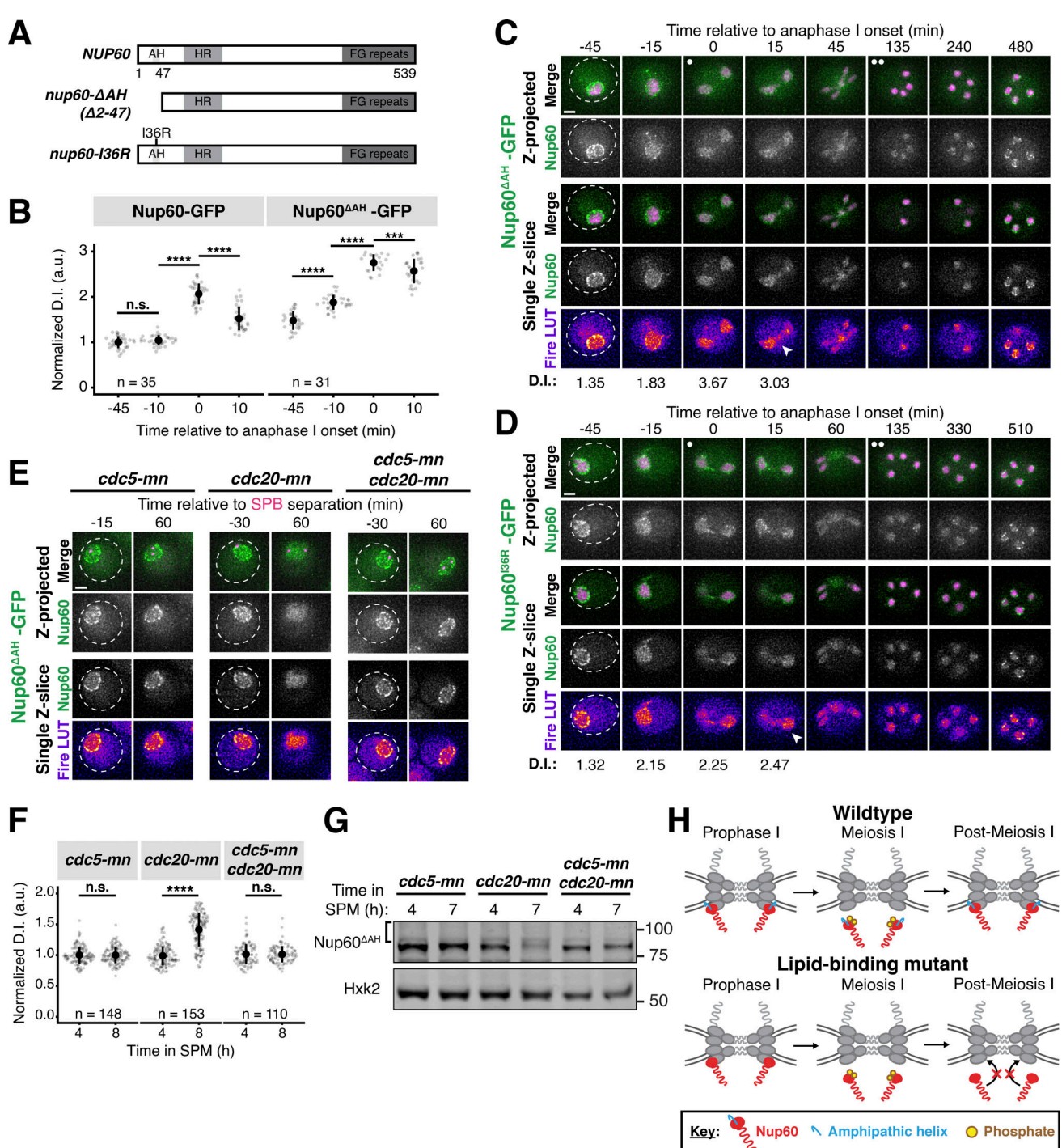

Figure 7. **The lipid-binding N-terminus of Nup60 mediates return to the nuclear periphery after meiotic detachment. (A)** Schematic of different *NUP60* lipid-binding mutants generated: *nup60-ΔAH*, which lacks its N-terminal AH, and *nup60-I36R*, containing a point mutation in its N-terminal AH. **(B)** Quantification of Nup60 detachment for Nup60-GFP (UB14646) and Nup60^ΔAH-GFP (UB25731) at −45, −10, 0, and +10 min relative to the onset of anaphase I. Individual DI values were normalized to the average DI of Nup60-GFP at the −45 min time point. Asterisks indicate statistical significance calculated using Wilcoxon signed-rank tests when each time point was compared to the previous time point for a given allele (see Table S5 for P values). Sample sizes (*n*) are the number of cells quantified for each allele. **(C and D)** Montages of cells with (C) Nup60^ΔAH-GFP (UB25731) or (D) Nup60^I36R-GFP (UB27189) and Htb1-mCherry progressing through meiosis. For C and D, the white arrowheads in the "Fire LUT" images denote nuclei after anaphase I that continue to exhibit Nup60 detachment. **(E)** Montages of cells with Nup60^ΔAH-GFP and Spc42-mCherry entering metaphase I arrest in the following strains: *cdc5-mn* (UB29257), *cdc20-mn* (UB29259), or *cdc5-mn cdc20-mn* (UB29255). **(F)** Quantification of Nup60^ΔAH-GFP detachment before (4 h in SPM) or during (8 h in SPM) metaphase I arrest in *cdc5-mn* (UB28494), *cdc20-mn* (UB28213), and *cdc5-mn cdc20-mn* (UB28616) cells. Htb1-mCherry, a histone, was used to define the nucleoplasm; due to slightly altered meiotic progression with Htb1-mCherry, a wild-type strain (UB25731) was used to assess sporulation progression and determine comparable timing to Spc42-mCherry containing strains. Asterisks indicate significance determined using Wilcoxon signed-rank tests when metaphase I arrest (8 h in SPM) values were compared with premeiotic entry (4 h in SPM) values for each genetic background (see Table S5 for P values). **(G)** Immunoblot for Nup60^ΔAH-GFP before

(4 h in SPM) or during (7 h in SPM) metaphase I arrest for the strains in E. Hxk2 was used as a loading control. The bracket to the left of the blot denotes the apparent phoshoshift. **(H)** A schematic that depicts the role of Nup60's AH in mediating return to the NPC after meiosis I detachment. For all panels, the onset of anaphase I was defined as the Htb1-mCherry chromatin mass exhibiting distortion from a spherical shape consistent with chromosome segregation. For each montage, normalized DI values are indicated when calculated. Scale bars, 2 μm. Source data are available for this figure: SourceData F7.

## A new type of NPC remodeling during cell division: Nuclear basket detachment

We characterize a new form of NPC remodeling during cell division: specific disassembly and reassembly of part or all of the nuclear basket (Fig. 1). Disassembly of the nuclear basket during other previously characterized NPC remodeling events is coupled to the disassembly of channel nucleoporins, driving the loss of nucleocytoplasmic compartmentalization (De Souza et al., 2004; Dey et al., 2020; Dultz et al., 2008; Osmani et al., 2006). In contrast, both core and channel nucleoporins remain peripheral during budding and fission yeast meiosis (Figs. 1 and 10), likely resulting in largely intact NPCs without some or all nuclear basket nucleoporins. Despite their distinct outcomes, meiosis I basket detachment and previously characterized NPC remodeling share a similar mechanistic underpinning:

phosphorylation driven by a cell-cycle kinase disrupts nucleoporin interactions to facilitate disassembly (Figs. 3, 4, 5, and 6). Moreover, Polo kinases are involved in both fungal and metazoan NPC remodeling—Cdc5 in budding yeast, PLK-1 in worms, and PLK1 in human cells—indicating that the same regulatory machinery can be used to achieve different NPC structural states (Linder et al., 2017; Martino et al., 2017). Future work should assess whether the Polo kinase Cdc5 directly phosphorylates Nup60, establishing further similarity to the phosphorylation events observed in worms and humans, or whether it acts in conjunction with other kinases.

The nuclear basket is the most dynamic and heterogenous subcomplex of the NPC, in addition to being the least structurally and functionally understood. Individual basket members

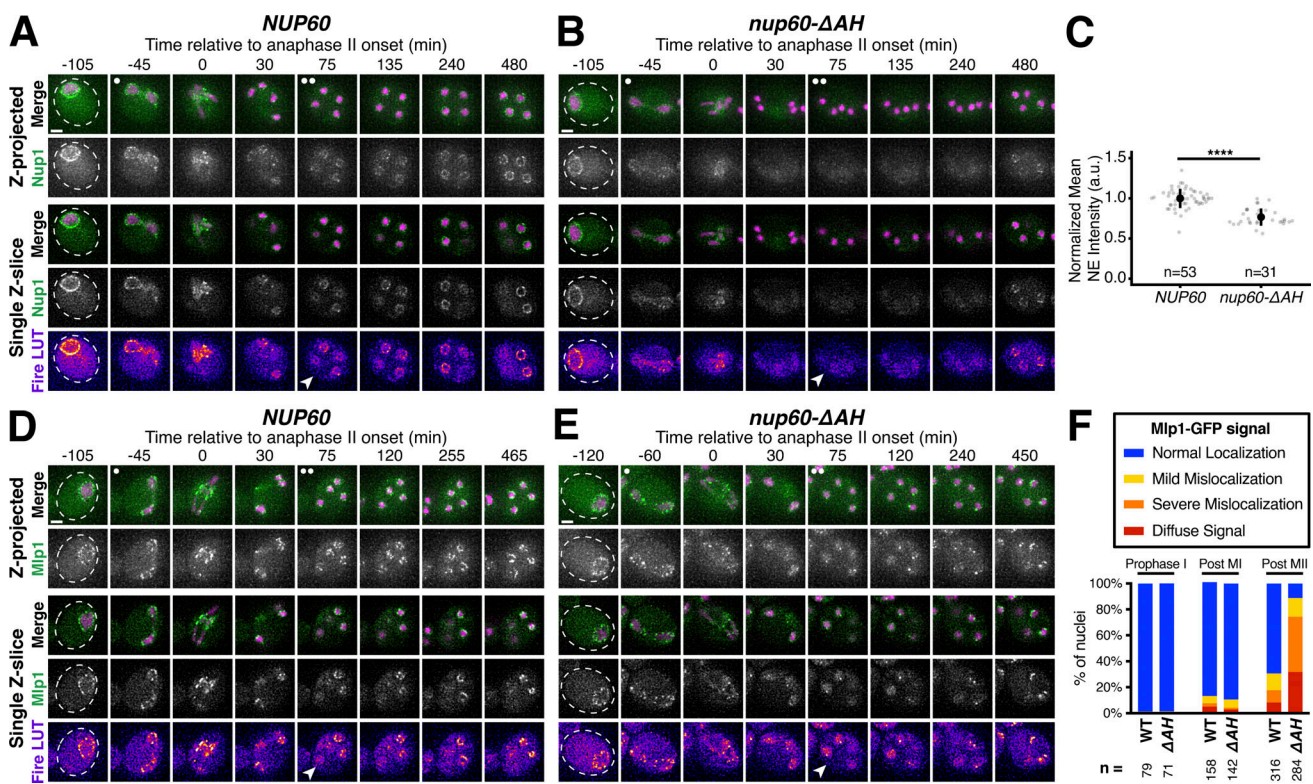

Figure 8. **Peripheral Nup60 is required for proper gamete nuclear basket organization. (A and B)** Montages of cells with Nup1-GFP, a nuclear basket nucleoporin, and Htb1-mCherry, a histone, progressing through meiosis in either (A) a *NUP60* (UB15303) or (B) a *nup60-ΔAH* (UB30628) genetic background. **(C)** Quantification of average nuclear envelope intensity of Nup1-GFP during prophase I (defined as 1 h prior to the anaphase I, the presence of two clear Htb1-mCherry lobes) in *NUP60* and *nup60-ΔAH* cells, as shown in A and B. Individual intensity values were normalized to the average nuclear envelope intensity in *NUP60* cells. Asterisks indicate significance determined using a Wilcoxon rank sum test (see Table S5 for P values). Sample sizes (*n*) are the number of cells quantified in each background. **(D and E)** Montages of cells with Mlp1-GFP, a nuclear basket nucleoporin, and Htb1-mCherry, a histone, progressing through meiosis in either (D) a *NUP60* (UB14648) or (E) a *nup60-ΔAH* (UB30632) genetic background. **(F)** Quantification of Mlp1-GFP organization for the strains in D and E during prophase I (defined as an hour before the post-MI time point), post-MI (defined as the presence of two clear Htb1-mCherry lobes), and post-MII (defined as 2 h after the presence of four clear Htb1-mCherry lobes). Sample sizes (*n*) indicate the number of nuclei scored for Mlp1 organization. For panels A, B, D, and E, the white arrowheads in the "Fire LUT" images denote cells after anaphase II when gamete nuclear basket organization or misorganization is apparent. For all panels, the onset of anaphase II was defined by the presence of four Htb1-mCherry lobes. Scale bars, 2 μm.

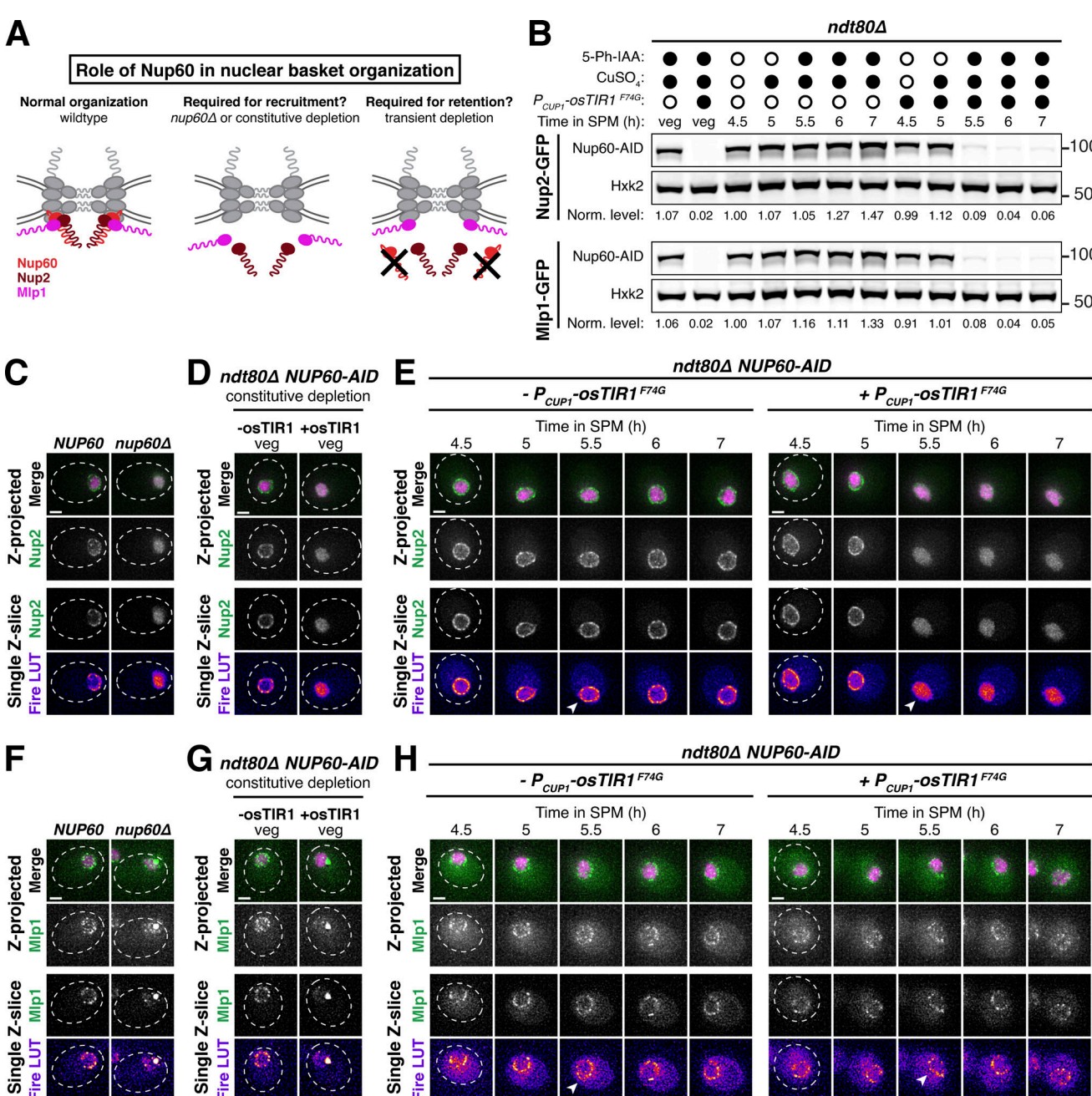

Figure 9. **Nup60 is not required for the retention of Mlp1 at the nuclear periphery, explaining differences in the meiotic dynamics of basket nucleoporins. (A)** Schematic depicting a hypothesis for two distinct roles of Nup60 in nuclear basket organization: (1) a role in recruitment (for both Mlp1 and Nup2), tested by monitoring nucleoporin localization in a *nup60Δ* background or upon constitutive Nup60 depletion using the *NUP60-AID* allele and (2) a role in retention (for Nup2, but not Mlp1), tested by monitoring nucleoporin localization upon transient Nup60 depletion using the *NUP60-AID* allele. **(B)** Immunoblot of Nup60-3V5-IAA17 (Nup60-AID) levels in strains with Nup2-GFP or Mlp1-GFP, in the absence or presence of $P_{CUP1}$-*osTIR1-F74G*. (Nup2-GFP: −osTIR1 = UB32240 and +osTIR1 = UB32238; Mlp1-GFP: −osTIR1 = UB32248 and +osTIR1 = UB32246). Hxk2 was used as a loading control. Vegetative depletion was induced with 50 µM $CuSO_4$ and 100 µM 5-Ph-IAA immediately upon inoculation of a logarithmic YPD culture at 0.25 OD/ml and continued for 10 h. For meiotic depletion, cells were maintained in prophase I arrest (*ndt80Δ*), and depletion was induced by adding 50 µM $CuSO_4$ at 4.5 h in SPM and 100 µM 5-Ph-IAA at 5 h in SPM. Background subtracted Nup60 protein levels were divided by Hxk2 levels to control for loading and then normalized to the -osTIR1 strain at 4.5 h in SPM. **(C)** Nup2-GFP localization in *NUP60* (UB15305) and *nup60Δ* (UB31600) cells from a saturated YPD culture. **(D)** Nup2-GFP localization in -osTIR1 (UB32240) or +osTIR1 (UB32238) cells from a saturated YPD culture after constitutive Nup60 depletion. **(E)** Nup2-GFP localization in −osTIR1 (UB32240) or +osTIR1 (UB32238) cells before and after Nup60 depletion during prophase I arrest. **(F)** Mlp1-GFP localization in *NUP60* (UB14648) and *nup60Δ* (UB31262) cells from a saturated YPD culture. **(G)** Mlp1-GFP localization in −osTIR1 (UB32248) or +osTIR1 (UB32246) cells from a saturated YPD culture after constitutive Nup60 depletion. **(H)** Mlp1-GFP localization in −osTIR1 (UB32248) or +osTIR1 (UB32246) cells before and after Nup60 depletion during prophase I arrest. For E and H, the white arrowheads in the "Fire LUT" images denote nuclei at the time point when Nup60-AID was largely depleted in +osTIR1 strains. Scale bars, 2 µm. Source data are available for this figure: SourceData F9.

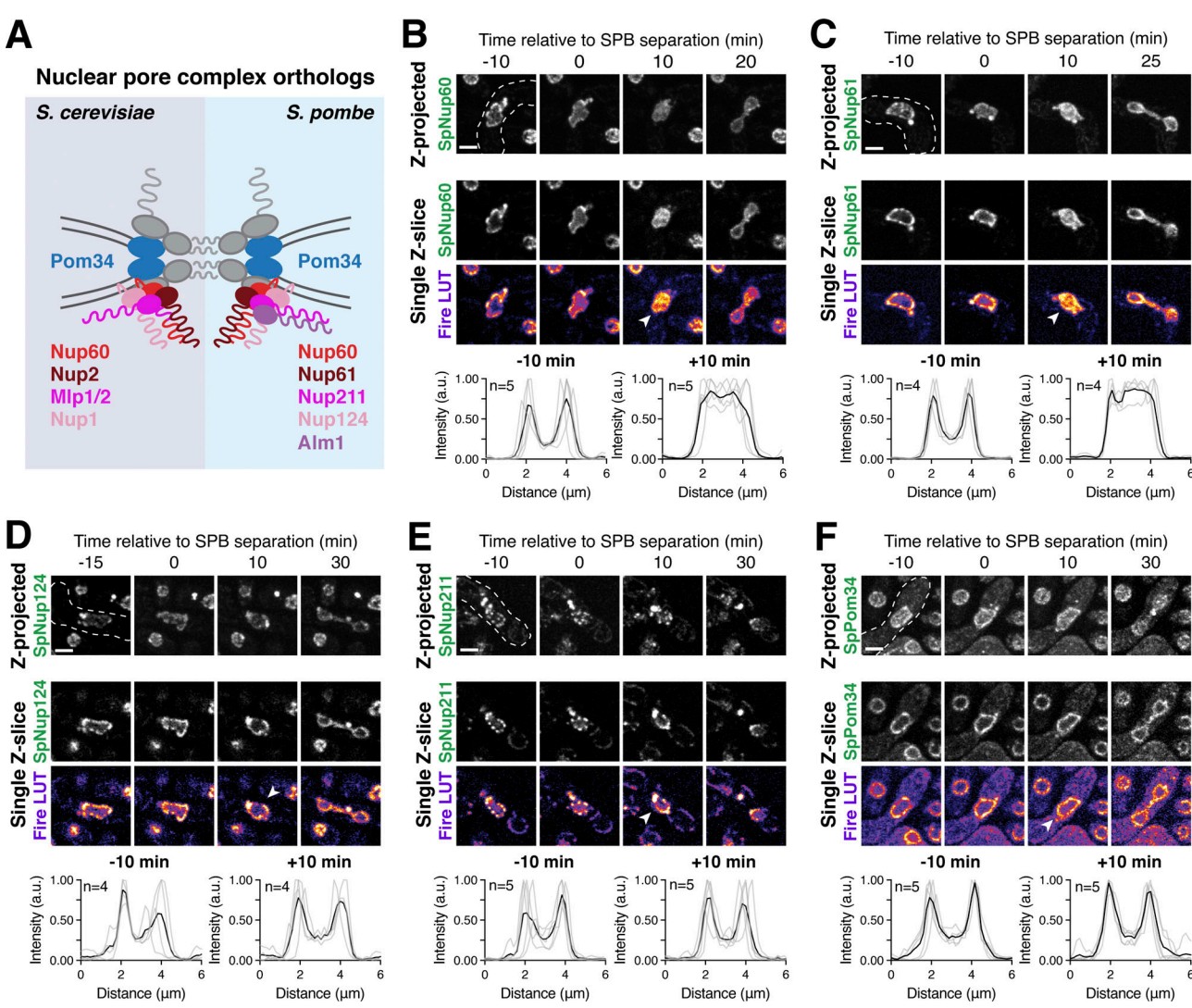

Figure 10. **Meiosis I NPC remodeling is conserved in *S. pombe*. (A)** A schematic depicting nuclear basket nucleoporin orthologs in *S. cerevisiae* and *S. pombe*. **(B–F)** Maximum intensity projections (top row) and single z-slice image montages of cells with various GFP-tagged nucleoporins progressing through meiosis I and staged according to the timing of meiosis I SPB separation as visualized using SpPpc89-mCherry (not shown). For each nucleoporin, individual (gray) and mean (black) Nup-GFP intensity profiles measured on single z-slices are shown below for the indicated number of individual nuclei (n), 10 min before and after SPB separation. The following nucleoporins were visualized: (B) SpNup60-GFP, the ortholog of ScNup60 (fySLJ730 × fySLJ479); (C) SpNup61-GFP, the ortholog of ScNup2 (fySLJ840 × fySLJ479); (D) SpNup124-GFP, the ortholog of ScNup1 (fySLJ456 × fySLJ989); (E) SpNup211-GFP, the ortholog of ScMlp1 and ScMlp2 (fySLJ456 × fySLJ990); and (F) SpPom34-GFP, the ortholog of ScPom34 (fySLJ1242 × fySLJ1243). For B–F, the white arrowheads in the "Fire LUT" images denote nuclei after meiosis I SPB separation, at the stage when SpNup60-SpNup61 detachment is observed. Scale bars, 3 μm.

disassociate and reassociate with the NPC with rapid kinetics in yeast and metazoan cells (Buchwalter et al., 2014; Hakhverdyan et al., 2021; Rabut et al., 2004). Individual NPCs within the same yeast cell exhibit variable basket composition, with "basketless" NPCs near the nucleolus lacking Mlp1 and Mlp2 (Akey et al., 2022; Galy et al., 2004; Varberg et al., 2022). Budding yeast meiosis represents a developmental context in which NPCs change their nuclear basket composition en masse (Fig. 1). Moreover, distinct nuclear basket states are achieved during meiosis I and meiosis II due to partial or full basket disassembly (Fig. 2). Studying nuclear transport during the meiotic divisions or upon ectopic modification of nuclear basket organization using meiotic mechanisms may help illuminate how nuclear basket plasticity affects NPC function.

## Insights into nuclear basket organization from meiotic remodeling

Our study provides key insights into how the structural principles of the nuclear basket can be exploited to achieve disassembly and reassembly (Figs. 3, 4, 5, 6, and 7). During meiosis I, we find that the N-terminus of Nup60 facilitates both detachment and reattachment: phosphorylation of its HR disrupts interaction with the NPC core, and lipid-binding by its AH allows for redocking on the NPCs. These principles may be similarly utilized in other cellular contexts. For example, mono-ubiquitylation of the HR of Nup60 has also been implicated in altering its association with the NPC (Niño et al., 2016). Excitingly, meiosis represents a developmental context where the direct membrane-binding function of the nuclear basket

becomes essential for proper NPC organization (Figs. 7 and 8). In addition to facilitating the reassociation of Nup60 and Nup2 with NPCs following meiosis I, the Nup60 AH is necessary for the timely recruitment of all nuclear basket nucleoporins to the gamete nuclear periphery after meiosis II (Fig. 8).

Moreover, the distinct behaviors of basket nucleoporins during partial nuclear basket disassembly drove insights into how Nup60 coordinates nuclear basket organization. Nup60 is involved in the recruitment of Mlp1 and Nup2 to the NPCs (Cibulka et al., 2022; Dilworth et al., 2001; Galy et al., 2004); however, Mlp1 remained associated with the nuclear periphery during meiosis I despite Nup60 exhibiting robust detachment (Fig. 1). By depleting Nup60 either constitutively or acutely using the auxin-AID system (Nishimura et al., 2009; Yesbolatova et al., 2020), we uncovered two distinct roles for Nup60: recruitment and retention. Although Nup60 is required to recruit both Nup2 and Mlp1, Nup60 is not required for the retention of Mlp1 at the nuclear periphery (Fig. 9). The ability of Mlp1 to remain associated with the NPC in the absence of Nup60 suggests the existence of additional interactions at the nuclear periphery. Consistent with this notion, Mlp1 is the slowest maturing member of the yeast NPC (Onischenko et al., 2020). It has been suggested that Mlp1 and Mlp2 interact to form a network between neighboring NPCs (Niepel et al., 2013), which could explain its persistence at the nuclear periphery in the absence of Nup60. Determining whether Nup60 mediates the retention of other proteins that require it for peripheral localization, such as the SUMO protease Ulp1 (Zhao et al., 2004), will provide insights into their meiotic regulation and the organizational hierarchy of the nuclear basket.

How the entire nuclear basket is detached from NPCs following meiosis II remains less well understood. Phosphorylation-resistant Nup60 mutants that fail to detach during meiosis I were still able to detach in meiosis II, indicating that the two nuclear basket detachment events are mechanistically distinct (Fig. 6). Additionally, individual nuclear basket nucleoporins detached from NPCs independently during meiosis II, with Mlp1 and Nup1 returning to gamete nuclei even when the bulk of Nup60 was targeted to the GUNC through artificial tethering (Fig. 2). It is therefore likely that modifications on multiple nucleoporins facilitate their dissociation from the NPC core, perhaps via the coordinated activity of multiple kinases as in metazoan mitosis (Laurell et al., 2011). Further elucidation of NPC remodeling in meiosis II will improve our understanding of how the basket is held together and how it can come apart.

### NPC remodeling in gametogenesis
Why NPCs exhibit such extensive modularity during meiosis remains a key area for future investigation. The basket nucleoporins Nup60 and Nup2 are required for proper gametogenesis in budding yeast (Chu et al., 2017; Komachi and Burgess, 2022). It has been hypothesized that their meiotic function involves tethering chromosomes to the nuclear periphery during prophase I, in parallel to the telomere bouquet protein Ndj1 and the linker of the nucleoskeleton and cytoskeleton (LINC) complex. Basket detachment during the meiotic divisions might therefore facilitate the release of chromatin from the nuclear envelope as

chromosomes undergo segregation. Moreover, full basket detachment during meiosis II enables inheritance of the nuclear basket but not the NPC core by nascent gametes. The metazoan basket nucleoporin Nup153 (the ortholog of ScNup60 and ScNup1) mediates NPC assembly in interphase cells (Vollmer et al., 2015); it is tempting to speculate that Nup60 and Nup1 could similarly contribute to the insertion of newly synthesized NPCs in gametes after large-scale meiotic turnover. It is worth noting that no gross defects in sporulation efficiency or gamete viability were observed for the basket mutants that affect NPC remodeling (Tables S7 and S8), suggesting the existence of redundant pathways that protect gamete health. A thorough assessment of how NPC remodeling affects cellular function promises to reveal novel roles for the nuclear basket in nuclear organization.

Strikingly, nuclear basket disassembly during meiosis I is conserved in the distantly related fission yeast *S. pombe* (Fig. 9). This conservation over 400 million years of evolution is consistent with basket detachment serving an important meiotic function (Sipiczki, 2000). It also establishes that nuclear basket detachment is an NPC remodeling event distinct from GUNC formation, since NPC sequestration does not take place in *S. pombe* (Asakawa et al., 2010). In addition to NPCs being partially disassembled, the nuclear permeability barrier is transiently disrupted during meiosis II in *S. pombe* and possibly *S. cerevisiae* (Arai et al., 2010; Asakawa et al., 2010; Shelton et al., 2021), suggesting that several hallmarks of metazoan cell divisions occur within the context of these fungal cell divisions. This work adds to growing evidence that nuclear periphery dynamics during cell divisions exist across an evolutionary spectrum rather than fitting a binary, where the nuclear periphery is either completely intact ("closed" divisions) or completely disrupted ("open" divisions).

Altogether, our study provides insights into a new type of NPC remodeling during cell divisions: nuclear basket disassembly and reassembly. Whether nuclear basket detachment occurs in other contexts, including mammalian meiosis, remains unclear. Notably, the metazoan basket nucleoporin Nup50 (the ortholog of ScNup2) was originally cloned from mice testes due to its high expression levels (Fan et al., 1997; Guan et al., 2000). During spermatogenesis, Nup50 exhibits a familiar localization pattern: it relocalizes from the nuclear periphery in the spermatocytes (before meiosis II) to the nucleoplasm in the spermatids (after meiosis II) before returning to the nuclear periphery in the spermatozoa (mature gametes). Nuclear basket detachment may therefore represent a widespread feature of gametogenesis programs. Future interrogation of how cells achieve complex and dynamic meiotic regulation of the NPC promises to improve our understanding of NPC organization and function.

## Materials and methods
### Yeast strains, plasmids, and primers
All strains and plasmids in this study are listed in Tables S1 and S2, respectively. All *S. cerevisiae* strains in this study are derivatives of SK1. Deletion and endogenous C-terminal tagging of

genes were performed using standard PCR-based methods, with the primers used listed in Table S3 (Knop et al., 1999; Longtine et al., 1998). The following alleles were constructed in a previous study: *HTB1-mCherry* and $P_{GPD}$-*GAL4-ER* (Matos et al., 2008); $P_{CUP1}$-*CDC5* (Grigaitis et al., 2020); *ndt80Δ::LEU2* (Xu et al., 1995); *ndt80Δ::NatMX* (Matos et al., 2011); $P_{TDH3}$-*CRE-EBD78* (Logie and Stewart, 1995); various fluorescently tagged nucleoporins including *NUP170-GFP, NUP120-GFP, POM34-GFP, NUP49-GFP, NUP60-GFP, NUP1-GFP, NUP2-GFP,* and *NUP84-GFP* (King et al., 2019); *SPC42-mCherry* (Miller et al., 2012); and $P_{CLB2}$-*CDC5* (cdc5-mn) and $P_{CLB2}$-*CDC20* (cdc20-mn; Lee and Amon, 2003).

To generate lipid-binding mutants (*nup60-ΔAH* [nup60-Δ2-47] and *nup60-I36R*) or phosphomutants (*NUP60-S89A, NUP60-Nterm3A, NUP60-Nterm5A, NUP60-Cterm4A,* and *NUP60-9A*; see Fig. 5 B for sites mutated in each allele) of *NUP60* at the endogenous locus, CRISPR/Cas9-directed repair was used unless otherwise noted (Anand et al., 2017). Golden Gate Assembly was used to integrate a gRNA sequence (5′-GTCGATTTTAGGATATCTCG-3′) binding upstream of the *NUP60* ORF into a *URA3*-marked centromeric plasmid containing Cas9 under the control of the *PGK1* promoter (a gift from Gavin Schlissel and Jasper Rine). This plasmid (pUB1729) and the appropriate repair template, amplified from a gBlock gene fragment (IDT), were co-transformed into yeast. All repair templates contained a mutation to abolish the PAM sequence (C→G at the −27 nt position upstream of *NUP60*) and prevent recutting. The following nucleotide changes were introduced to achieve specific point mutations: I36R (5′-ATT-3′ → 5′-CGT-3′), S89A (5′-TCC-3′ → 5′-GCG-3′), T112A (5′-ACA-3′ → 5′-GCA-3′), S118A (5′-TCC-3′ → 5′-GCC-3′), S162A (5′-TCA-3′ → 5′-GCA-3′), S171A (5′-TCG-3′ → 5′-GCC-3′), S371A (5′-TCC-3′ → 5′-GCC-3′), S374A (5′-TCA-3′ → 5′-GCA-3′), S394A (5′-TCC-3′ → 5′-GCC-3′), and S395A (5′-TCT-3′ → 5′-GCT-3′). A similar approach was used to generate the N-terminally tagged FKBP12-Nup60 with an FKBP12-containing repair template amplified from a plasmid (pUB651). Following transformation into yeast, all alleles were confirmed by sequencing. To generate *NUP60-S89A-9myc*, a single point mutation in *NUP60-9myc* was introduced by site-directed mutagenesis using primers OML1096 and OML1018 (Table S3). In brief, a mutagenic primer (S89A) was used to amplify the region coding for the C-terminus of *NUP60-9myc*, as well as the associated *TRP1* cassette. The PCR was then used to transform the wild-type cells, thus simultaneously introducing the S89A mutation and the 9myc tag. The following nucleotide change was confirmed by sequencing: S89A (5′-TCC-3′ → 5′-GCC-3′).

To generate a lipid-binding mutant of *NUP1* (*nup1-ΔAH* [nup1-Δ2-32]) at the endogenous locus, a CRISPR/Cas9 approach was used. A Cas9 plasmid containing a gRNA (5′-CTGTCCCTATAACCCTTTCG-3′) binding upstream of the *NUP1* ORF (pUB1727) was cloned, and the repair template—amplified from a gDNA gene block (IDT)—contained a mutation to abolish the PAM sequence (C→G at the −49 nt position upstream of *NUP1*). To generate N-terminally tagged *MLP1* (*FKBP12-MLP1*) at the endogenous locus, a CRISPR/Cas9 approach was again used. A Cas9 plasmid containing a gRNA (5′-GCCACATTTTAGGATAATGT-3′) overlapping with the 5′-end of the *MLP1* ORF was cloned (pUB2020), and the repair template—amplified from a plasmid

(pUB651)—contained a mutation to abolish the PAM sequence (a synonymous G→C mutation at the +6 nt position of *MLP1*). Both alleles were confirmed by sequencing.

To generate strains containing copper-inducible *CDC5* constructs ($P_{CUP1}$-*CDC5-3xFLAG-10xHis* [pUB2047] and $P_{CUP1}$-*CDC5$^{KD}$-3xFLAG-10xHis* [pUB2048]), we inserted the desired *CDC5* allele downstream of the *CUP1* promoter in a *TRP1* single integration vector backbone (pNH604, a gift from the Lim lab). The kinase-dead (KD) allele contained a K110M mutation (5′-AAA-3′ → 5′-ATG-3′) in the *CDC5* ORF (Charles et al., 1998). To generate strains carrying the β-estradiol-inducible $P_{GAL1}$-*CDC5-eGFP* (pML118) or $P_{GAL1}$-*CDC5$^{KD}$-eGFP* (pML120) alleles, the respective *CDC5* sequences were inserted into the pAG304GAL-ccdB-EGFP::TRP1 destination vector by Gateway Cloning (14183; Addgene; Hartley et al., 2000). The constructs (pML118 and pML120) were integrated at the *TRP1* locus of a strain (YML1110) carrying a $P_{GPD}$-*GAL4-ER* allele at the *URA3* locus. To generate strains containing a 5-Ph-IAA responsive F-box receptor osTIR1 ($P_{CUP1}$-*osTIR1$^{F74G}$*), kinase, ligase, and Dpn1 (KLD) site-directed mutagenesis was performed for $P_{CUP1}$-*osTIR1* in a *HIS3* integrating vector. osTIR1$^{F74G}$ exhibits lower background degradation since it is nonresponsive to the endogenous auxin present in yeast (Yesbolatova et al., 2020). All constructs were integrated into the genome following digestion with PmeI.

To generate Nup2 with a RITE cassette (Nup2-RITE; Verzijlbergen et al., 2010), we first replaced RFP in a RITE tagging cassette (V5-LoxP-HA-GFP-LoxP-T7-mRFP, a gift from the van Leeuwen lab) with mCherry (V5-LoxP-HA-GFP-LoxP-T7-mCherry, pUB1198). We then C-terminally tagged Nup2 at its endogenous locus with a standard PCR-based approach using specific RITE adaptors.

## Sporulation conditions

For *S. cerevisiae*, strains were first grown in YPD (1% yeast extract, 2% peptone, 2% glucose, 22.4 mg/l uracil, and 80 mg/l tryptophan) at room temperature (RT) or 30°C for ~20–24 h until saturation was reached ($OD_{600}$ ≥10). Cultures were then back-diluted in BYTA (1% yeast extract, 2% bacto tryptone, 1% potassium acetate, and 50 mM potassium phthalate) to an $OD_{600}$ = 0.25 and incubated at 30°C for ~14–20 h until reaching an $OD_{600}$ ≥5. To induce sporulation, cells were pelleted, washed with MilliQ water, and resuspended in sporulation media (SPM: 2% potassium acetate with 0.04 g/l adenine, 0.04 g/l uracil, 0.01 g/l histidine, 0.01 g/l leucine and 0.01 g/l tryptophan, adjusted to a pH of 7) at an $OD_{600}$ = 1.85. Cultures were maintained at 30°C for the duration of the experiment. During all steps, cultures were placed on a shaker and in flasks that had 10× culture volume to ensure proper aeration.

For Fig. 4 and Fig. S4, C–E, H and I, meiotic time courses were performed with diploid SK1 strains obtained from mating *MAT*a and *MAT*α haploids, as previously described (Oelschlaegel et al., 2005; Petronczki et al., 2006). In brief, cells were grown for 48 h at 30°C on YPG (2% glycerol) plates as single colonies and subsequently propagated as a thin lawn on YPD plates. Cells were used to inoculate the presporulation medium YPA (2% potassium acetate) at $OD_{600}$ = 0.3 and cultured for 11 h at 30°C on a shaker. Meiotic induction was initiated by switching cells to sporulation medium (SPM*, 2% potassium acetate) at $OD_{600}$ = 3.5–4. The time

of inoculation in SPM* was defined as t = 0. Cultures were up-scaled to a 10-L fermenter system for MS sample collection, as previously described (Grigaitis et al., 2018). $P_{GAL1}$-CDC5-eGFP or $P_{GAL1}$-CDC5$^{KD}$-eGFP were induced by the addition of β-estradiol to a final concentration of 2 μM. Cell cycle stage distribution was tracked by analyzing the cellular DNA content using propidium iodide staining on a FACS Canto cell sorter.

For *S. pombe,* strains were mated on sporulation agar medium with supplements (SPA5S; Petersen and Russell, 2016). After 15–18 h, a small toothpick of the mating patch was resuspended in 100 μl ddH₂O and 10 μl was spotted onto a new SPA5S plate. This region was then removed and transferred to a 35-mm glass bottom dish (MaTek, no. 1.5 coverslip) for imaging.

### RITE experiments

For experiments involving RITE tags, diploid cells were taken out of the freezer on a YPG plate before being transferred to a hygromycin plate to maintain selection for the intact RITE cassette. Cultures of YPD with hygromycin (200 μg/ml) were inoculated at a very low $OD_{600}$ (<0.05 OD/ml) using yeast from the hygromycin plate resuspended in YPD. The cultures were allowed to grow at RT for ~16–18 h until reaching the log phase (0.2 OD/ml < $OD_{600}$ < 2.0 OD/ml). The cultures were then back-diluted into reduced YPD (same as YPD except 1% glucose and no added tryptophan; Chia and van Werven, 2016) without hygromycin at an $OD_{600}$ = 0.2 OD/ml and allowed to grow for ~12 h until reaching saturation. Cassette conversion was induced by the addition of 1 μM β-estradiol (which activates Cre recombinase fused to estrogen binding domain [Cre-EBD]) and allowed to continue for ~12 h. Cells were then induced to sporulate in SPM, as described above.

Before and after Cre induction, cells were plated onto YPD, with a dilution series performed to get ~200–400 cells per plate. After ~48 h of growth at 30°C, the colonies were replica plated onto hygromycin plates. Colony growth was scored after 24 h to assess the presence or absence of the intact RITE cassette. For live-cell imaging experiments, individual cells were determined to have undergone tag exchange by confirming reduced tetrad Nup2-GFP signal intensity relative to tetrads from a strain without Cre recombinase.

### Fluorescence microscopy
#### Budding yeast imaging

All *S. cerevisiae* wide-field fluorescence microscopy was performed with a DeltaVision Elite microscope (GE Healthcare) using a 60×/1.42 NA oil-immersion objective and a PCO Edge sCMOS camera. For the specific acquisition settings used in each experiment, see Table S4. For live-cell imaging, the CellASIC ONIX2 platform (EMD Millipore) was used to maintain proper media flow and execute different treatment regimens. Cell loading was performed at 8 psi for 5 s, and media flow was maintained at 2 psi for the duration of the experiment. The acquisition was performed using Y04E microfluidic plates in a temperature-controlled chamber at 30°C.

For meiotic live-cell imaging experiments, cells at an $OD_{600}$ = 1.85 in SPM (see above for culture conditions) were sonicated and loaded onto a microfluidics plate. Conditioned SPM (prepared by

filter-sterilizing a sporulating culture of a diploid yeast strain after ~5 h at 30°C) was used to facilitate meiotic progression. For mitotic live-cell imaging experiments, cells were grown overnight in synthetic complete (SC) media (2 g dropout powder [US Bio Life Sciences], 3.3 g yeast nitrogen base containing ammonium sulfate, and 10 g dextrose in 500 ml MilliQ) and then diluted to an $OD_{600}$ = 0.1 OD/ml in fresh SC. After 5 h of growth, cells were sonicated and loaded onto a microfluidics plate. Fresh SC was used to facilitate vegetative growth.

For time course staging, 500 μl of meiotic culture was fixed in 3.7% formaldehyde for 15 min at RT. Cells were washed in 0.1 M potassium phosphate pH 6.4, resuspended in KPi sorbitol buffer (0.1 M potassium phosphate, 1.2 M sorbitol, pH 7.5), and stored at 4°C prior to imaging.

#### Fission yeast imaging

All *S. pombe* fluorescence microscopy was performed on a Nikon Ti-E microscope with a CSI W1 spinning disk (Yokogawa) using a 100 ×/ 1.4 NA Olympus Plan Apo oil objective and an iXon DU897 Ultra EMCCD camera (Andor). Fluorophores were excited at 488 nm (GFP) and 561 nm (mCherry) and collected through an ET525/36 m (GFP) or ET605/70 m (mCherry) bandpass filter. Images were collected over a z-volume of 8 μm with 0.5 μm spacing for 4–5 h at 5-min intervals. Image acquisition conditions for all experiments are listed in Table S4. Live imaging was performed in 35-mm glass bottom dishes (MaTek, no. 1.5 coverslip) maintained at 25°C using an Oko Lab stage top incubator.

For imaging meiotic divisions, strains were prepared as specified above. For imaging mitotic divisions, strains were grown in yeast extract with supplements (YES; 5 g yeast extract, 30 g dextrose, 0.2 g each adenine, uracil, histidine, leucine, and lysine, in 1 L of water) and maintained in exponential growth for at least 2 d prior to imaging. The 35-mm glass bottom dishes were coated with 1 mg/ml soybean lectin (in water) for 15 min, rinsed with 1 ml YES media, and 200 μl of log phase cell culture was added and allowed to settle for 30 min at 30°C. Cells were rinsed twice with prewarmed YES media and then imaged.

#### Super-resolution microscopy

Samples for SIM were prepared by fixing 500 μl of meiotic cultures in 4% paraformaldehyde supplemented with 200 mM glucose for 15–20 min at RT. Cells were then washed and resuspended in phosphate-buffered-saline (PBS), prior to being stored at 4°C until imaging. Fixed cells were imaged in PBS on an Applied Precision OMX Blaze V4 (GE Healthcare) microscope using a 60×/ 1.42 NA Olympus Plan Apo oil objective and two PCO Edge sCMOS cameras. GFP and mCherry were imaged with 488 nm (GFP) or 561 nm (mCherry) lasers with alternating excitation using 405/488/561/640 dichroic with 504–552 and 590–628 nm emission filters. Image stacks were acquired over a volume covering the full nucleus (typically 3–4 μm) with a z-spacing of 125 nm. Images were reconstructed in SoftWoRx (Applied Precision Ltd) using a Wiener filter of 0.001.

### Image quantification and statistics

All image analysis was performed in FIJI (RRID:SCR_002285; Schindelin et al., 2012). Maximum z-projection and single

z-slices are shown for each image and were modified for presentation using linear brightness and contrast adjustments in FIJI. For any experiments in which successful sporulation was possible, only cells that became normally packaged tetrads were analyzed.

Two methods were used to quantify nucleoporin detachment in *S. cerevisiae*. Unless otherwise noted, individual cells were cropped from deconvolved images with background subtracted using a rolling ball method with a radius of 15 pixels. For the time points of interest, the individual z-slices containing the middle of the nucleus were selected and a nuclear mask was generated using the Htb1-mCherry signal. This nuclear mask was eroded and then dilated to generate masks for the nucleoplasmic and nuclear envelope regions, respectively. These masks were used to measure the intensities for each region in the Nup-GFP channel. The signal corresponding to the nuclear envelope was derived by subtracting the nucleoplasmic intensity from the total nuclear intensity. The detachment index (DI) was calculated for each nucleus image by dividing the mean nucleoplasmic intensity by the mean nuclear envelope intensity.

During mitotic divisions, we observed the exclusion of Htb1-mCherry from portions of the nucleoplasm (presumably the nucleolar space). This prevented us from reliably segmenting the nucleoplasm using the masking approach outlined above. To quantify DI in mitosis (Fig. S2 F), we, therefore, measured Nup-GFP intensity along a five-pixel line profile across middle z-slices of nuclei in FIJI after background subtraction. These profiles were used as input for peak detection using the "find_peaks" function from scipy.signal (https://scipy.org), with occasional manual correction of NE peak detection in instances where peaks were misidentified. The intensities of the two NE peaks were averaged to derive a mean NE intensity, while the nucleoplasmic intensity was calculated as the mean value for the middle 50% of the distance between the two peaks (to avoid signal from the tails of the NE peaks). For comparisons of DI in mitotic and meiotic anaphase (Fig. S2 F), meiosis images were reanalyzed using the line profile method. Raw data from these measurements are available at http://www.stowers.org/research/publications/libpb-1689, and an example Jupyter notebook containing the source code used for peak calling is available at https://github.com/jmvarberg/King_et_al_2022.

Throughout the manuscript, individual DI values were normalized to the mean intensity of a reference group, as indicated in the figure legends. We note that changes in nuclear geometry during the meiotic divisions seem to affect DI measurements (compare "pre" and "anaphase I" values for the transmembrane nucleoporin Pom34-GFP, Fig. 1 C), perhaps because a smaller effective nuclear volume (e.g., a single nuclear lobe during anaphase I vs. the entire nucleus during prophase I) results in more "nucleoplasmic" signal from out of focus z-slices. As such, we used Pom34-GFP as our reference for different meiotic stages when initially assessing detachment. Downstream analysis, plotting, and statistics were conducted using R 4.1.1 (R Core Team, 2020). Information on the specific statistical test used for each experiment is provided in the corresponding figure legends. The full summary statistics and statistical test output for each experiment are provided in Table S5, and all source

codes used for analysis are provided at https://github.com/jmvarberg/King_et_al_2022. For all experiments, asterisks represent the following P values: *, $P < 0.05$; **, $P < 0.01$; ***, $P < 0.001$; ****, $P < 0.0001$.

To visualize nucleoporin detachment in *S. pombe*, line profiles were measured in FIJI using a line width of five pixels on a single middle image of nuclei at the indicated time points relative to SPB separation (visualized using SpPpc89-mCherry). Individual line profiles were normalized (min-max) and plotted in Graph-Pad Prism (v. 9.0).

To quantify the timing of Nup60 detachment relative to other meiotic events (Fig. S1 A and Fig. 3 B), the following definitions were used: "detachment" was defined as the first time point with near maximum relocalization of Nup60-GFP to the nucleoplasm, and "reattachment" was defined as the first time point with near maximum relocalization of Nup60-GFP to the nuclear periphery. To quantify Mlp1-GFP mislocalization (Fig. 8 F and Fig. S7 E), the following definitions were used: "normal localization" was defined as Mlp1-GFP signal/puncta distributed relatively evenly around the nuclear periphery; "mild mislocalization" was defined as one or two major Mlp1-GFP puncta with additional weaker peripheral Mlp1-GFP localization, or clustered and uneven peripheral Mlp1-GFP signal; "severe mislocalization" was defined as almost all Mlp1-GFP signal existing in one or two major puncta; and "diffuse signal" was defined as no clear enrichment of Mlp1-GFP at the nuclear periphery.

## Immunoblotting
### For all figures, unless otherwise noted

To prepare protein samples for immunoblotting, trichloroacetic acid (TCA) extraction was performed. For each time point, 3.33 $OD_{600}$ of cells were fixed with 5% TCA for ≥15 min at 4°C. Cells were washed with 1 ml 10 mM Tris pH 8.0 and then 1 ml acetone before the pellets were allowed to completely dry overnight. Glass beads (~100 µl) and 100 µl protein breakage buffer (50 mM Tris-HCl pH 8.0, 1 mM EDTA, 20 mM Tris pH 9.5, 3 mM DTT, and 1× cOmplete EDTA-free protease inhibitor cocktail [Roche]) were added to the samples, which were then pulverized for 5 min using the Mini-Beadbeater-96 (BioSpec). To denature proteins, 50 µl of 3× SDS buffer (250 mM Tris pH 6.8, 8% β-mercaptoethanol, 40% glycerol, 12% SDS, and 0.00067% bromophenol blue) was added to the samples prior to incubation at 95°C for 5 min.

To visualize the proteins, the samples were then subjected to polyacrylamide gel electrophoresis. Samples were run on 4–12% Bis-Tris Bolt gels (Thermo Fisher Scientific) at 100 V for 90 min along with the Precision Plus Protein Dual Color Standards (Bio-Rad) ladder to determine protein size. Transfer to a nitrocellulose membrane (0.45 µm; Bio-Rad) was performed at 4°C in a transfer buffer (25 mM Tris, 195 mM glycine, and 15% methanol) using a Mini-PROTEAN Tetra tank (Bio-Rad) run at 70 V for 3 h. Ponceau S (Sigma-Aldrich) staining was often performed to assess protein transfer, followed by blocking in PBS Intercept Blocking Buffer (LI-COR Biosciences). Blots were incubated overnight in primary antibody diluted in PBS Intercept Blocking Buffer at 4°C. The following primary antibodies were used: 1:2,000 mouse anti-GFP (RRID:AB_2313808, 632381; Clontech);

1:2,000 mouse anti-V5 (RRID:AB_2556564; Invitrogen); 1:10,000 rabbit anti-hexokinase (RRID:AB_219918, 100-4159; Rockland); and 1:1,000 rabbit anti-FLAG (RRID:AB_2217020, 2368; Cell Signaling Technology). Blots were then washed in PBST (PBS with 0.1% Tween-20) and incubated in secondary antibody diluted in PBS Intercept Blocking Buffer + 0.1% Tween-20 for 3–6 h at RT. The following secondary antibodies were used: 1:15,000 donkey anti-mouse conjugated to IRDye 800CW (RRID: AB_621847, 926–32212; LI-COR Biosciences) and 1:15,000 goat anti-rabbit conjugated to IRDye 680RD (RRID:AB_10956166, 926–68071; LI-COR Biosciences). Blots were then washed in PBST and PBS before imaging using the Odyssey CLx system (LI-COR Biosciences). Image analysis and quantification were performed in FIJI (RRID:SCR_002285, Schindelin et al., 2012).

### For Fig. 4; and Fig. S4, C–E, H, and I

TCA extracts were performed as described previously (Matos et al., 2008). Briefly, pellets from meiotic cultures (OD$_{600}$ ~3.5, 10 ml) were disrupted in 10% TCA using glass beads. Precipitates were collected by centrifugation, resuspended in 2× NuPAGE sample buffer (50% 4× NuPAGE, 30% H$_2$O, 20% DTT 1 M), and neutralized with 1 M Tris-Base. Samples were incubated for 10 min at 95°C, cleared by centrifugation, and separated in NuPAGE 3–8% Tris-Acetate gels (Invitrogen). Proteins were then transferred onto PVDF membranes (Amersham Hybond 0.45 μm, GE Healthcare). For immunoblotting, the following antibodies were used: 1:15,000 rabbit anti-Myc HRP-conjugated (RRID:AB_299800), 1:5,000 rabbit anti-Crm1 (a gift from K. Weis, ETH Zurich, Onischenko et al., 2009), 1:5,000 rabbit anti-Puf6 (a gift from V. Panse, University of Zurich, Gerhardy et al., 2021), 1:2,500 mouse anti-Cdc5 clone 4F10 (MM-0192-1-100; MédiMabs). The following secondary antibodies were used: 1:5,000 goat anti-mouse immunoglobulin conjugated to HRP (RRID:AB_2617137, P0447; Agilent) and 1:5,000 swine anti-rabbit immunoglobulin conjugated to HRP (RRID:AB_2617141, P0399; Agilent).

### MS data acquisition and analysis
#### Sample preparation
100 ml of meiotic yeast culture OD$_{600}$ = 3.5 in SPM* (2% KAc) was collected per time point and supplemented with 100% TCA to a final concentration of 6.25%. Samples were washed twice with cold acetone, pelleted, and snap-frozen in liquid nitrogen. Proteins were extracted in batches of 12 samples by 5 min of bead beating using 400 μl of 425–600 μm glass beads in a cell Disruptor Genie and 400 μl of lysis buffer (8 M urea, 100 mM ammonium bicarbonate, 5 mM EDTA, pH = 8). The samples were then centrifuged at 16,000 × g for 10 min and the supernatant was collected. The extraction procedure was repeated for a total of five times using 400 μl fresh lysis buffer each time. The total amount of protein was measured with bicinchoninic acid (BCA) assay (Pierce) and 3 mg of total protein was further used per sample. Samples were sequentially incubated with (1) 5 mM Tris (2-carboxyethyl) phosphine (TCEP) for 1 h at 25°C, (2) 12 mM iodoacetamide for 1 h at 25°C in dark, (3) diluted eight times with 100 mM ammonium bicarbonate (reducing urea to 1 M), and (4) trypsin-digested overnight at 37°C (protein to

trypsin ratio 1:100). Sample pH was reduced to ~2.5 with formic acid (FA) and the peptides were enriched on C18 silica reversed-phase chromatography columns (Sep-Pak C18 3cc, Waters). Peptides were eluted in 50% acetonitrile (ACN) and 0.1% FA, SpeedVac-dried, and reconstituted in 50 μl 0.1% FA. For the total proteome assessment ("non-enriched"), 1 μl was taken up and diluted to 1 μg/μl. The remaining sample was used for phosphopeptide enrichment by TiO$_2$ affinity purification. To this end, samples were incubated for 1 h RT on a rotator with 6.25 mg of TiO$_2$ resin (GL Science) pre-equilibrated twice with methanol and twice with lactic acid binding buffer (5 ml ACN, 2.92 ml lactic acid, 20 μl 50% trifluoroacetic acid (TFA), filled with H$_2$O to 10 ml). Beads were washed sequentially twice with lactic acid binding buffer, then 80% ACN 0.1% TFA, and 0.1% TFA. Phosphopeptides were eluted in 50 mM ammonium phosphate pH = 10.8 and acidified immediately to pH 2 with 50% FA. Samples were cleaned on C18 Micro Spin Columns (The Nest group), eluted in 50% ACN 0.1% FA, SpeedVac-dried, and resuspended in 0.1% FA. All samples were spiked with iRT reference peptides (Biognosys). Samples for library preparation consisted of the following triplicate pools of selected time points after Cdc5 or Cdc5$^{KD}$ induction: t = 0.5, 2, and 5 h.

#### Mass spectrometry data acquisition
1 μg of peptides was injected on a 6600 Sciex TripleTOF mass spectrometer interfaced with an Eksigent NanoLC Ultra 1D Plus system. The peptides were separated on a 75-μm-diameter, 20-cm-long new Objective emitter packed with Magic C18 AQ 3 μm resin (Michrom BioResources) and eluted at 300 nl/min with a linear gradient of 2–30% Buffer B for 60 min (Buffer B: 98% ACN, 0.1% FA). MS data acquisition for the individual meiotic time course samples was performed in data-independent acquisition (DIA) SWATH-MS mode using 64 variable precursor isolation windows with 1-Da overlaps acquired each for 50 ms plus one MS1 scan acquired for 250 ms, as described in Gillet et al. (2012). Library generation was performed in data-dependent acquisition mode (DDA, top20, with 20 s dynamic exclusion after 1 MS/MS). For either mode, the mass ranges recorded were 360–1,460 m/z for MS1 and 50–2,000 m/z for MS2, and the collision energy was set to 0.0625 × m/z—6.5 with a 15-eV collision energy spread regardless of the precursor charge state.

#### Data-dependent acquisition (DDA) mode data analysis
The DDA search and spectral library were performed essentially as described in Schubert et al. (2015). In short, the raw DDA files were converted to mzXML using gtofpeakpicker (ProteoWizard v 3.0.9992) and further to mgf using MzXML2Search (TPP 4.7). The converted files were searched with Comet (2014.02 rev. 0) and Mascot (version 2.5.1) using the yeast SGD database (release 13.01.2015) appended with the SK1 (for a total containing 12,043 proteins including one protein entry for the concatenated sequence of the iRT peptides and as many decoy protein entries generated by pseudo-reversing the tryptic peptide sequences) or the W303 entries (for a total containing 12,071 proteins including one protein entry for the concatenated sequence of the iRT peptides and as many decoy protein entries generated by

pseudo-reversing the tryptic peptide sequences) for data acquired for the SK1 or W303 strain, respectively. The search parameters were as follows: ±25 ppm tolerance for MS1 and MS2, fixed cysteine carbamidomethylation, either variable methionine oxidation (for the non-enriched datasets) or variable methionine oxidation and variable serine/threonine/tyrosine phosphorylation (for the phospho-enriched datasets), and semitryptic and two missed cleavages were allowed. The comet and mascot search results were further processed using peptide-Prophet (Keller et al., 2002) and aggregated using iProphet (Shteynberg et al., 2011; TPP v4.7 rev 0). The search results were filtered for an iProphet cutoff of 1% false discovery rate. The pep.xml file was further processed with LuciPhor for determining the confidence in the phosphorylation site localization. Two consensus spectral libraries were generated using spectrast (Lam et al., 2008) depending on the confidence in phosphorylation-site localization, and the two assay libraries thereof were exported using the spectrast2tsv.py script (Schubert et al., 2015) with the following parameters: five highest intensity fragments (of charge 1+ or 2+) per peptide, within the mass range 350–2,000 m/z, allowing fragments with –79.97 or –97.98 neutral losses, and excluding the fragments within the precursor isolation window of the corresponding SWATH. The final library was retained in priority assays from the high-confidence localization and then the assays from the low-confidence localization. The assay library was finally exported to TraML with shuffled decoys appended as described in Schubert et al., 2015.

### SWATH-MS targeted data extraction and data filtering
The SWATH-MS data were extracted with the above-mentioned assay library through the iPortal interface (Kunszt et al., 2015) with openSWATH (Röst et al., 2014; openMS 2.1.0), pyProphet (Teleman et al., 2015), and TRIC alignment (Röst et al., 2016) using the same parameters as described in Navarro et al. (2016) and further processed in R. The precursor intensities were first $log_2$-transformed and normalized (mean-centering). Assays identified in two out of three triplicates for at least one condition were kept. The missing values were imputed at the precursor level using either a random value amongst a distribution centered at the mean of the other replicate values of that triplicate series (when at least 1 value was found for that triplicate) or centered on a value threefold lower than the lowest value of that precursor, and with a standard deviation equal to the mean standard deviation of all the replicate precursor values. All the precursor intensities were then summed to a protein intensity value for the non-enriched datasets.

The raw dataset consisted of 14,710 phosphopeptides with single and multiphosphorylated sites that were annotated to the yeast proteomes W303 and SK1 (17,387 proteins; Engel et al., 2014). W303 is identical to the proteome release S288C_reference_genome_R64-2-1_20150113.tgz available at http://sgd-archive.yeastgenome.org/sequence/S288C_reference/genome_releases/. The median intensities of phosphosites mapping to the same amino acid and position per protein were merged to generate 7,552 unique phosphopeptides. Additionally, multi-phosphorylated peptides were split into single phosphosites and

they were retained for further analysis if they were not part of the set of monophosphorylated peptides.

The $log_2$ ratios in Fig. 4, B and C comparing Cdc5$^{WT}$ vs Cdc5$^{KD}$ were calculated using the mean values across biological replicates and time points (2, 3, 4, and 5 h after Cdc5 induction). The reported P values were corrected for false discovery rate (FDR) and tested against an α value of 0.05. All statistical methods were implemented using R v3.5.1 (R Core Team, 2020) and the limma package v3.38.3 (Ritchie et al., 2015).

A few considerations are important to correctly interpret this dataset. (1) Cdc5-mediated phosphorylation of a peptide may not necessarily lead to an increase in its abundance. Protein phosphorylation can be linked to the co-occurrence of other PTMs or to protein degradation, which could lead to a decrease in the ability of our approach to detect the phosphorylated peptide. In addition, the low expression level of some proteins could intrinsically limit their detection by SWATH-MS. As such, some, possibly many, Cdc5 targets may be absent in the current dataset. (2) Phosphorylation of a peptide that is prephosphorylated at a different site can result in a decrease in the abundance of the prephosphorylated peptide without the detection of the double phosphorylated peptide species. A possible example of this is shown in Fig. 4 D and Fig. S4 H, in which Swi6 phosphorylation at S176 decreases sharply upon Cdc5 expression. This could be explained by the modification of additional sites in the vicinity, which would render the double/multiple phosphorylated peptide difficult to monitor by MS. Therefore, a decrease in abundance in cells expressing Cdc5 (Cdc5/Cdc5$^{KD}$) could indicate that another site in the vicinity is phosphorylated in response to Cdc5. (3) Given the complexity of the samples analyzed, it is likely that the putative Cdc5 target sites identified constitute only a subset of all residues phosphorylated in cells. Due to these considerations, complementary approaches should be used to further validate individual Cdc5 targets, as well as to expand on the identity of the phosphorylation sites when studying them in-depth.

### Online supplemental material
Fig. S1 contains data pertaining to the meiotic behavior of nucleoporins from various NPC subcomplexes. Fig. S2 contains data pertaining to nuclear basket behavior during S. cerevisiae mitosis. Fig. S3 contains data pertaining to two distinct nuclear basket remodeling events during budding yeast meiosis. Fig. S4 contains data pertaining to Cdc5-dependent phosphorylation of Nup60 and other targets. Fig. S5 contains data pertaining to Nup60 phosphomutant phosphorylation and localization. Fig. S6 contains data pertaining to the meiotic role of Nup60's AH. Fig. S7 contains data pertaining to gamete basket misorganization in the absence of peripheral Nup60. Fig. S8 contains data pertaining to nuclear basket behavior during S. pombe meiosis. Fig. S9 contains data pertaining to nuclear basket behavior during S. pombe mitosis. Table S1 lists all strains used in this study. Table S2 lists all plasmids used in this study. Table S3 lists all primers used for gene deletion and C-terminal tagging in this study. Table S4 contains the imaging conditions used in this study. Table S5 contains information about the statistics and significance values used in the figures. Table S6 contains the SWATH-

MS data used to identify phosphopeptides that accumulate in response to Cdc5 activity, including protein identifier, modified site, log₂ fold change, and P values. Table S7 contains sporulation efficiency values for various alleles used in this study. Table S8 contains gamete viability values for various alleles used in this study.

### Data availability

All the reagents used in this study are available upon request. All of the ImageJ macros and R scripts used for image analysis, plotting, and statistics are available at http://www.stowers.org/research/publications/libpb-1689 and at https://github.com/jmvarberg/King_et_al_2022. For the proteomics screen to identify novel Cdc5 targets, all raw MS data files and search results thereof have been deposited to the ProteomeXchange consortium via the PRIDE (Perez-Riverol et al., 2019) partner repository with the dataset identifier PXD033245. The raw MS data files and search results originating from an independent proteomics dataset (Wettstein et al., in preparation) contributing to the phosphosite identifications in Fig. 5 have been deposited to the ProteomeXchange with identifier PXD033675.

### Acknowledgments

We thank Cyrus Ruediger, Tina Sing, Eric Sawyer, Kate Morse, Anthony Harris, Amanda Su, Andrea Higdon, Emily Powers, Helen Vander Wende, Gloria Brar, Maia Reyes, and Benjamin Styler for comments on this manuscript, and Jasper Rine, Rebecca Heald, James Olzmann, Jeremy Thorner, and the entire Brar-Ünal Lab for valuable discussions.

This work was supported by funds from the National Institutes of Health (R01AG071801 and DP2AG055946-01) and Astera Institute to E. Ünal; a National Science Foundation Graduate Research Fellowship (DGE 1752814) and a National Institutes of Health Traineeship (T32 GM007232) to G.A. King and M.E. Walsh; ETH Zürich, the Swiss National Science Foundation (155823 and 176108), the Max Perutz Labs, and the University of Vienna to R. Wettstein and J. Matos; a Ruth L. Kirschstein NRSA Postdoctoral Fellowship (F32GM133096) to J.M. Varberg; The Stowers Institute for Medical Research to S.L. Jaspersen; the Helmut Horten Stiftung and the ETH Zurich Foundation to P. Beltrao; and a European Research Council AdvG grant (670821) to R. Aebersold. The content is solely the responsibility of the authors and does not necessarily represent the official views of the National Institute of Health. We thank Zulin Yu and Jay Unruh at the Stowers Institute for assistance in super-resolution microscopy data collection, and Jennifer Gerton and members of the Gerton lab for their support and discussion. We acknowledge the technical support from Cyrus Ruediger, Alena Bishop, and Jordan Ngo.

Author contributions: G.A. King, R. Wettstein, J. Matos, and E. Ünal designed the research. G.A. King, R. Wettstein, J.M. Varberg, K. Chetlapalli, and M.E. Walsh performed the experiments. G.A. King and J.M. Varberg performed the image analysis. R. Wettstein, L. Gillet, C. Hernández-Armenta, P. Beltrao, and R. Aebersold performed the mass spectrometry processing and analysis. J.M. Varberg and S.L. Jaspersen provided technical

insight and assistance in executing super-resolution imaging. G.A. King and E. Ünal wrote the original draft of the manuscript. All authors edited and revised the final manuscript.

Disclosures: The authors declare no competing interests exist.

Submitted: 15 April 2022

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

# Supplemental material

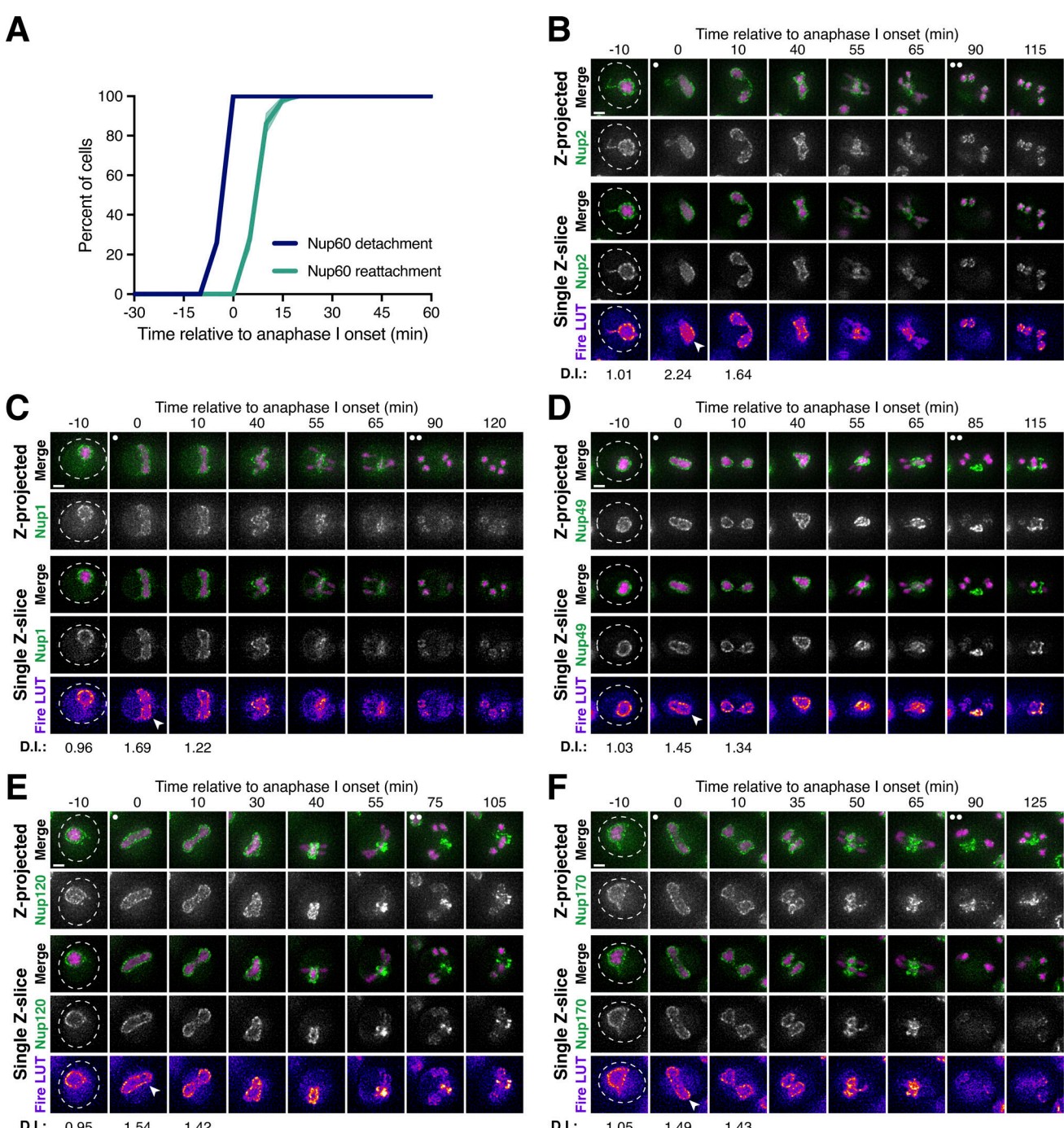

Figure S1.   **Supporting data pertaining to the meiotic behavior of nucleoporins from various NPC subcomplexes. (A)** Quantification of Nup60-GFP detachment and reattachment timing relative to anaphase I onset, corresponding to Fig. 1 D. The mean ± range (shaded range) of two independent biological replicates is displayed (*n* = 58 cells for replicate 1 and 53 cells for replicate 2). **(B–F)** Montages of cells with different fluorescently tagged nucleoporins and Htb1-mCherry, a histone, progressing through meiosis: (B) Nup2-GFP, a nuclear basket nucleoporin (UB15305); (C) Nup1-GFP, a nuclear basket nucleoporin (UB15303); (D) Nup49-GFP, a channel nucleoporin (UB13509); (E) Nup120-GFP, a Y-complex nucleoporin (UB13499); and (F) Nup170-GFP, an inner ring complex nucleoporin (UB11513). For each montage, normalized DI values (relative to the average value at the pre-anaphase I time point for each nucleoporin) are indicated when calculated. The onset of anaphase I was defined as the Htb1-mCherry chromatin mass exhibiting distortion from a spherical shape consistent with chromosome segregation. The white arrowheads in the "Fire LUT" images denote nuclei at the onset of anaphase I, the stage when Nup60-Nup2 detachment is observed. Scale bars, 2 µm.

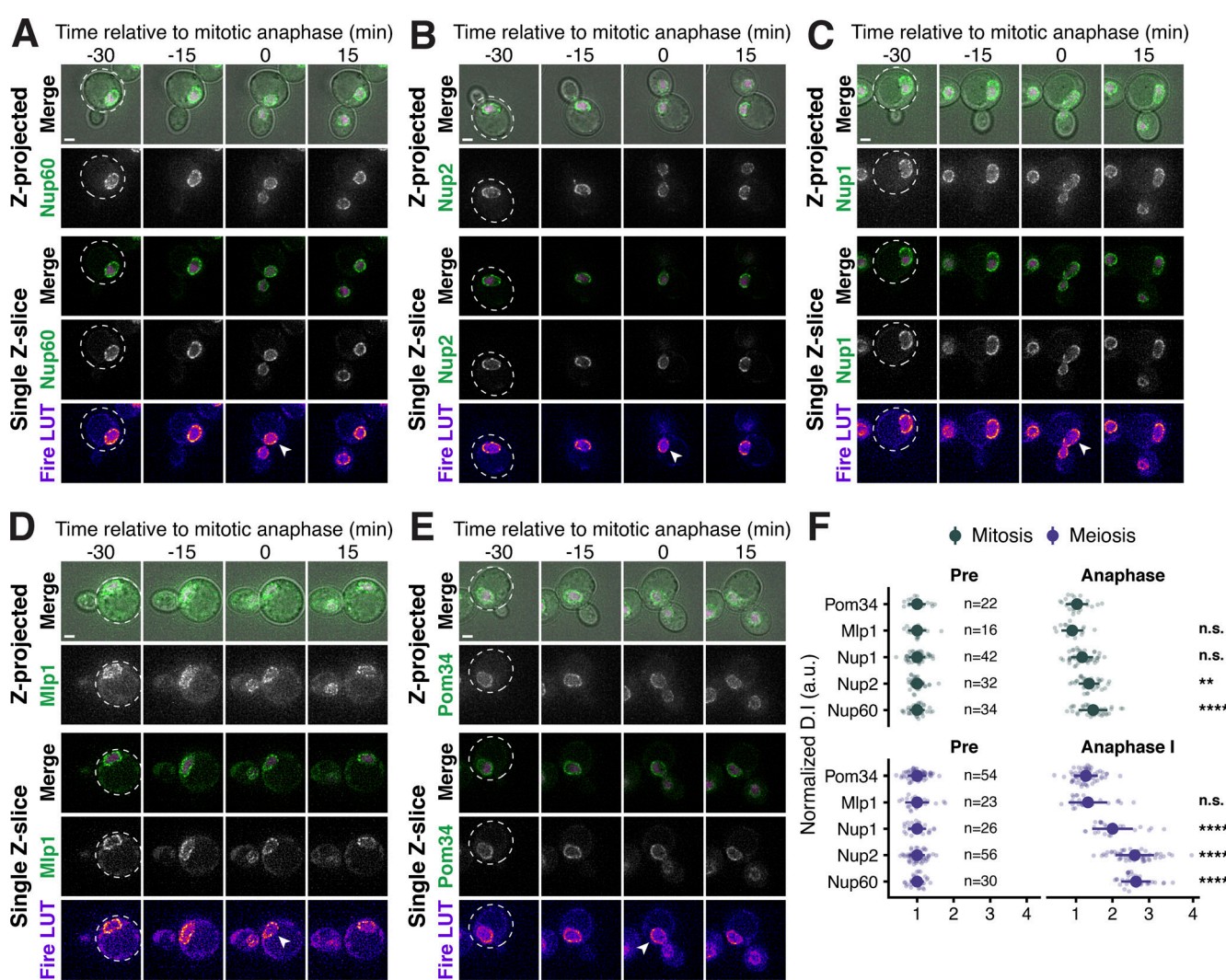

Figure S2.  **Supporting data pertaining to nuclear basket behavior during *S. cerevisiae* mitosis. (A–E)** Montages of strains with GFP-tagged nucleoporins and a fluorescently tagged histone Htb1-mCherry during mitosis. (A) Nup60-GFP (UB14646); (B) Nup2-GFP (UB15305); (C) Nup1-GFP (UB15303); (D) Mlp1-GFP (UB14648); and (E) Pom34-GFP (UB13503). Mitotic anaphase was defined as the first time point when a significant fraction of the histone mass had traversed the bud neck; the white arrowheads in the "Fire LUT" images denote nuclei at this stage. **(F)** Quantification of nucleoporin detachment before (−15 min for mitosis and −10 min for meiosis; "Pre") and coincident with mitotic anaphase or meiotic anaphase I (0 min; "Anaphase"). The detachment index (DI) for individual cells was calculated from single z-slices using a line profile method. For each nucleoporin during mitosis or meiosis, individual DI values were normalized to the average DI at the "Pre" time point. Asterisks indicate statistical significance calculated using Dunn's test for multiple comparisons when each nucleoporin was compared to Pom34-GFP, a transmembrane nucleoporin, for a given time point and type of cell division (see Table S5 for P values). Sample sizes (*n*) are the number of cells quantified for each nucleoporin. Scale bars, 2 µm.

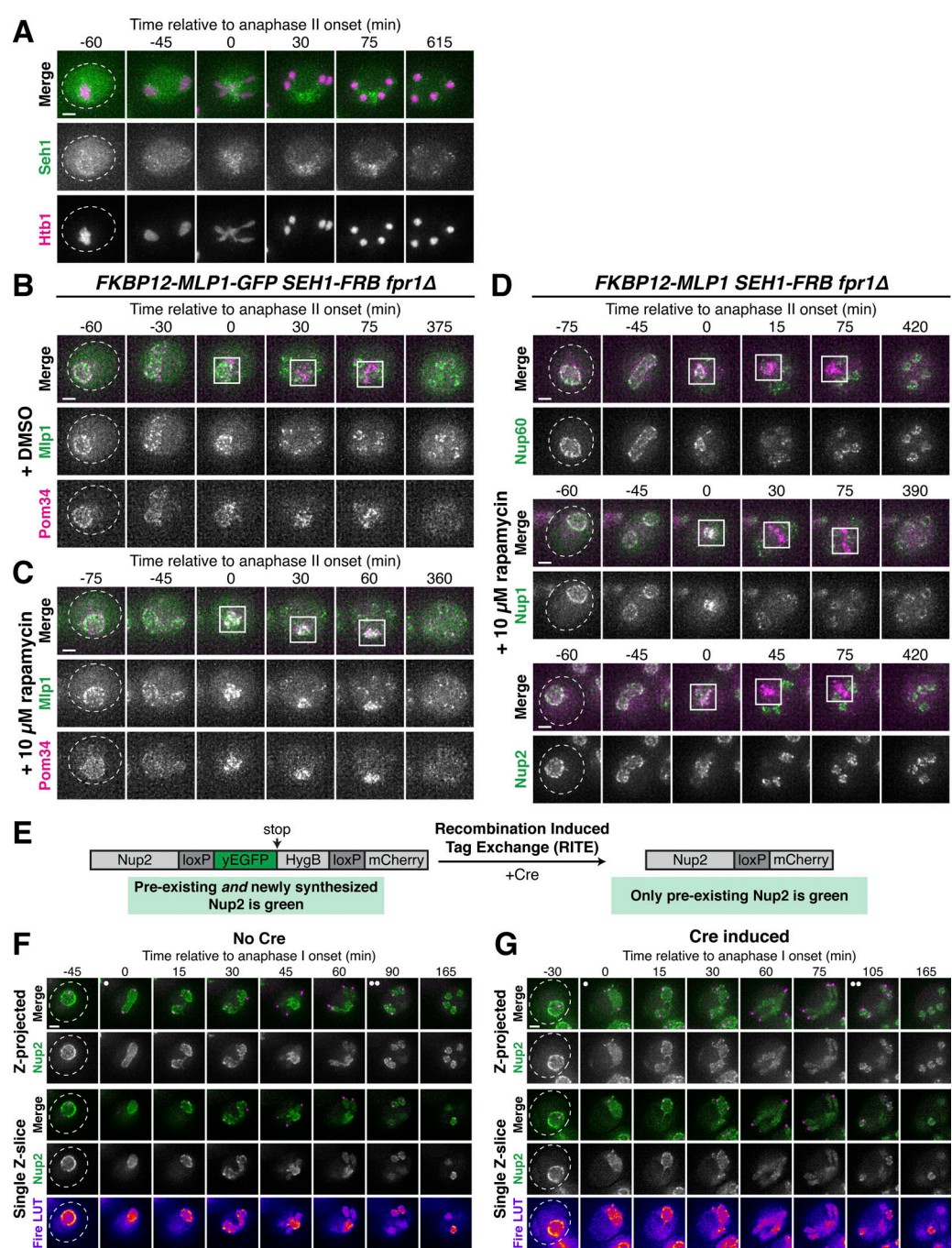

Figure S3. **Supporting data pertaining to two distinct nuclear basket remodeling events during budding yeast meiosis. (A)** Montage of a cell with Seh1-GFP, a Y-complex nucleoporin, and Htb1-mCherry, a histone, progressing through meiosis (UB24613). The onset of anaphase II was defined by the presence of four Htb1-mCherry lobes. **(B and C)** Montages of cells containing FKBP12-Mlp1-GFP and Seh1-FRB, treated with either (B) DMSO or (C) 10 μM rapamycin after 4 h in SPM (UB29337). **(D)** Montages of cells with different fluorescently tagged basket nucleoporins—Nup60-GFP (UB30174), Nup2-GFP (UB30168), and Nup1-GFP (UB30166)—and the inducible Mlp1 tether (FKBP12-Mlp1 and Seh1-FRB) treated with 10 μM rapamycin after 4 h in SPM. For B–D, the transmembrane nucleoporin Pom34-mCherry was used to monitor the core of the NPC, with the GUNC indicated by a white box. The onset of anaphase II was defined as the first time point with GUNC formation. All cells were *fpr1Δ* to facilitate rapamycin access to the tether. **(E)** A schematic depicting RITE (Verzijlbergen et al., 2010), a technique that facilitates differentiation between pre-existing and newly synthesized protein pools. Nup2 was tagged with a RITE cassette (Nup2-RITE), allowing for genetically encoded tag switching upon induction of Cre recombinase. After tag exchange, any GFP-tagged Nup2 represents a protein that was present prior to the conversion event. Tag exchange was induced with over 95% efficiency prior to meiotic induction, as assayed by loss of hygromycin resistance. Tetrad end-point analysis of Nup2-GFP signal intensity allowed individual cells to be scored for successful tag exchange by microscopy. **(F and G)** Montages of cells containing Nup2-RITE, a basket nucleoporin, and Spc42-mCherry, a spindle pole body (SPB) component, progressing through meiosis. **(F)** No Cre-EBD present, such that GFP-tagged protein represents either pre-existing or newly synthesized Nup2 (UB34454). **(G)** Cre-EBD present and induced, such that GFP-tagged protein represents only pre-existing Nup2 (UB34452). The onset of anaphase I was defined as the Nup2-GFP signal exhibiting distortion from a spherical shape consistent with chromosome segregation. Note that GFP intensity was not scaled identically in F and G, and that the selected RFP imaging settings did not allow for robust detection of newly synthesized Nup2-mCherry. Scale bars, 2 μm.

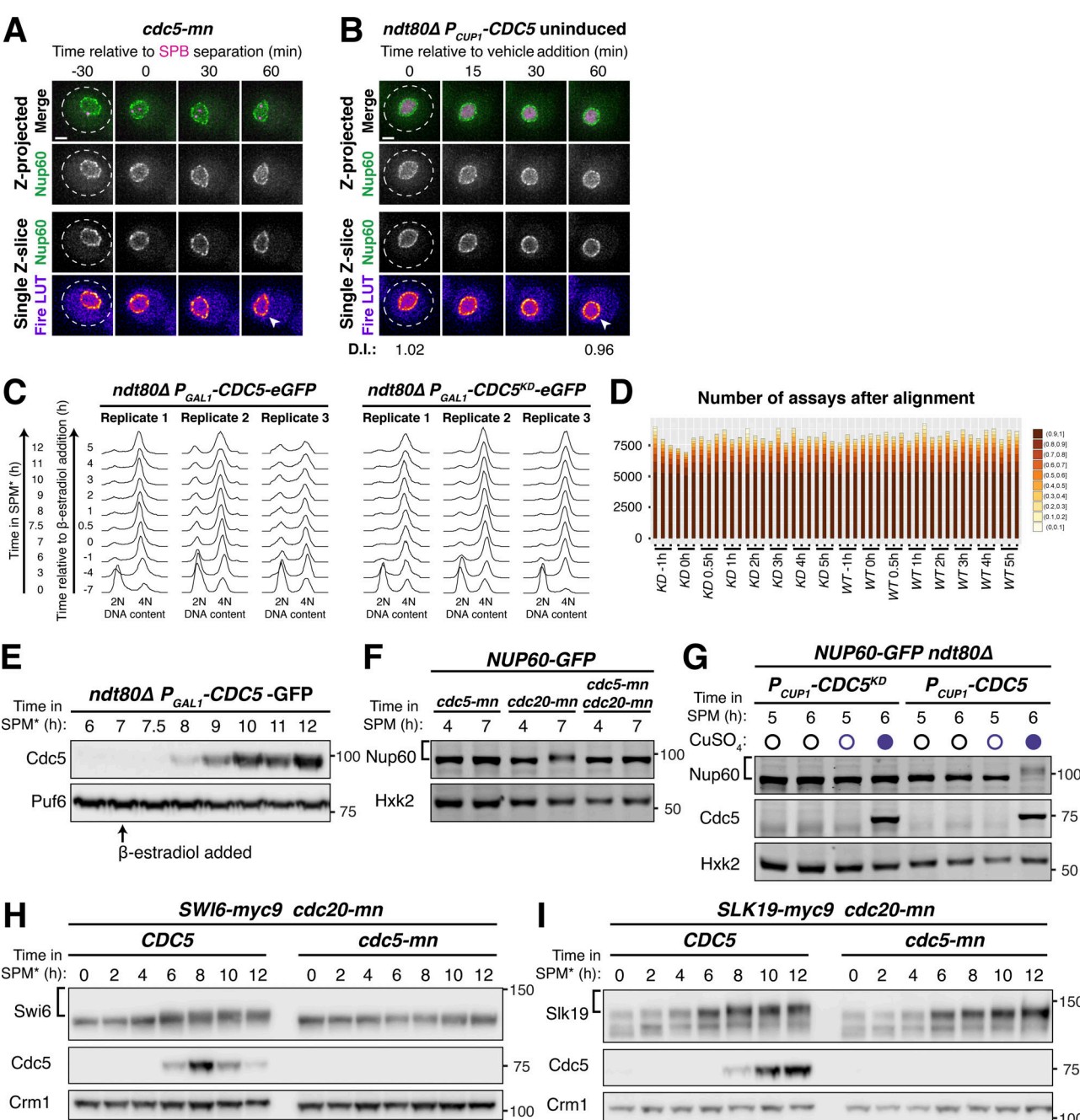

Figure S4. **Supporting data pertaining to Cdc5-dependent phosphorylation of Nup60 and other novel meiotic targets. (A)** Montage of a cell with Nup60-GFP, a nuclear basket nucleoporin, and Spc42-mCherry, a spindle pole body component, entering metaphase I arrest caused by *cdc5-mn* (UB29251). **(B)** Montage of a cells with Nup60-GFP, a nuclear basket nucleoporin, and Htb1-mCherry, a histone, in prophase I arrest (*ndt80Δ*) with P_CUP1-CDC5-3xFLAG-10xHis uninduced (UB29129). Normalized DI values are indicated when calculated (see Fig. 3 H legend for description of normalization). For A and B, the white arrowheads in the "Fire LUT" images denote nuclei exhibiting Nup60-GFP detachment or nuclei from a relevant control at an equivalent time point. Scale bars, 2 µm. **(C)** FACS profiles of DNA content of meiotic time courses used for SWATH-MS proteomics in Fig. 4, A–D (Cdc5: YML3993, Cdc5^KD: YML3994). Three independent replicates are shown for each strain. **(D)** Number of phosphopeptide precursors identified in each sample. The color scale represents the consistency of identification as a fraction over all runs. **(E)** Representative immunoblot of Cdc5 expression in cells treated as described in Fig. 4 A. Cells with *ndt80Δ* and P_GAL1-CDC5-eGFP (YML3993) were induced to enter meiosis by transfer to SPM* and, after 7 h in SPM*, treated with 2 µM β-estradiol to initiate Cdc5 expression. Puf6 was used as a loading control. **(F)** Immunoblot for Nup60-GFP before (4 h in SPM) or during (7 h in SPM) metaphase I arrest for the strains imaged in panels 3C-D and S4A (*cdc5-mn*: UB29251, *cdc20-mn*: UB29253, and *cdc5-mn cdc20-mn*: UB29249). Hxk2 was used as a loading control. **(G)** Immunoblots for Nup60-GFP and Cdc5^KD-3xFLAG-10xHis (UB29069) or Cdc5-3xFLAG-10xHis (UB29129) before (5 h in SPM) or after (6 h in SPM) treatment (either addition of copper or not) during prophase I arrest (*ndt80Δ*). The protein samples were collected from the strains imaged in 3F-G and S4B. Hxk2 was used as a loading control. **(H)** Immunoblots of Swi6-9myc and Cdc5 in *cdc20-mn* (YML8836) or *cdc20-mn cdc5-mn* (YML8837) strains induced to enter meiosis and arrest in metaphase I. Crm1 was used as the loading control. **(I)** Immunoblots for Slk19-9myc and Cdc5 in *cdc20-mn* (YML7800) or *cdc20-mn cdc5-mn* (YML7801) cells induced to enter meiosis and arrest in metaphase I. Crm1 was used as the loading control. For F–I, the brackets to the left of the blots denote apparent phosphoshifts. Source data are available for this figure: SourceData FS4.

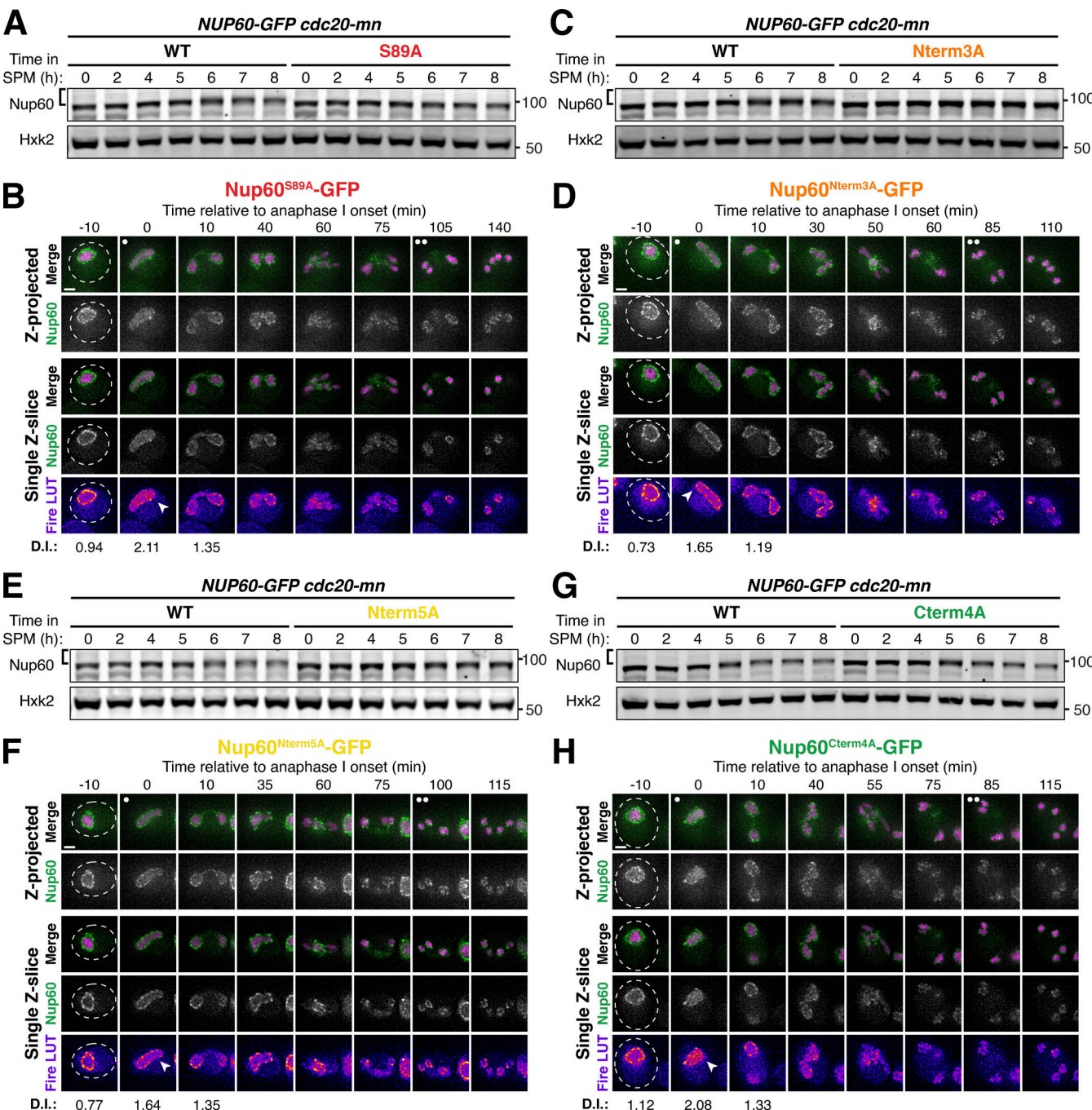

Figure S5. **Supporting data pertaining to Nup60 phosphomutant localization and phosphorylation. (A, C, E, and G)** Immunoblots of different Nup60 phosphomutants compared to Nup60-GFP (UB29253) in *cdc20-mn* background. Hxk2 was used as a loading control. The phosphomutants tested are: (A) Nup60[S89A]-GFP (UB30327); (C) Nup60[Nterm3A]-GFP (UB30329); (E) Nup60[Nterm5A]-GFP (UB30331); and (G) Nup60[Cterm4A]-GFP (UB30333). The Nup60-GFP control for A, C, and E are all reruns of the same replicate; the Nup60-GFP control for G is a rerun of the replicate from Fig. 5 G. The brackets to the left of the blots denote apparent phosphoshifts. **(B, D, F, and H)** Montages of cells with different fluorescently tagged Nup60 phosphomutants and Htb1-mCherry progressing through meiosis. The alleles visualized are: (B) Nup60[S89A]-GFP (UB29265); (D) Nup60[Nterm3A]-GFP (UB29441); (F) Nup60[Nterm5A]-GFP (UB29443); and (H) Nup60[Cterm4A]-GFP (UB29267). For B, D, F, and H, the white arrowheads in the "Fire LUT" images denote nuclei at the onset of anaphase I, the stage when Nup60-Nup2 detachment is observed. Scale bars, 2 μm. Source data are available for this figure: SourceData FS5.

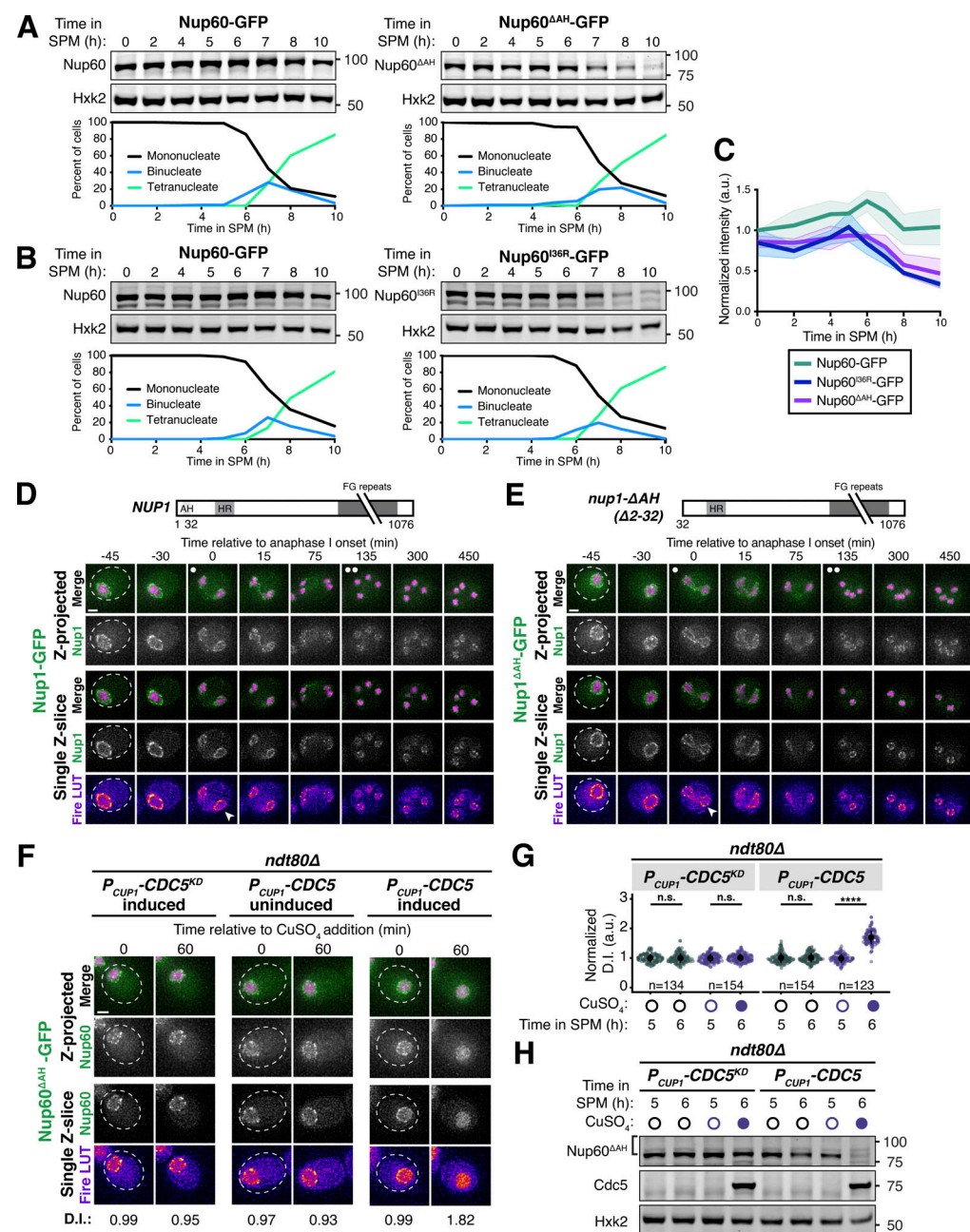

Figure S6. **Supporting data pertaining to a meiotic role for the Nup60 AH. (A)** Immunoblots of Nup60-GFP (UB14646) and Nup60$^{\Delta AH}$-GFP (UB25731) in cells progressing through meiosis. **(B)** Immunoblots of Nup60-GFP (UB14646) and Nup60$^{I36R}$-GFP (UB27189) in cells progressing through meiosis. For A and B, Hxk2 was used as a loading control and meiotic staging corresponding to the displayed blots was assessed using Htb1-mCherry. **(C)** Quantification of different Nup60 mutant protein levels during meiosis, corresponding to A and B. Background subtracted Nup60 intensity was divided by Hxk2 intensity to control for loading and then normalized to Nup60-GFP levels at 0 h in SPM (using the control run on the same blot). For Nup60-GFP, the mean ± standard deviation (shaded range) of four independent biological replicates is displayed. For Nup60$^{\Delta AH}$-GFP and Nup60$^{I36R}$-GFP, the mean ± range (shaded area) of two independent biological replicates is displayed. **(D and E)** Montages of cells with either (D) Nup1-GFP (UB15303) or (E) Nup1$^{\Delta AH}$-GFP (UB25727) and Htb1-mCherry going through meiosis. Schematics of *NUP1* and *nup1-ΔAH*, which lacks its N-terminal AH, are displayed above their respective montages. For D and E, the white arrowheads in the "Fire LUT" images denote nuclei during meiosis I, a time point when Nup60$^{\Delta AH}$ detachment is apparent. **(F)** Montages of cells with Nup60$^{\Delta AH}$-GFP and Htb1-mCherry exposed to different Cdc5 treatments in prophase I arrest (*ndt80Δ*): $P_{CUP1}$-*CDC5$^{KD}$-3xFLAG-10xHis* induced (UB29073), $P_{CUP1}$-*CDC5-3xFLAG-10xHis* uninduced (UB29071), and $P_{CUP1}$-*CDC5-3xFLAG-10xHis* induced (UB29071). Induction of Cdc5 protein was performed at 5 h in SPM with 50 μM CuSO$_4$. **(G)** Quantification of Nup60$^{\Delta AH}$ detachment for the experiment depicted in F. Individual DI values were normalized to the average DI for uninduced $P_{CUP1}$-*CDC5$^{KD}$-3xFLAG-10xHis* cells at the pre-induction time point (5 h). Asterisks indicate statistical significance calculated using a Wilcoxon signed-rank test when post-induction (6 h in SPM) values were compared to preinduction (5 h in SPM) values for each treatment regimen (see Table S5 for P values). Sample sizes (*n*) are the number of cells quantified for each treatment regimen. **(H)** Immunoblots for Nup60$^{\Delta AH}$-GFP and Cdc5$^{KD}$-3xFLAG-10xHis (UB29073) or Cdc5-3xFLAG-10xHis (UB29071) before (5 h in SPM) or after (6 h in SPM) treatment (either addition of copper or not) during prophase I arrest (*ndt80Δ*), corresponding to the images in F. Hxk2 was used as a loading control. The bracket to the left of the blot denotes the apparent phosphoshift. For each montage, normalized DI values are indicated when calculated. Scale bars, 2 μm. Source data are available for this figure: SourceData FS6.

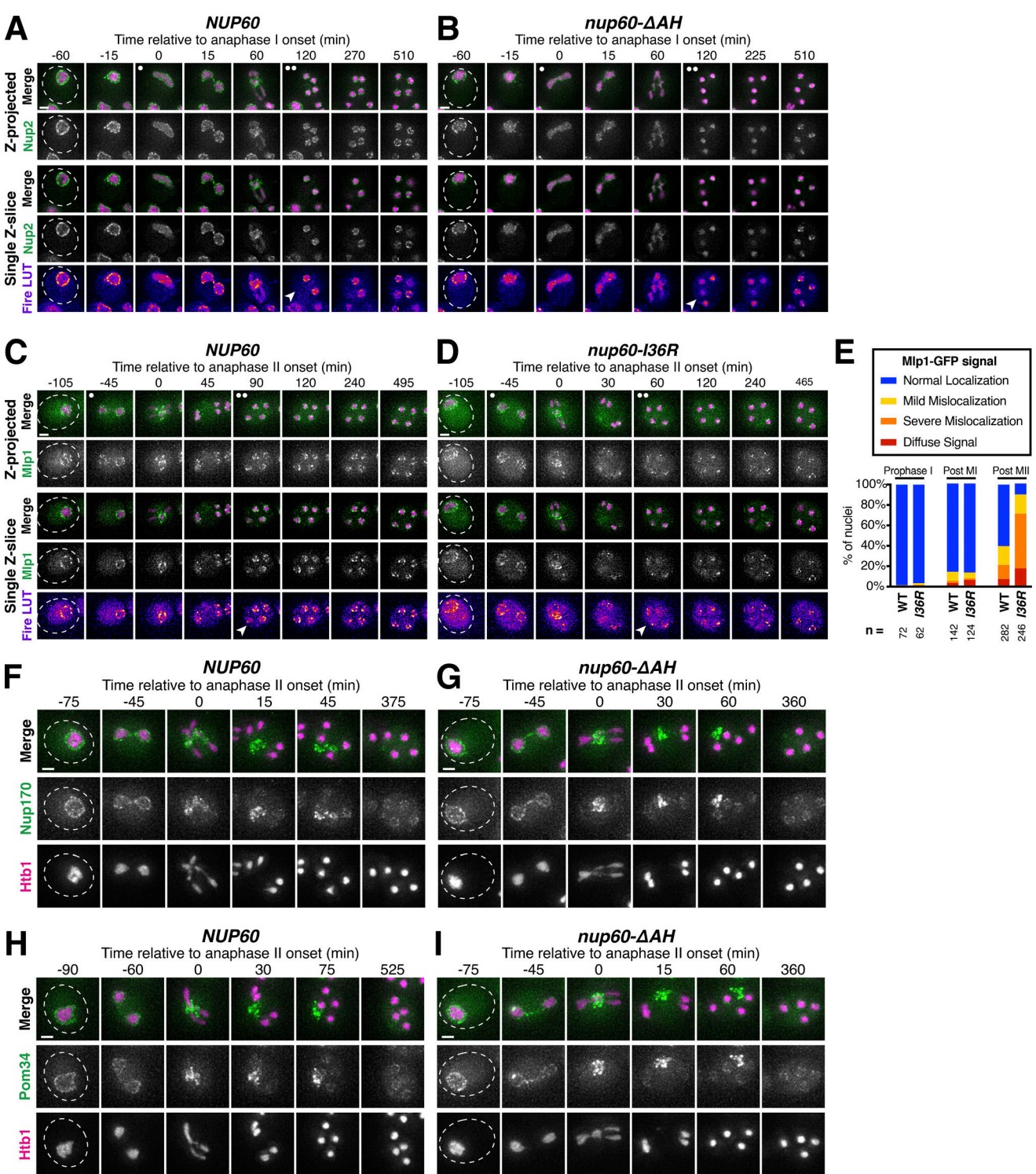

Figure S7. **Supporting data pertaining to the role of Nup60 in gamete nuclear basket organization. (A and B)** Montages of cells with Nup2-GFP, a nuclear basket nucleoporin, and Htb1-mCherry, a histone, progressing through meiosis in either (A) a *NUP60* (UB15305) or (B) a *nup60-ΔAH* (UB30630) genetic background. **(C and D)** Montages of cells with Mlp1-GFP, a nuclear basket nucleoporin, and Htb1-mCherry, a histone, progressing through meiosis in either (C) a *NUP60* (UB14648) or (D) a *nup60-I36R* (UB30640) genetic background. **(E)** Quantification of Mlp1-GFP organization for the strains in C and D during prophase I (defined as an hour before the post-MI time point), post-MI (defined as the presence of two clear Htb1-mCherry lobes), and post-MII (defined as 2 h after the presence of four clear Htb1-mCherry lobes). Sample sizes (*n*) indicate the number of nuclei scored for Mlp1 organization. For A–D, the white arrowheads in the "Fire LUT" images denote cells after anaphase II when gamete nuclear basket organization or misorganization is apparent. **(F and G)** Montages of cells with Nup170-GFP, a core nucleoporin, and Htb1-mCherry, a histone, progressing through meiosis in either (F) a *NUP60* (UB11513) or (G) a *nup60-ΔAH* (UB34088) genetic background. **(H and I)** Montages of cells with Pom34-GFP, a core nucleoporin, and Htb1-mCherry, a histone, progressing through meiosis in either (H) a *NUP60* (UB13503) or (I) a *nup60-ΔAH* (UB34086) genetic background. For all panels, the onset of anaphase II was defined by the presence of four Htb1-mCherry lobes. Scale bars, 2 µm.

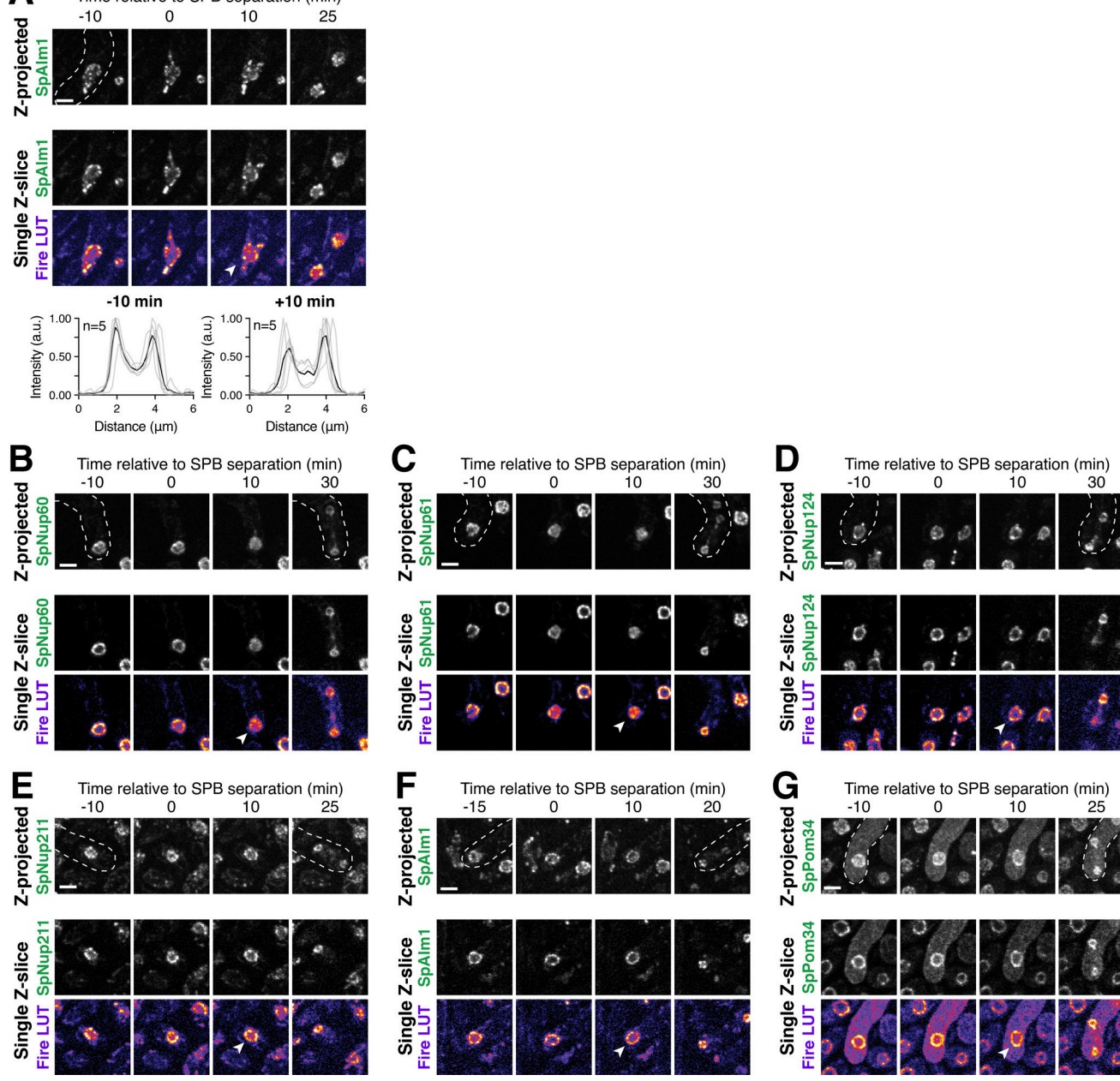

Figure S8. **Supporting data pertaining to nuclear basket behavior during *S. pombe* meiosis. (A)** Maximum intensity projections (top row) and single z-slice image montages of cells with SpAlm1-GFP, a putative Tpr-like nucleoporin with no apparent *S. cerevisiae* ortholog (fySLJ842 × fySLJ479), progressing through meiosis I and staged according to the timing of meiosis I SPB separation as visualized using SpPpc89-mCherry (not shown). The individual (gray) and mean (black) SpAlm1-GFP intensity profiles measured on single z-slices are shown below for the indicated number of individual nuclei (*n*), 10 min before and after SPB separation. The white arrowhead in the "Fire LUT" images denote a nucleus after meiosis I SPB separation, at the stage when SpNup60-SpNup61 detachment is observed. **(B–G)** Montages of strains with the various GFP-tagged nucleoporins during meiosis II: (B) SpNup60-GFP (fySLJ730 × fySLJ479); (C) SpNup61-GFP (fySLJ840 × fySLJ479); (D) SpNup124-GFP (fySLJ456 × fySLJ989); (E) SpNup211-GFP (fySLJ456 × fySLJ990); (F) SpAlm1-GFP (fySLJ842 × fySLJ479); and (G) SpPom34-GFP (fySLJ1242 × fySLJ1243). Cells were staged according to the timing of meiosis II SPB separation as visualized using SpPpc89-mCherry. For B–G, the white arrowheads in the "Fire LUT" images denote nuclei after meiosis II SPB separation, at the time point when the second SpNup60-SpNup61 detachment occurs. Scale bars, 3 μm.

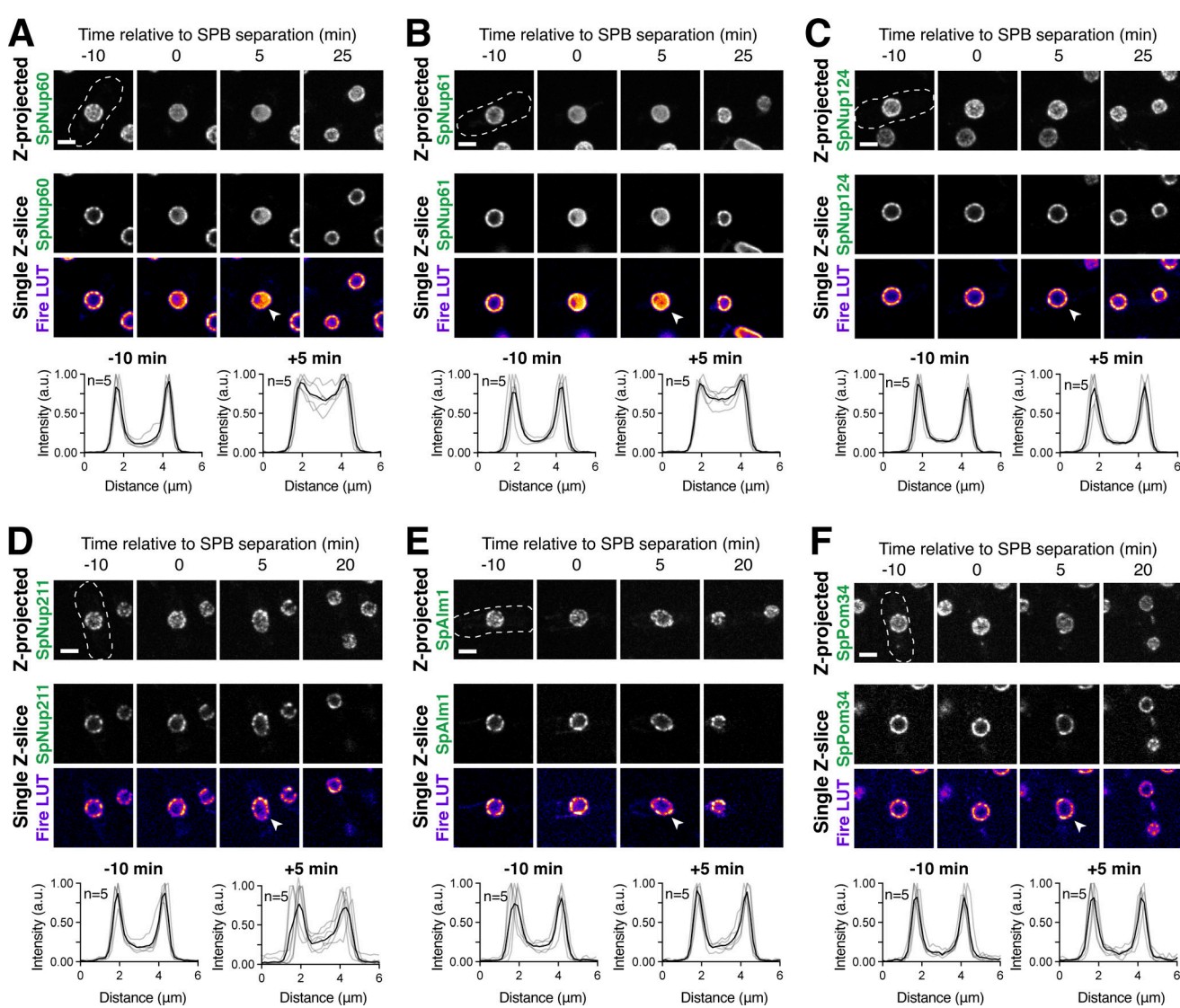

Figure S9. **Supporting data pertaining to nuclear basket behavior during *S. pombe* mitosis. (A–F)** Montages of strains with GFP-tagged nucleoporins during mitosis: (A) SpNup60-GFP (fySLJ745); (B) SpNup61-GFP (fySLJ870); (C) SpNup124-GFP (fySLJ1018); (D) SpNup211-GFP (fySLJ1019); (E) SpAlm1-GFP (fySLJ867); and (F) SpPom34-GFP (fySLJ537). Cells were staged according to the timing of SPB separation as visualized using SpPpc89-mCherry (not shown). The white arrowheads in the "Fire LUT" images denote nuclei after mitotic SPB separation, at the time point when SpNup60-SpNup61 detachment occurs. The individual (gray) and mean (black) nucleoporin intensity profiles measured on single *z*-slices are shown below for the indicated number of individual nuclei (*n*), 10 min before and 5 min after SPB separation. Scale bars, 3 μm.

Video 1.   **The cell depicted in Fig. 1 D undergoing meiosis.** The cell has Nup60-GFP, a nuclear basket nucleoporin, and Htb1-mCherry, a histone (UB14646). The movie is four frames per second, with images acquired every 5 min. Maximum intensity projections over 8 μm are shown. Scale bar, 2 μm.

Video 2.   **The cell depicted in Fig. S1 B undergoing meiosis.** The cell has Nup2-GFP (UB15305), a nuclear basket nucleoporin, and Htb1-mCherry, a histone. The movie is four frames per second, with images acquired every 5 min. Maximum intensity projections over 8 μm are shown. Scale bar, 2 μm.

Video 3.   **The cell depicted in Fig. S1 C undergoing meiosis.** The cell has Nup1-GFP, a nuclear basket nucleoporin, and Htb1-mCherry, a histone (UB15303). The movie is four frames per second, with images acquired every 5 min. Maximum intensity projections over 8 μm are shown. Scale bar, 2 μm.

Video 4.   **The cell depicted in** Fig. 1 E **undergoing meiosis.** The cell has Mlp1-GFP, a nuclear basket nucleoporin, and Htb1-mCherry, a histone (UB14648). The movie is four frames per second, with images acquired every 5 min. Maximum intensity projections over 8 µm are shown. Scale bar, 2 µm.

Video 5.   **The cell depicted in** Fig. 1 F **undergoing meiosis.** The cell has Pom34-GFP, a transmembrane nucleoporin, and Htb1-mCherry, a histone (UB13503). The movie is four frames per second, with images acquired every 5 min. Maximum intensity projections over 8 µm are shown. Scale bar, 2 µm.

Video 6.   **The cell depicted in** Fig. 6 B **undergoing meiosis.** The cell has Nup60$^{9A}$-GFP, a phosphomutant, and Htb1-mCherry, a histone (UB29358). The movie is four frames per second, with images acquired every 5 min. Maximum intensity projections over 8 µm are shown. Scale bar, 2 µm.

Video 7.   **The cell depicted in** Fig. 7 C **undergoing meiosis.** The cell has Nup60$^{\Delta AH}$-GFP, a mutant of the lipid-binding N-terminus, and Htb1-mCherry, a histone (UB25731). The movie is four frames per second, with images acquired every 15 min. Maximum intensity projections over 8 µm are shown. Scale bar, 2 µm.

Video 8.   **The cell depicted in** Fig. 7 D **undergoing meiosis.** The cell has Nup60$^{I36R}$-GFP, a mutant of the lipid-binding N-terminus, and Htb1-mCherry, a histone (UB27189). The movie is four frames per second, with images acquired every 15 min. Maximum intensity projections over 8 µm are shown. Scale bar, 2 µm.

Video 9.   **The cell depicted in** Fig. 8 E **undergoing meiosis.** The cell has Mlp1-GFP, a nuclear basket nucleoporin, and Htb1-mCherry, a histone, in a *nup60-ΔAH* mutant background (UB30632). The movie is four frames per second, with images acquired every 5 min. Maximum intensity projections over 8 µm are shown. Scale bar, 2 µm.

Video 10.   **The *S. pombe* cell depicted in 10B undergoing meiosis.** The cell has SpNup60-GFP, the ortholog of ScNup60, and SpPpc89-mCherry, a spindle pole body component (fySLJ730 × fySLJ479). The movie is four frames per second, with images acquired every 5 min. Maximum intensity projections over 8 µm are shown. Scale bar, 3 µm.

**Provided online are Table S1, Table S2, Table S3, Table S4, Table S5, Table S6, Table S7, and Table S8. Table S1 lists the strains used in this study. Table S2 lists the plasmids used in this study. Table S3 shows the primers used for deletion and C-terminal tagging in this study. Table S4 shows the imaging conditions used in this study. Table S5 contains information about the statistics and significance values used in the figures. Table S6 contains the SWATH-MS data used to identify phosphopeptides that accumulate in response to Cdc5 activity, including protein identifier, modified site, log$_2$ fold change, and P values. Table S7 lists sporulation efficiency for various alleles in this study. Table S8 lists gamete viability for various alleles in this study.**

