## [Peer Review File · The Journal of Cell Biology]

Meiotic Nuclear Pore Complex Remodeling Provides Key Insights into Nuclear Basket Organization

Grant King, Rahel Wettstein, Joseph Varberg, Keerthana Chetlapalli, Madison Walsh, Ludovic Gillet, Claudia Hernández-Armenta, Pedro Beltrao, Ruedi Aebersold, Sue Jaspersen, Joao Matos, and Elcin Unal

Corresponding Author(s): Elcin Unal, University of California, Berkeley and Joao Matos, University of Vienna

Review Timeline:

Submission Date:	2022-04-15
Editorial Decision:	2022-05-24
Revision Received:	2022-09-12
Editorial Decision:	2022-10-11
Revision Received:	2022-11-02

Monitoring Editor: Martin Hetzer

Scientific Editor: Dan Simon

Transaction Report:

DOI: <https://doi.org/10.1083/jcb.202204039>

May 24, 2022

Re: JCB manuscript #202204039

Dr. Elcin Unal
University of California, Berkeley
16 Barker Hall Rm 622
Berkeley, CA 94720

Dear Dr. Unal,

Thank you for submitting your manuscript "Meiotic Nuclear Pore Complex Remodeling Provides Key Insights into Nuclear Basket Organization." The manuscript was assessed by expert reviewers, whose comments are appended to this letter. We invite you to submit a revision if you can address the reviewers' key concerns, as outlined here.

You will see that the reviewers are highly enthusiastic about your study, which provides a substantial advance in our understanding of NPC remodeling during meiosis. There are however a few conclusions that the reviewers feel are not as strongly supported by the data and so they ask that you either tone these down or provide additional data. We feel that these are important comments that should be answered and we encourage you to address these with new experiments if this is possible within a reasonable timeframe.

GENERAL GUIDELINES:

Text limits: Character count for an Article is < 40,000, not including spaces. Count includes title page, abstract, introduction, results, discussion, and acknowledgments. Count does not include materials and methods, figure legends, references, tables, or supplemental legends.

Figures: Articles may have up to 10 main text figures. Figures must be prepared according to the policies outlined in our Instructions to Authors, under Data Presentation, <https://jcb.rupress.org/site/misc/ifora.xhtml>. All figures in accepted manuscripts will be screened prior to publication.

Supplemental information: There are strict limits on the allowable amount of supplemental data. Articles may have up to 5 supplemental figures. Up to 10 supplemental videos or flash animations are allowed. A summary of all supplemental material should appear at the end of the Materials and methods section.

Furthermore, as noted in prior communication with Tim Spencer, JCB requires that all proteomic datasets be deposited in a public repository by the time a paper would be published.

Source Data:

Please note that JCB now requires authors to submit Source Data used to generate figures containing gels and Western blots with all revised manuscripts. This Source Data consists of fully uncropped and unprocessed images for each gel/blot displayed in the main and supplemental figures. Since your paper includes cropped gel and/or blot images, please be sure to provide one Source Data file for each figure that contains gels and/or blots along with your revised manuscript files. File names for Source Data figures should be alphanumeric without any spaces or special characters (i.e., SourceDataF#, where F# refers to the associated main figure number or SourceDataFS# for those associated with Supplementary figures). The lanes of the gels/blots should be labeled as they are in the associated figure, the place where cropping was applied should be marked (with a box), and molecular weight/size standards should be labeled wherever possible. Source Data files will be made available to reviewers during evaluation of revised manuscripts and, if your paper is eventually published in JCB, the files will be directly linked to specific figures in the published article.

The typical timeframe for revisions is three to four months. While most universities and institutes have reopened labs and allowed researchers to begin working at nearly pre-pandemic levels, we at JCB realize that the lingering effects of the COVID-19 pandemic may still be impacting some aspects of your work, including the acquisition of equipment and reagents. Therefore, if you anticipate any difficulties in meeting this aforementioned revision time limit, please contact us and we can work with you to find an appropriate time frame for resubmission. Please note that papers are generally considered through only one revision cycle, so any revised manuscript will likely be either accepted or rejected.

Thank you for this interesting contribution to Journal of Cell Biology. You can contact us at the journal office with any questions, cellbio@rockefeller.edu or call (212) 327-8588.

Sincerely,

Martin Hetzer, PhD
Monitoring Editor
Journal of Cell Biology

Dan Simon, PhD
Scientific Editor
Journal of Cell Biology

Reviewer #1 (Comments to the Authors (Required)):

In this manuscript, King and Wettstein characterize the remodeling of the nuclear pore complex during meiosis in yeast. They find that nuclear basket proteins transiently dissociate during meiosis 1 and 2 in *S. cerevisiae*. In meiosis 1, they discover that phosphorylation of Nup60 by the Polo-kinase Cdc5 is critical for the observed detachment. However, NPC remodeling in meiosis 2 appears to be mechanistically distinct. Interestingly, this NPC remodeling is evolutionarily conserved and can also be seen in fission yeast.

This manuscript contains a large number of interesting results that should be of great interest to the NPC community and to people studying meiosis. Overall, the data are of very high quality and I have no major technical concerns. The manuscript could be improved if the authors addressed the following two points.

1. Functional significance. The function of the NPC remodeling events in meiosis is not addressed and this would also go beyond the scope of the current paper. However, did the authors observe any obvious phenotypes in the Nup60 phosphomutants or in the FKBP12-FRP tethering experiments during meiosis or germination? If no phenotypes are observed this should be discussed.
2. Cdc5 functions also during the mitotic cell cycle. Is there any indication that the nuclear basket transiently dissociates during mitosis (or in mitotic cell cycle mutants). Does overexpression of Cdc5 in the mitotic cell cycle lead to Nup60 detachment? Alternatively, does artificial nuclear accumulation of Cdc5 promote basket detachment in mitosis (Cdc5-NLS; see Botchkarev et al, Cell Cycle 2014)?

Reviewer #2 (Comments to the Authors (Required)):

King, Wettstein, and colleagues use live cell imaging and proteomics to demonstrate that components of the nuclear basket undergo transient, Cdc5 kinase-dependent remodeling during meiosis I and II in budding and fission yeast. The experiments are well-designed and make use of state-of-the-art tools for functional dissection of proteins such as auxin-inducible degrons, the "anchor-away" system, and SWATH-MS for sensitive PTM identification. With these sensitive tools, the authors demonstrate that Nup60 is a target of the Cdc5 kinase in meiosis I; that this phosphorylation regulates NPC detachment; and that reattachment requires an amphipathic helix in Nup60. Interestingly, the authors show that Nup60 is required for the initial recruitment of Nup2 and Mlp1 to the NPC, but that Mlp1 can be retained at the NPC through meiosis without Nup60.

The data are rigorous and self-consistent. However, in my opinion, ambiguity remains over whether meiotic NPC remodeling reflects (i) detachment and reattachment of nuclear basket proteins to intact NPCs, (ii) rejuvenation of preexisting NPCs by regulated degradation and replacement with newly synthesized nuclear basket proteins or (iii) a wave of de novo NPC assembly that is initiated by Nup60.

It seems that the authors favor model (i), as they state that "the entire nuclear basket is inherited: it detaches from the NPC core and returns to nascent gamete nuclei" (line 73). However, it is alternatively possible that the basket Nups are degraded (in the GUNC or elsewhere) and then that newly synthesized basket Nups are recruited to the NE after meiosis II. Indeed, data from a previous manuscript by this group (King et al. 2019) and in this manuscript do show more transient basket Nup signal at the GUNC vs. more sustained core Nup signal there; this could be explained either by lesser targeting of basket Nups to the GUNC, as the authors propose, or by faster degradative flux of the basket Nups vs. the core Nups within the GUNC. The authors could resolve this ambiguity either by further experiments to evaluate whether new protein synthesis is required for re-enrichment of basket Nups after meiosis II, or could alternatively modify their interpretations in the text. Specifically, stating that the nuclear basket is "inherited" or "reattached" after meiosis is not directly supported by the data. This is an important distinction: are (parts of) NPCs inherited transgenerationally, or are gamete NPCs "rejuvenated"? Either outcome is important and interesting, but care should be taken not to over-state the current understanding.

Specific points

1. Nature of mislocalization of basket Nups: the manuscript describes and quantifies a displacement index of Nups into the nucleoplasm in meiosis I. Given the short timescale (~10 minutes) and short distance (nucleoplasm vs. nuclear periphery) the interpretation that these proteins transiently dissociate and then reassociate with the NPC seems reasonable in meiosis I. However, it seems clear from images that basket Nups can also be seen at a centrally located focus that resembles the GUNC in meiosis II (for example Fig. 1D,E). Given previous work (King et al 2019) indicating that the GUNC is a degradative compartment, it seems possible that the basket Nups are being degraded in the GUNC during meiosis II, and that the re-enrichment to NPCs after meiosis II could reflect newly synthesized protein.
2. Nup60 and its mammalian homolog Nup153 each have an amphipathic helix that is required for assembly of NPCs into the intact NE (Meszaros et al., Dev Cell 2015; Vollmer et al., Dev Cell 2015). The requirement for Nup60's amphipathic helix in re-enrichment of basket Nups to the nuclear periphery is suggestive of Nup60 engaging directly with the NE membrane to promote assembly of new NPCs. The authors state that the amphipathic helix mediates NPC re-association, but it seems equivalently possible that the protein is engaging directly with the membrane. Also, NPCs appear clustered in the presence of Nup60 amphipathic helix mutants after meiosis II (Figure 7C and 7D, last frames), which could indicate an NPC assembly defect. Finally, the fact that many core Nups are sequestered or degraded in the GUNC suggests that new NPC assembly may be required after meiosis. Have the authors considered that Nup60 might promote de novo NPC assembly after meiosis II? EM could reveal the presence of NPC assembly intermediates.

Reviewer #3 (Comments to the Authors (Required)):

During mitosis, various nuclear pore complex proteins (Nups) have been shown to undergo posttranslational modifications, most notably phosphorylation. In metazoan cells, these modifications have been linked to NPC disassembly. In yeast, while a few Nups have been shown to be phosphorylated during mitosis, little is known about the extent of mitotic Nup phosphorylation nor what impact these modifications have on NPC structure and function during this stage of the cell cycle. In this manuscript, the authors have examined the assembly state of NPCs during meiosis I and II of budding yeast. They present results which lead them to conclude that nuclear basket structures of the NPC dissociate from NPC core structures during meiosis I and II. During Meiosis I they show that Nup60 and Nup2 dissociate from nuclear envelope (NE), and this event is driven by Polo kinase Cdc5 through its phosphorylation of Nup60. In support of this conclusion, the authors have identified phosphorylation sites in Nup60 that are required for its release from the NE during meiosis I. In addition, the authors propose that reassembly of Nup60-Nup2 and its binding to the NPC core following meiosis is mediated by a lipid-binding amphipathic helix domain of Nup60.

The authors present a large amount of high quality data throughout the manuscript. Clear strengths of the paper are data characterizing changes in the interactions of Nup60 and binding partners with the NE, the identification of Cdc5 as a regulator of Nup60 phosphorylation, and the analysis of the role of Nup60 phosphorylation in NPC disassembly through the analysis of nup60 mutants. These provides several new and interesting insights into the structural changes that occur in the NPC during nuclear division in yeast, specifically as it relates to the NPC basket proteins. For the most part, the data presented in the manuscript sufficiently support the authors' conclusions, but there are a few instances where this was not the case, and revision are required. Specific issues are outlined below.

Recommendations for the authors

1) On the basis of their Nup60-degron experiments and the analysis of cells with reduced Nup60, the authors conclude that (line 472) "Mlp1 can ... remain associated with the NPC core independently of Nup60 after its initial recruitment." While this is possible, this conclusion is not well supported. First, this is a negative result, i.e. the absent of any effect on Mlp1 localization upon Nup60 depletion does not imply Nup60 has no role in Mlp1 association with the NPC. There are various reasons why a Mlp1 mislocalization phenotype is absent upon degron-induced reduction of Nup60. Perhaps fragments of Nup60 remain (or are protected) following degradation that support Mlp1 association with the NPC. In this regard, the degree to which Nup60 is degraded in these experiments is not clear as only the full length protein is evaluated. Even some full length Nup60 appears to

still be present following the induction of depletion (Fig.9). This possible scenario should be discussed, and the authors' interpretations adjusted accordingly.

2) The cause of reduced binding of the nup60- Δ AH and nup60-I36R constructs to the NE/NPCs during meiotic divisions is unclear as these proteins show reduced stability. The authors conclude that the reduced NE association of these AH mutants is caused by membrane binding defects associated with the loss of the AH, and, because of this reduced NPC association, the AH mutants are unstable and degraded. However, it is just as likely that the nup60 mutants fail to show detectable association with the NE because they are unstable and degraded during meiosis and prior to their reassociation with the reforming NPCs. Hence, the authors cannot solely conclude that the AH is required for binding to NPCs. The authors need to alter the text to reflect this possibility. As the manuscript stands, there is no evidence that the AH plays a direct role in attachment Nup60 to reforming NPCs.

3) It remains unclear whether Cdc5 directly phosphorylates Nup60 or it is indirectly required for Nup60 phosphorylation. The authors should word their conclusions accordingly.

4) The authors state on line 227 that "Nup60-GFP remained associated with the nuclear periphery in *cdc5-mn cdc20-mn* cells (Figure 3D-E). Similar results were obtained using the *cdc5-mn* allele alone (Figure S3A)." The latter data are also shown in Fig.3E, and here the figure indicates a statistically significant difference between the 4 and 8 hr point of in the *cdc5-mn* allele. Is this correct? Please clarify. Perhaps I missed something here.

Reviewer #1 (Comments to the Authors (Required)):

1. Functional significance. The function of the NPC remodeling events in meiosis is not addressed and this would also go beyond the scope of the current paper. However, did the authors observe any obvious phenotypes in the Nup60 phosphomutants or in the FKBP12-FRP tethering experiments during meiosis or germination? If no phenotypes are observed this should be discussed.

Sporulation efficiency and gamete viability are not impaired in strains with the *NUP60* alleles used in this study. This finding is neither surprising nor uncommon given that these two readouts primarily assess functions related to chromosome segregation, ascospore formation, and germination. Furthermore, even if the NPC remodeling events that we've uncovered affect one or more of these functions, redundant mechanisms might be in place and mask potential phenotypes. For example, although *nup2Δ* cells don't exhibit a sporulation efficiency defect, combining this mutation with defects in telomere bouquet formation (*ndj1Δ* or *csm4Δ*) results in almost no sporulation (Chu et al., 2017). As the reviewer pointed out, full investigation of biological significance goes beyond the scope of the current paper.

To address the reviewer's request, we have added two supplemental tables summarizing the sporulation efficiency and gamete viability data (Table S7-S8), included a corresponding statement in the discussion section (line 616-621), and added relevant strain genotypes in Table S1. As part of the discussion, we have also highlighted some possible biological roles of NPC remodeling, which we are currently investigating.

2. Cdc5 functions also during the mitotic cell cycle. Is there any indication that the nuclear basket transiently dissociates during mitosis (or in mitotic cell cycle mutants)? Does overexpression of Cdc5 in the mitotic cell cycle lead to Nup60 detachment? Alternatively, does artificial nuclear accumulation of Cdc5 promote basket detachment in mitosis (Cdc5-NLS; see Botchkarev et al, Cell Cycle 2014)?

We have included additional data on the mitotic behavior of basket nucleoporins in both *S. cerevisiae* and *S. pombe* (Supplemental Figures 2 and 9). In *S. cerevisiae*, minor detachment of Nup60 and Nup2 is observed at the onset of mitotic anaphase, consistent with increased Cdc5 activity during this time. However, the detachment occurs to a far lesser extent than in the meiotic context. This could be due to meiosis-specific regulation of Cdc5 or differences in cell cycle length that affect the level and/or duration of Cdc5 activity (e.g., G2/M is longer in meiosis than in mitosis in budding yeast). Notably, both SpNup60 and SpNup61 robustly detach at the onset of mitotic anaphase in *S. pombe*, which has a longer G2/M than *S. cerevisiae* during the mitotic cell cycle. We have added these new results in several locations throughout the manuscript: lines 135-137, lines 503-505, lines 794-799, lines 816-823, and lines 857-871.

Future work can help determine whether mitotic overexpression of Polo kinase leads to Nup60 detachment in budding yeast, and whether Polo kinase plays a role in the mitotic and meiotic NPC remodeling events in *S. pombe*.

Reviewer #2 (Comments to the Authors (Required)):

1. Nature of mislocalization of basket Nups: the manuscript describes and quantifies a displacement index of Nups into the nucleoplasm in meiosis I. Given the short timescale (~10 minutes) and short distance (nucleoplasm vs. nuclear periphery) the interpretation that these proteins transiently dissociate and then reassociate with the NPC seems reasonable in meiosis I. However, it seems clear from images that basket Nups can also be seen at a centrally located focus that resembles the GUNC in meiosis II (for example Fig. 1D,E). Given previous work (King et al 2019) indicating that the GUNC is a degradative compartment, it seems possible that the basket Nups are being degraded in the GUNC during meiosis II, and that the re-enrichment to NPCs after meiosis II could reflect newly synthesized protein.

To address the reviewer's point, we constructed an allele of *NUP2* tagged with a Recombination Inducible Tag Exchange (RITE) cassette, which allows swapping of fluorescence tags at genomic loci using an inducible Cre recombinase (Figure S3E; Verzijlbergen et al., 2010). Accordingly, pre-existing and newly synthesized pools of a protein of interest can be distinguished from one another. Using this method, we were able to confirm that, during both anaphase I and anaphase II, the same pool of Nup2 detaches from and returns to the nuclear periphery (Figure S3F-G). This is consistent with the NPC remodeling events described in our manuscript (Figure 2G). The text has been revised to include these new results: lines 197-200, lines 719-723, and lines 758-778. New strains, plasmids, primers, and imaging conditions have also been added to Table S1-S4.

We further note that GUNC-targeted proteins are degraded upon vacuolar membrane permeabilization, which releases vacuolar proteases to the cytoplasmic milieu (King et al., 2019). Vacuolar membrane permeabilization occurs several hours following anaphase II (Eastwood et al., 2012), so this timing is not consistent with the NPC remodeling events reported here.

2. Nup60 and its mammalian homolog Nup153 each have an amphipathic helix that is required for assembly of NPCs into the intact NE (Meszaros et al., Dev Cell 2015; Vollmer et al., Dev Cell 2015). The requirement for Nup60's amphipathic helix in re-enrichment of basket Nups to the nuclear periphery is suggestive of Nup60 engaging directly with the NE membrane to promote assembly of new NPCs. The authors state that the amphipathic helix mediates NPC re-association, but it seems equivalently possible that the protein is engaging directly with the membrane. Also, NPCs appear clustered in the presence of Nup60 amphipathic helix mutants after meiosis II (Figure 7C and 7D, last frames), which could indicate an NPC assembly defect. Finally, the fact that many core Nups are sequestered or degraded in the GUNC suggests that new NPC assembly may be required after meiosis. Have the authors considered that Nup60

might promote *de novo* NPC assembly after meiosis II? EM could reveal the presence of NPC assembly intermediates.

We agree with the reviewer that the amphipathic helix (AH) in Nup60 may mediate association with the nuclear periphery via direct binding to the nuclear envelope, especially during anaphase II when a large portion of NPCs are targeted to the GUNC. We maintain that the AH does promote association with individual NPCs, either directly or indirectly, as the AH is required for Nup60 localization to the nuclear periphery after anaphase I and Nup60 associates with individual NPCs at this time as seen by super-resolution microscopy (Figure 2A). We have updated the text to make more explicit the possibility that the AH functions by interacting with the nuclear envelope (lines 396-399).

To begin addressing whether Nup60 might contribute to *de novo* NPC assembly, we monitored the behavior of two core nucleoporins – Nup170-GFP, an inner ring nucleoporin, and Pom34-GFP, a transmembrane nucleoporin – in a *nup60-ΔAH* background (Supplemental Figure 7F-I). We found that core nucleoporins were sequestered to the GUNC as in wild type cells; moreover, we did not observe gross perturbations in NPC distribution around gamete nuclei (e.g., clustering as is observed in some NPC assembly mutants [Webster et al., 2014]). We updated the results to include these data in lines 435-439. New strains were also added to Table S1. We agree with the reviewer that higher resolution techniques – such as electron microscopy or super-resolution microscopy – will be very helpful in future studies to detect NPC assembly intermediates or more subtle defects. Interestingly, we observed that core nucleoporins exhibit abnormal localization during meiosis I in *nup60-ΔAH* cells, often accumulating in clusters that subsequently disperse (see Supplemental Figure 7I for an example). The composition and formation of these NPC clusters requires further characterization; as such, we have opted not to include discussion of these results in the manuscript and are instead currently following up with additional experiments.

Reviewer #3 (Comments to the Authors (Required)):

Recommendations for the authors

1) On the basis of their Nup60-degron experiments and the analysis of cells with reduced Nup60, the authors conclude that (line 472) "Mlp1 can ... remain associated with the NPC core independently of Nup60 after its initial recruitment." While this is possible, this conclusion is not well supported. First, this is a negative result, i.e. the absence of any effect on Mlp1 localization upon Nup60 depletion does not imply Nup60 has no role in Mlp1 association with the NPC. There are various reasons why a Mlp1 mislocalization phenotype is absent upon degron-induced reduction of Nup60. Perhaps fragments of Nup60 remain (or are protected) following degradation that support Mlp1 association with the NPC. In this regard, the degree to which Nup60 is degraded in these experiments is not clear as only the full-length protein is evaluated. Even some full length Nup60 appears to still be present following the induction of depletion (Fig.9). This possible scenario should be discussed, and the authors' interpretations adjusted accordingly.

We agree with the reviewer that our data do not imply Nup60 has no role in Mlp1 association with the NPC. Nup60 is essential for Mlp1 recruitment to the NPC, based on our data and that of others (Cibulka et al., 2022). However, we argue on the basis of the following points that, once Mlp1 is recruited, it can remain associated with the NPC core independently of Nup60. First, Nup60 levels are reduced by >90% within half an hour of the addition of auxin analog 5-Ph-IAA, suggesting that the vast majority of NPCs no longer contain Nup60. Since any direct role of Nup60 in maintaining Mlp1-GFP at the nuclear periphery is likely to be stoichiometric, the ability of Mlp1-GFP to remain at the nuclear periphery after Nup60 depletion is consistent with a Nup60-independent mechanism for maintenance at the nuclear periphery. Second, proteasome-mediated degradation, as is triggered with auxin-inducible degradation, results in peptide fragments of ~7 amino acids (Nussbaum et al., 1998). Consistent with this, no sizeable 3V5-containing fragments are visible in Nup60-AID immunoblots after inducing degradation (see the source data for Figure 9B). Given that the Mlp1-binding motif (MBM) in Nup60 has been described as 78 amino acids (Cibulka et al., 2022), we find it unlikely that such small proteolytic fragments would be able to bridge Mlp1 and the NPC core. Finally, the same *NUP60* allele and depletion conditions were used to examine Nup2-GFP localization. In this case, Nup60 depletion fully disrupts Nup2's association with the NPC, both at the level of recruitment and maintenance. We have revised the text for further clarification.

2) The cause of reduced binding of the *nup60-ΔAH* and *nup60-I36R* constructs to the NE/NPCs during meiotic divisions is unclear as these proteins show reduced stability. The authors conclude that the reduced NE association of these AH mutants is caused by membrane binding defects associated with the loss of the AH, and, because of this reduced NPC association, the AH mutants are unstable and degraded. However, it is just as likely that the *nup60* mutants fail to show detectable association with the NE because they are unstable and degraded during meiosis and prior to their reassociation with the reforming NPCs. Hence, the authors cannot solely conclude that the AH is required for binding to NPCs. The authors need to alter the text to reflect this possibility. As the manuscript stands, there is no evidence that the AH plays a direct role in attachment Nup60 to reforming NPCs.

Despite the impaired stability of the AH mutants, Nup60^{ΔAH}-GFP and Nup60^{I36R}-GFP remain visible throughout the meiotic divisions (Figure 7B-C), allowing for us to determine whether or not the mutants localize to the nuclear periphery. Accordingly, we can state that the AH is necessary for Nup60 association with the nuclear periphery during the meiotic divisions. However, it is unclear whether the AH is required for re-association with NPCs directly or indirectly via its interaction with NE (see our response to Reviewer 2, Point 2). The relationship between the localization and stability of Nup60 remains unclear, and so we have softened the language related to this point (line 395-396).

3) It remains unclear whether Cdc5 directly phosphorylates Nup60 or it is indirectly required for Nup60 phosphorylation. The authors should word their conclusions accordingly.

We agree with the reviewer that the current data do not allow for differentiation between whether Cdc5 directly or indirectly results in Nup60 phosphorylation. Throughout the text, we refer to Nup60 phosphorylation as “Cdc5-dependent” or “Cdc5-responsive” to communicate this point. We have also added a sentence in the discussion, making this limitation explicit (lines 540-542).

4) The authors state on line 227 that "Nup60-GFP remained associated with the nuclear periphery in *cdc5-mn cdc20-mn* cells (Figure 3D-E). Similar results were obtained using the *cdc5-mn* allele alone (Figure S3A)." The latter data are also show in Fig. 3E, and here the figure indicates a statistically significant different between the 4 and 8 hr point of in the *cdc5-mn* allele. Is this correct? Please clarify. Perhaps I missed something here.

The reviewer astutely notices that there is a significant difference between 4 hours (prophase) and 8 hours (metaphase arrest) for the *cdc5-mn* allele. However, this effect is in the opposite direction than increased detachment (what is observed for the *cdc20-mn* allele), with Nup60-GFP exhibiting increased peripheral localization at 8 hours relative to 4 hours. Since this effect is subtle (<6% change in detachment index) and not observed for *cdc5-mn cdc20-mn* cells, we find it unlikely to be biologically meaningful and so did not devote further discussion to it in the manuscript.

Citations

- Chu, D.B., Gromova, T., Newman, T.A.C., Burgess, S.M., 2017. The Nucleoporin Nup2 Contains a Meiotic-Autonomous Region that Promotes the Dynamic Chromosome Events of Meiosis. *Genetics* 206, 1319–1337. <https://doi.org/10.1534/genetics.116.194555>
- Cibulka, J., Bisaccia, F., Radisavljević, K., Gudino Carrillo, R.M., Köhler, A., 2022. Assembly principle of a membrane-anchored nuclear pore basket scaffold. *Sci Adv* 8, eabl6863. <https://doi.org/10.1126/sciadv.abl6863>
- Eastwood, M.D., Cheung, S.W.T., Lee, K.Y., Moffat, J., Meneghini, M.D., 2012. Developmentally programmed nuclear destruction during yeast gametogenesis. *Dev Cell* 23, 35–44. <https://doi.org/10.1016/j.devcel.2012.05.005>
- King, G.A., Goodman, J.S., Schick, J.G., Chetlapalli, K., Jorgens, D.M., McDonald, K.L., Ünal, E., 2019. Meiotic cellular rejuvenation is coupled to nuclear remodeling in budding yeast. *eLife* 8, e47156. <https://doi.org/10.7554/eLife.47156>
- Nussbaum, A.K., Dick, T.P., Keilholz, W., Schirle, M., Stevanović, S., Dietz, K., Heinemeyer, W., Groll, M., Wolf, D.H., Huber, R., Rammensee, H.-G., Schild, H., 1998. Cleavage motifs of the yeast 20S proteasome β subunits deduced from digests of enolase 1. *Proceedings of the National Academy of Sciences* 95, 12504–12509. <https://doi.org/10.1073/pnas.95.21.12504>
- Verzijlbergen, K.F., Menendez-Benito, V., van Welsem, T., van Deventer, S.J., Lindstrom, D.L., Ovaa, H., Neefjes, J., Gottschling, D.E., van Leeuwen, F., 2010. Recombination-induced tag exchange to track old and new proteins. *Proc Natl Acad Sci U S A* 107, 64–68. <https://doi.org/10.1073/pnas.0911164107>
- Webster, B.M., Colombi, P., Jäger, J., Lusk, C.P., 2014. Surveillance of nuclear pore complex assembly by ESCRT-III/Vps4. *Cell* 159, 388–401. <https://doi.org/10.1016/j.cell.2014.09.012>

October 11, 2022

RE: JCB Manuscript #202204039R

Dr. Elcin Unal
University of California, Berkeley
16 Barker Hall Rm 622
Berkeley, CA 94720

Dear Dr. Unal,

Thank you for submitting your revised manuscript entitled "Meiotic Nuclear Pore Complex Remodeling Provides Key Insights into Nuclear Basket Organization." We would be happy to publish your paper in JCB pending final text revisions necessary to address the remaining comment from Reviewer #3 and to meet our formatting guidelines (see details below).

A. MANUSCRIPT ORGANIZATION AND FORMATTING:

- 1) Text limits: Character count for Articles is generally < 40,000, not including spaces. Count includes title page, abstract, introduction, results, discussion, and acknowledgments. Count does not include materials and methods, figure legends, references, tables, or supplemental legends.
- 2) Figures: Articles may have up to 10 main text figures. Scale bars must be present on all microscopy images, including inset magnifications. Molecular weight or nucleic acid size markers must be included on all gel electrophoresis. Please add MW markers to blots in Figures 4E/F/G/H, 5F/G, 7G, 9B, S4E/F/G/H/I, S5A/C/E/G, & S6A/B/H.
- 3) Statistical analysis: Error bars on graphic representations of numerical data must be clearly described in the figure legend. The number of independent data points (n) represented in a graph must be indicated in the legend. Statistical methods should be explained in full in the materials and methods. For figures presenting pooled data the statistical measure should be defined in the figure legends. Please also be sure to indicate the statistical tests used in each of your experiments (both in the figure legend itself and in a separate methods section) as well as the parameters of the test (for example, if you ran a t-test, please indicate if it was one- or two-sided, etc.). Also, if you used parametric tests, please indicate if the data distribution was tested for normality (and if so, how). If not, you must state something to the effect that "Data distribution was assumed to be normal but this was not formally tested."
- 4) Materials and methods: Should be comprehensive and not simply reference a previous publication for details on how an experiment was performed. Please provide full descriptions (at least in brief) in the text for readers who may not have access to referenced manuscripts. The text should not refer to methods "...as previously described." Please also indicate the acquisition and quantification methods for immunoblotting/western blots.
- 5) For all cell lines, vectors, constructs/cDNAs, etc. - all genetic material: please include database / vendor ID (e.g., Addgene, ATCC, etc.) or if unavailable, please briefly describe their basic genetic features, even if described in other published work or gifted to you by other investigators (and provide references where appropriate). Please be sure to provide the sequences for all of your oligos: primers, si/shRNA, RNAi, gRNAs, etc. in the materials and methods. You must also indicate in the methods the source, species, and catalog numbers/vendor identifiers (where appropriate) for all of your antibodies, including secondary. If antibodies are not commercial please add a reference citation if possible.
- 6) Microscope image acquisition: The following information must be provided about the acquisition and processing of images:
 - a. Make and model of microscope
 - b. Type, magnification, and numerical aperture of the objective lenses
 - c. Temperature
 - d. Imaging medium
 - e. Fluorochromes
 - f. Camera make and model
 - g. Acquisition software
 - h. Any software used for image processing subsequent to data acquisition. Please include details and types of operations involved (e.g., type of deconvolution, 3D reconstitutions, surface or volume rendering, gamma adjustments, etc.).

7) References: There is no limit to the number of references cited in a manuscript. References should be cited parenthetically in the text by author and year of publication. Abbreviate the names of journals according to PubMed.

8) Supplemental materials: Articles generally may have up to 5 supplemental figures and 10 videos. You currently exceed this limit but, in this case, we will be able to give you the extra space but please try not to add to the current total. Please also note that tables, like figures, should be provided as individual, editable files. A summary of all supplemental material should appear at the end of the Materials and methods section. Please include one brief sentence per item.

9) Video legends: Should describe what is being shown, the cell type or tissue being viewed (including relevant cell treatments, concentration and duration, or transfection), the imaging method (e.g., time-lapse epifluorescence microscopy), what each color represents, how often frames were collected, the frames/second display rate, and the number of any figure that has related video stills or images.

10) eTOC summary: A ~40-50 word summary that describes the context and significance of the findings for a general readership should be included on the title page. The statement should be written in the present tense and refer to the work in the third person. It should begin with "First author name(s) et al..." to match our preferred style.

11) Conflict of interest statement: JCB requires inclusion of a statement in the acknowledgements regarding competing financial interests. If no competing financial interests exist, please include the following statement: "The authors declare no competing financial interests." If competing interests are declared, please follow your statement of these competing interests with the following statement: "The authors declare no further competing financial interests."

12) A separate author contribution section is required following the Acknowledgments in all research manuscripts. All authors should be mentioned and designated by their first and middle initials and full surnames. We encourage use of the CRediT nomenclature (<https://casrai.org/credit/>).

13) ORCID IDs: ORCID IDs are unique identifiers allowing researchers to create a record of their various scholarly contributions in a single place. At resubmission of your final files, please consider providing an ORCID ID for as many contributing authors as possible.

14) Please note that JCB now requires authors to submit Source Data used to generate figures containing gels and Western blots with all revised manuscripts. This Source Data consists of fully uncropped and unprocessed images for each gel/blot displayed in the main and supplemental figures. Since your paper includes cropped gel and/or blot images, please be sure to provide one Source Data file for each figure that contains gels and/or blots along with your revised manuscript files. File names for Source Data figures should be alphanumeric without any spaces or special characters (i.e., SourceDataF#, where F# refers to the associated main figure number or SourceDataFS# for those associated with Supplementary figures). The lanes of the gels/blots should be labeled as they are in the associated figure, the place where cropping was applied should be marked (with a box), and molecular weight/size standards should be labeled wherever possible. Source Data files will be directly linked to specific figures in the published article.

B. FINAL FILES:

**The license to publish form must be signed before your manuscript can be sent to production. A link to the electronic license to

publish form will be sent to the corresponding author only. Please take a moment to check your funder requirements before choosing the appropriate license.**

Thank you for this interesting contribution, we look forward to publishing your paper in Journal of Cell Biology.

Sincerely,

Martin Hetzer, PhD
Monitoring Editor
Journal of Cell Biology

Dan Simon, PhD
Scientific Editor
Journal of Cell Biology

Reviewer #2 (Comments to the Authors (Required)):

The authors have rigorously addressed my critiques, and I am happy to recommend publication of the manuscript in its current form.

Reviewer #3 (Comments to the Authors (Required)):

The authors have reasonably addressed most the points (3 of 4) raised in the previous review, with the exception of point 2. Point 2 from the initial review and the response of the authors are as follows:

Previous Reviewer point 2) The cause of reduced binding of the nup60- Δ AH and nup60-I36R constructs to the NE/NPCs during meiotic divisions is unclear as these proteins show reduced stability. The authors conclude that the reduced NE association of these AH mutants is caused by membrane binding defects associated with the loss of the AH, and, because of this reduced NPC association, the AH mutants are unstable and degraded. However, it is just as likely that the nup60 mutants fail to show detectable association with the NE because they are unstable and degraded during meiosis and prior to their reassociation with the reforming NPCs. Hence, the authors cannot solely conclude that the AH is required for binding to NPCs. The authors need to alter the text to reflect this possibility. As the manuscript stands, there is no evidence that the AH plays a direct role in attachment Nup60 to reforming NPCs.

Authors' Response: Despite the impaired stability of the AH mutants, Nup60 Δ AH-GFP and Nup60I36R-GFP remain visible throughout the meiotic divisions (Figure 7B-C), allowing for us to determine whether or not the mutants localize to the nuclear periphery. Accordingly, we can state that the AH is necessary for Nup60 association with the nuclear periphery during the meiotic divisions. However, it is unclear whether the AH is required for reassociation with NPCs directly or indirectly via its interaction with NE (see our response to Reviewer 2, Point 2). The relationship between the localization and stability of Nup60 remains unclear, and so we have softened the language related to this point (line 395-396).

Response to Authors' response: The authors state that "Despite the impaired stability of the AH mutants, Nup60 Δ AH-GFP and Nup60I36R-GFP remain visible throughout the meiotic divisions (Figure 7B-C), allowing for us to determine whether or not the mutants localize to the nuclear periphery." Since both the Nup60 Δ AH-GFP and Nup60I36R-GFP fusion proteins show significant degradation during the period of the meiotic divisions (Figure S6A-C), how do the authors conclude that the fluorescent signals they see during the meiotic division in Figure 7B-C are indeed the Nup60 Δ AH-GFP and Nup60I36R-GFP fusions and not Nup60 degradation products attached to GFP or GFP alone (which would be unable to reassociate with NPCs)? Again, since the Nup60 Δ AH-GFP and Nup60I36R-GFP fusion proteins are largely degraded, one can only conclude that the AH is required to

maintain protein stability once released from the NPC. The degradation caused by mutations in the AH precludes one from concluding that the lack of reassociation of the Nup60 Δ AH-GFP and Nup60I36R-GFP fusions with the NE (or more specifically the NPCs) is solely linked to a targeting role for the AH. The authors have failed to address this point. They state that they have softened their language on this point, but they conclude that the AH directs rebinding of Nup60 to NPCs in meiosis in the Results section, and it is front and center in the Discussion (see lines 420-430). Here they have completely ignored the concerns of degradation and they state that "redocking by the amphipathic helix results in Nup60 reattachment" to the NPC. In conclusion, I do not feel the authors have significantly addressed point 2.

Reviewer #1 (Comments to the Authors (Required)):

1. Functional significance. The function of the NPC remodeling events in meiosis is not addressed and this would also go beyond the scope of the current paper. However, did the authors observe any obvious phenotypes in the Nup60 phosphomutants or in the FKBP12-FRP tethering experiments during meiosis or germination? If no phenotypes are observed this should be discussed.

Sporulation efficiency and gamete viability are not impaired in strains with the *NUP60* alleles used in this study. This finding is neither surprising nor uncommon given that these two readouts primarily assess functions related to chromosome segregation, ascospore formation, and germination. Furthermore, even if the NPC remodeling events that we've uncovered affect one or more of these functions, redundant mechanisms might be in place and mask potential phenotypes. For example, although *nup2Δ* cells don't exhibit a sporulation efficiency defect, combining this mutation with defects in telomere bouquet formation (*ndj1Δ* or *csm4Δ*) results in almost no sporulation (Chu et al., 2017). As the reviewer pointed out, full investigation of biological significance goes beyond the scope of the current paper.

To address the reviewer's request, we have added two supplemental tables summarizing the sporulation efficiency and gamete viability data (Table S7-S8), included a corresponding statement in the discussion section (line 618-623), and added relevant strain genotypes in Table S1. As part of the discussion, we have also highlighted some possible biological roles of NPC remodeling, which we are currently investigating.

2. Cdc5 functions also during the mitotic cell cycle. Is there any indication that the nuclear basket transiently dissociates during mitosis (or in mitotic cell cycle mutants)? Does overexpression of Cdc5 in the mitotic cell cycle lead to Nup60 detachment? Alternatively, does artificial nuclear accumulation of Cdc5 promote basket detachment in mitosis (Cdc5-NLS; see Botchkarev et al, Cell Cycle 2014)?

We have included additional data on the mitotic behavior of basket nucleoporins in both *S. cerevisiae* and *S. pombe* (Supplemental Figures 2 and 9). In *S. cerevisiae*, minor detachment of Nup60 and Nup2 is observed at the onset of mitotic anaphase, consistent with increased Cdc5 activity during this time. However, the detachment occurs to a far lesser extent than in the meiotic context. This could be due to meiosis-specific regulation of Cdc5 or differences in cell cycle length that affect the level and/or duration of Cdc5 activity (e.g., G2/M is longer in meiosis than in mitosis in budding yeast). Notably, both SpNup60 and SpNup61 robustly detach at the onset of mitotic anaphase in *S. pombe*, which has a longer G2/M than *S. cerevisiae* during the mitotic cell cycle. We have added these new results in several locations throughout the manuscript: lines 135-137, lines 505-507, lines 796-801, lines 818-825, and lines 859-873.

Future work can help determine whether mitotic overexpression of Polo kinase leads to Nup60 detachment in budding yeast, and whether Polo kinase plays a role in the mitotic and meiotic NPC remodeling events in *S. pombe*.

Reviewer #2 (Comments to the Authors (Required)):

1. Nature of mislocalization of basket Nups: the manuscript describes and quantifies a displacement index of Nups into the nucleoplasm in meiosis I. Given the short timescale (~10 minutes) and short distance (nucleoplasm vs. nuclear periphery) the interpretation that these proteins transiently dissociate and then reassociate with the NPC seems reasonable in meiosis I. However, it seems clear from images that basket Nups can also be seen at a centrally located focus that resembles the GUNC in meiosis II (for example Fig. 1D,E). Given previous work (King et al 2019) indicating that the GUNC is a degradative compartment, it seems possible that the basket Nups are being degraded in the GUNC during meiosis II, and that the re-enrichment to NPCs after meiosis II could reflect newly synthesized protein.

To address the reviewer's point, we constructed an allele of *NUP2* tagged with a Recombination Inducible Tag Exchange (RITE) cassette, which allows swapping of fluorescence tags at genomic loci using an inducible Cre recombinase (Figure S3E; Verzijlbergen et al., 2010). Accordingly, pre-existing and newly synthesized pools of a protein of interest can be distinguished from one another. Using this method, we were able to confirm that, during both anaphase I and anaphase II, the same pool of Nup2 detaches from and returns to the nuclear periphery (Figure S3F-G). This is consistent with the NPC remodeling events described in our manuscript (Figure 2G). The text has been revised to include these new results: lines 197-200, lines 721-725, and lines 760-780. New strains, plasmids, primers, and imaging conditions have also been added to Table S1-S4.

We further note that GUNC-targeted proteins are degraded upon vacuolar membrane permeabilization, which releases vacuolar proteases to the cytoplasmic milieu (King et al., 2019). Vacuolar membrane permeabilization occurs several hours following anaphase II (Eastwood et al., 2012), so this timing is not consistent with the NPC remodeling events reported here.

2. Nup60 and its mammalian homolog Nup153 each have an amphipathic helix that is required for assembly of NPCs into the intact NE (Meszaros et al., Dev Cell 2015; Vollmer et al., Dev Cell 2015). The requirement for Nup60's amphipathic helix in re-enrichment of basket Nups to the nuclear periphery is suggestive of Nup60 engaging directly with the NE membrane to promote assembly of new NPCs. The authors state that the amphipathic helix mediates NPC re-association, but it seems equivalently possible that the protein is engaging directly with the membrane. Also, NPCs appear clustered in the presence of Nup60 amphipathic helix mutants after meiosis II (Figure 7C and 7D, last frames), which could indicate an NPC assembly defect. Finally, the fact that many core Nups are sequestered or degraded in the GUNC suggests that new NPC assembly may be required after meiosis. Have the authors considered that Nup60

might promote *de novo* NPC assembly after meiosis II? EM could reveal the presence of NPC assembly intermediates.

We agree with the reviewer that the amphipathic helix (AH) in Nup60 may mediate association with the nuclear periphery via direct binding to the nuclear envelope, especially during anaphase II when a large portion of NPCs are targeted to the GUNC. We maintain that the AH does promote association with individual NPCs, either directly or indirectly, as the AH is required for Nup60 localization to the nuclear periphery after anaphase I and Nup60 associates with individual NPCs at this time as seen by super-resolution microscopy (Figure 2A). We have updated the text to make more explicit the possibility that the AH functions by interacting with the nuclear envelope (lines 399-401).

To begin addressing whether Nup60 might contribute to *de novo* NPC assembly, we monitored the behavior of two core nucleoporins – Nup170-GFP, an inner ring nucleoporin, and Pom34-GFP, a transmembrane nucleoporin – in a *nup60-ΔAH* background (Supplemental Figure 7F-I). We found that core nucleoporins were sequestered to the GUNC as in wild type cells; moreover, we did not observe gross perturbations in NPC distribution around gamete nuclei (e.g., clustering as is observed in some NPC assembly mutants [Webster et al., 2014]). We updated the results to include these data in lines 437-441. New strains were also added to Table S1. We agree with the reviewer that higher resolution techniques – such as electron microscopy or super-resolution microscopy – will be very helpful in future studies to detect NPC assembly intermediates or more subtle defects. Interestingly, we observed that core nucleoporins exhibit abnormal localization during meiosis I in *nup60-ΔAH* cells, often accumulating in clusters that subsequently disperse (see Supplemental Figure 7I for an example). The composition and formation of these NPC clusters requires further characterization; as such, we have opted not to include discussion of these results in the manuscript and are instead currently following up with additional experiments.

Reviewer #3 (Comments to the Authors (Required)):

Recommendations for the authors

1) On the basis of their Nup60-degron experiments and the analysis of cells with reduced Nup60, the authors conclude that (line 472) "Mlp1 can ... remain associated with the NPC core independently of Nup60 after its initial recruitment." While this is possible, this conclusion is not well supported. First, this is a negative result, i.e. the absence of any effect on Mlp1 localization upon Nup60 depletion does not imply Nup60 has no role in Mlp1 association with the NPC. There are various reasons why a Mlp1 mislocalization phenotype is absent upon degron-induced reduction of Nup60. Perhaps fragments of Nup60 remain (or are protected) following degradation that support Mlp1 association with the NPC. In this regard, the degree to which Nup60 is degraded in these experiments is not clear as only the full-length protein is evaluated. Even some full length Nup60 appears to still be present following the induction of depletion (Fig.9). This possible scenario should be discussed, and the authors' interpretations adjusted accordingly.

We agree with the reviewer that our data do not imply Nup60 has no role in Mlp1 association with the NPC. Nup60 is essential for Mlp1 recruitment to the NPC, based on our data and that of others (Cibulka et al., 2022). However, we argue on the basis of the following points that, once Mlp1 is recruited, it can remain associated with the NPC core independently of Nup60. First, Nup60 levels are reduced by >90% within half an hour of the addition of auxin analog 5-Ph-IAA, suggesting that the vast majority of NPCs no longer contain Nup60. Since any direct role of Nup60 in maintaining Mlp1-GFP at the nuclear periphery is likely to be stoichiometric, the ability of Mlp1-GFP to remain at the nuclear periphery after Nup60 depletion is consistent with a Nup60-independent mechanism for maintenance at the nuclear periphery. Second, proteasome-mediated degradation, as is triggered with auxin-inducible degradation, results in peptide fragments of ~7 amino acids (Nussbaum et al., 1998). Consistent with this, no sizeable 3V5-containing fragments are visible in Nup60-AID immunoblots after inducing degradation (see the source data for Figure 9B). Given that the Mlp1-binding motif (MBM) in Nup60 has been described as 78 amino acids (Cibulka et al., 2022), we find it unlikely that such small proteolytic fragments would be able to bridge Mlp1 and the NPC core. Finally, the same *NUP60* allele and depletion conditions were used to examine Nup2-GFP localization. In this case, Nup60 depletion fully disrupts Nup2's association with the NPC, both at the level of recruitment and maintenance. We have revised the text for further clarification.

2) The cause of reduced binding of the nup60- Δ AH and nup60-I36R constructs to the NE/NPCs during meiotic divisions is unclear as these proteins show reduced stability. The authors conclude that the reduced NE association of these AH mutants is caused by membrane binding defects associated with the loss of the AH, and, because of this reduced NPC association, the AH mutants are unstable and degraded. However, it is just as likely that the nup60 mutants fail to show detectable association with the NE because they are unstable and degraded during meiosis and prior to their reassociation with the reforming NPCs. Hence, the authors cannot solely conclude that the AH is required for binding to NPCs. The authors need to alter the text to reflect this possibility. As the manuscript stands, there is no evidence that the AH plays a direct role in attachment Nup60 to reforming NPCs.

Despite the impaired stability of the AH mutants, Nup60 ^{Δ AH}-GFP and Nup60^{I36R}-GFP remain visible throughout the meiotic divisions (Figure 7B-C), allowing for us to determine whether or not the mutants localize to the nuclear periphery. Accordingly, we can state that the AH is necessary for Nup60 association with the nuclear periphery during the meiotic divisions. However, it is unclear whether the AH is required for re-association with NPCs directly or indirectly via its interaction with NE (see our response to Reviewer 2, Point 2). The relationship between the localization and stability of Nup60 remains unclear, and so we have softened the language related to this point (line 395-396).

Response to Authors' response: The authors state that "Despite the impaired stability of the AH mutants, Nup60 Δ AH-GFP and Nup60I36R-GFP remain visible throughout the meiotic divisions (Figure 7B-C), allowing for us to determine whether or not the mutants

localize to the nuclear periphery." Since both the Nup60 Δ AH-GFP and Nup60I36R-GFP fusion proteins show significant degradation during the period of the meiotic divisions (Figure S6A-C), how do the authors conclude that the fluorescent signals they see during the meiotic division in Figure 7B-C are indeed the Nup60 Δ AH-GFP and Nup60I36R-GFP fusions and not Nup60 degradation products attached to GFP or GFP alone (which would be unable to reassociate with NPCs)? Again, since the Nup60 Δ AH-GFP and Nup60I36R-GFP fusion proteins are largely degraded, one can only conclude that the AH is required to maintain protein stability once released from the NPC. The degradation caused by mutations in the AH precludes one from concluding that the lack of reassociation of the Nup60 Δ AH-GFP and Nup60I36R-GFP fusions with the NE (or more specifically the NPCs) is solely linked to a targeting role for the AH. The authors have failed to address this point. They state that they have softened their language on this point, but they conclude that the AH directs rebinding of Nup60 to NPCs in meiosis in the Results section, and it is front and center in the Discussion (see lines 420-430). Here they have completely ignored the concerns of degradation and they state that "redocking by the amphipathic helix results in Nup60 reattachment" to the NPC. In conclusion, I do not feel the authors have significantly addressed point 2.

We find it unlikely that the changes observed in localization for the Nup60 lipid-binding mutants are due to degradation or truncation, for the following reasons. (1) If Nup60 Δ AH-GFP or Nup60^{I36R}-GFP were degraded via autophagy, we would expect to observe GFP signal within the vacuole. Because GFP is relatively resistant to vacuolar degradation, autophagic degradation of a GFP-tagged protein leads to the accumulation of free GFP in the vacuole (Kanki and Klionsky, 2008). However, the GFP signal clearly remains nucleoplasmic. Therefore, autophagy is unlikely to be the mechanism driving relocalization for the time points used in image quantification. (2) Proteasome-dependent degradation, on the other hand, is expected to generate peptide fragments of ~7 amino acids (Nussbaum et al., 1998); these are unlikely to be detected as a nuclear signal, since GFP is far larger than this fragment size. (3) A third possibility is that a cleavage fragment of Nup60 Δ AH-GFP or Nup60^{I36R}-GFP is generated such that this fragment carries both GFP and a nuclear localization signal (NLS). However, we are unable to detect such a fragment that is specific to strains carrying Nup60 Δ AH-GFP (Source Data S6A) or Nup60^{I36R}-GFP (Source Data S6B). There is a smaller fragment (~40kDa) that is detectable by 10h for both alleles, but this band is also present in the wild-type strains. Additionally, this time point is much later than when we perform the image analysis to quantify localization. In conclusion, we currently have no evidence that the nuclear GFP signal we detect *in vivo* is due to a degradation product of Nup60 Δ AH-GFP or Nup60^{I36R}-GFP. We have edited the text to include some of this rationale (lines 396-399).

3) It remains unclear whether Cdc5 directly phosphorylates Nup60 or it is indirectly required for Nup60 phosphorylation. The authors should word their conclusions accordingly.

We agree with the reviewer that the current data do not allow for differentiation between whether Cdc5 directly or indirectly results in Nup60 phosphorylation. Throughout the

text, we refer to Nup60 phosphorylation as “Cdc5-dependent” or “Cdc5-responsive” to communicate this point. We have also added a sentence in the discussion, making this limitation explicit (lines 542-544).

4) The authors state on line 227 that "Nup60-GFP remained associated with the nuclear periphery in *cdc5-mn cdc20-mn* cells (Figure 3D-E). Similar results were obtained using the *cdc5-mn* allele alone (Figure S3A)." The latter data are also show in Fig. 3E, and here the figure indicates a statistically significant different between the 4 and 8 hr point of in the *cdc5-mn* allele. Is this correct? Please clarify. Perhaps I missed something here.

The reviewer astutely notices that there is a significant difference between 4 hours (prophase) and 8 hours (metaphase arrest) for the *cdc5-mn* allele. However, this effect is in the opposite direction than increased detachment (what is observed for the *cdc20-mn* allele), with Nup60-GFP exhibiting increased peripheral localization at 8 hours relative to 4 hours. Since this effect is subtle (<6% change in detachment index) and not observed for *cdc5-mn cdc20-mn* cells, we find it unlikely to be biologically meaningful and so did not devote further discussion to it in the manuscript.

Citations

- Chu, D.B., Gromova, T., Newman, T.A.C., Burgess, S.M., 2017. The Nucleoporin Nup2 Contains a Meiotic-Autonomous Region that Promotes the Dynamic Chromosome Events of Meiosis. *Genetics* 206, 1319–1337. <https://doi.org/10.1534/genetics.116.194555>
- Cibulka, J., Bisaccia, F., Radisavljević, K., Gudino Carrillo, R.M., Köhler, A., 2022. Assembly principle of a membrane-anchored nuclear pore basket scaffold. *Sci Adv* 8, eabl6863. <https://doi.org/10.1126/sciadv.abl6863>
- Eastwood, M.D., Cheung, S.W.T., Lee, K.Y., Moffat, J., Meneghini, M.D., 2012. Developmentally programmed nuclear destruction during yeast gametogenesis. *Dev Cell* 23, 35–44. <https://doi.org/10.1016/j.devcel.2012.05.005>
- Kanki, T., Klionsky, D.J., 2008. Mitophagy in yeast occurs through a selective mechanism. *J Biol Chem* 283, 32386–32393. <https://doi.org/10.1074/jbc.M802403200>
- King, G.A., Goodman, J.S., Schick, J.G., Chetlapalli, K., Jorgens, D.M., McDonald, K.L., Ünal, E., 2019. Meiotic cellular rejuvenation is coupled to nuclear remodeling in budding yeast. *eLife* 8, e47156. <https://doi.org/10.7554/eLife.47156>
- Nussbaum, A.K., Dick, T.P., Keilholz, W., Schirle, M., Stevanović, S., Dietz, K., Heinemeyer, W., Groll, M., Wolf, D.H., Huber, R., Rammensee, H.-G., Schild, H., 1998. Cleavage motifs of the yeast 20S proteasome β subunits deduced from digests of enolase 1. *Proceedings of the National Academy of Sciences* 95, 12504–12509. <https://doi.org/10.1073/pnas.95.21.12504>
- Verzijlbergen, K.F., Menendez-Benito, V., van Welsem, T., van Deventer, S.J., Lindstrom, D.L., Ovaa, H., Neefjes, J., Gottschling, D.E., van Leeuwen, F., 2010. Recombination-induced tag exchange to track old and new proteins. *Proc Natl Acad Sci U S A* 107, 64–68. <https://doi.org/10.1073/pnas.0911164107>
- Webster, B.M., Colombi, P., Jäger, J., Lusk, C.P., 2014. Surveillance of nuclear pore complex assembly by ESCRT-III/Vps4. *Cell* 159, 388–401. <https://doi.org/10.1016/j.cell.2014.09.012>